# Non asymptotic analysis of Adaptive stochastic gradient algorithms and applications

**Antoine Godichon-Baggioni**                                                                *antoine.godichon__baggioni@sorbonne-universite.fr*
*Laboratoire de Probabilités, Statistique et Modélisation*
*Sorbonne Université*

**Pierre Tarrago**                                                                                *pierre.tarrago@sorbonne-universite.fr*
*Laboratoire de Probabilités, Statistique et Modélisation*
*Sorbonne Université*

**Reviewed on OpenReview:** *https: // openreview. net/ forum? id= iyfbGyAkKt*

## Abstract

In stochastic optimization, a widely used approach for handling large samples sequentially is the stochastic gradient algorithm (SGD). However, a key limitation of SGD is that its step size sequence remains uniform across all gradient directions, which can lead to poor performance in practice, particularly for ill-conditioned problems. To address this issue, adaptive gradient algorithms, such as Adagrad and stochastic Newton methods, have been developed. These algorithms adapt the step size to each gradient direction, providing significant advantages in such challenging settings. This paper focuses on the non-asymptotic analysis of these adaptive gradient algorithms for strongly convex objective functions. The theoretical results are further applied to practical examples, including linear regression and regularized generalized linear models, using both Adagrad and stochastic Newton algorithms.

**Keywords:** Non asymptotic analysis; Online estimation; Adaptive gradient algorithm; Adagrad; Stochastic Newton algorithm.

## 1 Introduction

A usual problem in stochastic optimization is to estimate the minimizer $\theta$ of a convex functional $G : \mathbb{R}^d \longrightarrow \mathbb{R}$ of the form

$$G(h) = \mathbb{E}\left[g(X, h)\right]$$

where $g : \mathcal{X} \times \mathbb{R}^d \longrightarrow \mathbb{R}$, and $X$ is an $\mathcal{X}$-valued random variable. This framework encompasses numerous classical problems, such as linear and logistic regression (Bach, 2014), or the estimation of geometric medians and quantiles (Cardot et al., 2013; 2015; Godichon-Baggioni, 2016) to name a few. Various methods have been developed to solve this optimization problem, generally categorized into iterative and recursive approaches. Iterative methods involve approximating the minimizer of an empirical function derived from the sample using convex optimization techniques (Boyd & Vandenberghe, 2004), or more advanced refinements like mini-batch algorithms (Konečný et al., 2015). While effective, these methods face scalability issues with large datasets and are unsuitable for sequential data. In contrast, recursive methods adapt naturally to sequential data and are computationally efficient.

Among recursive methods, the stochastic gradient algorithm (SGD) (Robbins & Monro, 1951) and its averaged version (Ruppert, 1988; Polyak & Juditsky, 1992) are particularly well-known. Given sequential data $X_1, \ldots, X_n, X_{n+1}, \ldots$, the stochastic gradient algorithm $(\theta_n)_{n\geq 0}$ and its averaged version $(\bar{\theta}_n)_{n\geq 0}$ are defined recursively for all $n \geq 0$ by

$$\theta_{n+1} = \theta_n - \gamma_{n+1}\nabla_h g\left(X_{n+1}, \theta_n\right), \qquad\qquad \bar{\theta}_{n+1} = \bar{\theta}_n + \frac{1}{n+2}\left(\theta_{n+1} - \bar{\theta}_n\right)$$

where $(\gamma_n)$ is a positive step sequence converging to 0. These algorithms have been extensively studied, with asymptotic results in works like (Pelletier, 1998; 2000) and non-asymptotic results focusing on quadratic mean convergence (Bach & Moulines, 2013; Gadat & Panloup, 2017; Gower et al., 2019). Averaged estimates are particularly appealing as they achieve asymptotic efficiency under regularity conditions, often attaining the Cramer-Rao bound (up to negligible terms).

Despite these advantages, a significant limitation of SGD lies in its step size sequence $(\gamma_n)$, which cannot adapt to varying gradient directions, leading to suboptimal performance in ill-conditioned problems. This issue has motivated the development of adaptive stochastic gradient algorithms. These methods take the form:
$$\theta_{n+1} = \theta_n - \gamma_{n+1} A_n \nabla_h g\left(X_{n+1}, \theta_n\right)$$
where $(A_n)$ is a sequence of (random) matrices which enables the descent step to be adapted in each coordinate. Prominent examples include Adagrad (Duchi et al., 2011), which effectively standardizes the gradient, and stochastic Newton algorithms that use estimates of the inverse Hessian (Bercu et al., 2020; Boyer & Godichon-Baggioni, 2020). These methods are particularly advantageous when the Hessian has eigenvalues of different magnitudes.

While asymptotic properties of adaptive methods are well-studied (e.g., (Leluc & Portier, 2020; Gadat & Gavra, 2020)), non-asymptotic results remain less explored. Notable exceptions include high-probability bounds for Kalman recursions in logistic regression (De Vilmarest & Wintenberger, 2021) and $L^2$ convergence rates for Adagrad and Adam (Défossez et al., 2020). Furthermore, Bercu et al. (2021) obtain the rate of convergence in quadratic mean of stochastic Gauss-Newton algorithms for optimal transport. However, these results often assume uniformly bounded gradients, a condition violated in cases such as linear regression.

This paper addresses these gaps by focusing on non-asymptotic convergence rates for strongly convex functions with unbounded gradients. Our contributions include: (i) establishing convergence rates for adaptive methods when $A_n$ may diverge, with a controlled divergence bound, (ii) deriving standard convergence rates under the additional assumption that $A_n$ has uniformly bounded fourth-order moments, (iii) providing a general framework for analyzing the convergence of stochastic Newton and Adagrad algorithms, with applications to linear regression and ridge-regularized generalized linear models.

The paper is organized as follows: Section 2 introduces the general framework. Section 3 presents the algorithms and theoretical convergence results. Applications to linear regression and generalized linear models are detailed in Sections 4 and 5, respectively. Proofs are provided in Section 7 and the Appendix.

## 2   Framework

In what follows, we consider a random variable $X$ taking values in a measurable space $\mathcal{X}$ and fix $d \geq 2$. We focus on the estimation of the minimizer $\theta$ of a strongly convex function $G : \mathbb{R}^d \longrightarrow \mathbb{R}$ defined for all $h \in \mathbb{R}^d$ by
$$G(h) := \mathbb{E}\left[g\left(X, h\right)\right],$$
with $g : \mathcal{X} \times \mathbb{R}^d \longrightarrow \mathbb{R}$. Let us suppose from now on that the following assumptions are fulfilled:

**(A1)** For almost every $x \in \mathcal{X}$ with respect to the distribution of $X$, the functional $g(x, .)$ is differentiable on $\mathbb{R}^d$. Moreover, there exist $p \geq 2$ and non-negative constants $C_1^{(p)}$, $C_2^{(p)}$ such that for all $h \in \mathbb{R}^d$,

$$\mathbb{E}\left[\|\nabla_h g\left(X, h\right)\|^{2p}\right] \leq C_1^{(p)} + C_2^{(p)} \|h - \theta\|^{2p}.$$

**(A2)** The functional $G$ is twice continuously differentiable.

**(A3)** The Hessian of $G$ is uniformly bounded on $\mathbb{R}^d$, i.e there is a positive constant $L_{\nabla G}$ such that for all $h \in \mathbb{R}^d$,
$$\left\|\nabla^2 G(h)\right\|_{op} \leq L_{\nabla G}$$
where $\|.\|_{op}$ is the usual spectral norm for matrices.

**(A4)** There exists $\mu > 0$ such that the functional $G$ is $\mu$-strongly convex : for all $h, h' \in \mathbb{R}^d$,

$$G(h') - G(h) \geq \nabla G(h)^T (h' - h) + \frac{\mu}{2} \|h' - h\|^2.$$

Let us now discuss these assumptions. First, note that Assumption **(A1)**, for $p = 1$, ensures that the second-order moment of the gradient is bounded by a constant and a term that, at worst, grows quadratically—an assumption already present in the literature (Bach & Moulines, 2013). When combined with Assumption **(A3)**, this can be linked to the expected smoothness condition (Gower et al., 2019). Remark that we require a higher order of boundedness in Assumption **(A1)**, compared for example to (Bach & Moulines, 2013), because of the presence of the random preconditioning $A_n$. Assumption **(A2)** justifies the use of gradient-based algorithms, while twice differentiability is crucial for Newton-type methods.

It is worth noting that Theorems 3.1 and 3.2, which only concern the convergence of $G(\theta_n)$, remain valid even without assuming strong convexity or the convexity of the function $G$, provided that $G$ satisfies the Polyak–Lojasiewicz condition (see Guo et al. (2025); Gower et al. (2021); Karimi et al. (2016), among others). That is, if the following condition holds:

**(A4')** There is $\mu > 0$ such that for all $h \in \mathbb{R}^d$,

$$\|\nabla G(h)\|^2 \geq 2\mu \left( G(h) - G(\theta) \right).$$

Theorem 3.3 however needs strong convexity in order to translate the convergence of $G(\theta_n)$ into a convergence of $\theta_n$. In the case where the convergence of $A_n$ only requires the one of $G(\theta_n)$, **(A4)** can be replaced by **(A4')** for Theorem 3.3 to hold with $\theta_n$ replaced by $G(\theta_n)$.

## 3 Adaptive stochastic gradient algorithms

### 3.1 The algorithms

Let $X_1, \ldots, X_n, X_{n+1}, \ldots$ be an i.i.d sequence of random variables with the same distribution as $X$. Then, an adaptive stochastic gradient algorithm is defined recursively for all $n \geq 0$ by

$$\theta_{n+1} = \theta_n - \gamma_{n+1} A_n \nabla_h g \left( X_{n+1}, \theta_n \right),$$

where $\theta_0$ is arbitrarily chosen, $\gamma_n = c_\gamma n^{-\gamma}$ with $c_\gamma > 0$, $\gamma \in (0, 1)$ and $A_n$ is a sequence of symmetric and positive matrices such that there is a filtration $(\mathcal{F}_n)_{n \geq 0}$ satisfying:

- For all $n \geq 0$, $A_n$ is $\mathcal{F}_n$-measurable.

- $X_{n+1}$ is independent of $\mathcal{F}_n$.

Typically, one can consider $A_n$ only depending on $X_1, \ldots, X_n, \theta_0, \ldots, \theta_n$ and consider the filtration generated by the sample, i.e $\mathcal{F}_n = \sigma \left( X_1, \ldots, X_n \right)$. Considering $A_n$ diagonal with $(A_n)_{k,k} = \left( \frac{1}{n+1} \left( a_k + \sum_{i=1}^n \nabla_h g \left( X_i, \theta_{i-1} \right)_{i,i}^2 \right) \right)^{-1/2}$ leads to Adagrad algorithm (Duchi et al., 2011). The case where $A_n$ is a recursive estimate of the inverse of the Hessian corresponds then to the stochastic Newton algorithm (Bercu et al., 2020; Boyer & Godichon-Baggioni, 2020), while the case where $A_n = \frac{1}{n+1} \left( \left( A_0 + \sum_{i=1}^n \nabla_h g \left( X_i, \theta_{i-1} \right) \nabla_h g \left( X_i, \theta_{i-1} \right)^T \right) \right)^{-1}$ corresponds to the stochastic Gauss-Newton algorithm (Cénac et al., 2020; Bercu et al., 2021).

### 3.2 Convergence results

#### 3.2.1 A first convergence result

In order to obtain a first rate of convergence of the estimates, let us now introduce some assumptions on the sequence of random matrices $(A_n)_{n \geq 0}$:

**(H1 )** One can control the smallest and largest eigenvalues of $A_n$:

**(H1a)** There exist $(v_n)_{n\geq 0}, \lambda_0 > 0$ and $\delta, q \geq 0$ such that

$$\mathbb{P}\left[\lambda_{\min}(A_n) \leq \lambda_0 t\right] \leq v_{n+1} t^q (n+1)^{-\delta},$$

for $0 < t \leq 1$, with $(v_{n+1}(n+1)^{-\delta})_{n\geq 0}$ decreasing.

If $\gamma \leq 1/2$, one also assumes the stronger hypothesis of the existence of $\lambda'_n = \lambda'_0(n+1)^{-\lambda'}$ with $\lambda'_0 > 0$, $\lambda' < \gamma$ such that for all $n \geq 0$,

$$\lambda_{\min}(A_n) \geq \lambda'_n.$$

**(H1b)** There exists a sequence $\beta_n = c_\beta n^\beta$ for $n \geq 0$ with $c_\beta \geq 0$ and $0 < \beta < \frac{\gamma}{2}$ if $\gamma \leq 1/2$ or $0 < \beta < \gamma - 1/2$ if $\gamma > 1/2$ such that for all $n \geq 0$,

$$\|A_n\|_{op} \leq \beta_{n+1}.$$

Remark that the case $\delta = 0$ is allowed in **(H1a)** and that one can always choose $\beta$ in the allowed range of **(H1b)**. In most cases and especially for Adagrad and stochastic Newton algorithm, **(H1a)** is easily verified. The presence of the decreasing term $v_n$ in **(H1a)** takes into account a general phenomenon (usually implied by Rosenthal inequality) that error contributions from higher moments of $X$, albeit dominant for small $n$, fade as $n$ goes to infinity. Concerning **(H1b)**, some counter-examples showing that the estimates possibly diverge in the case where this last assumption is not fulfilled are given in Appendix F, meaning that this assumption is unfortunately crucial. Up to our knowledge, it is still an open problem to know whether such assumption can be lifted in the specific case of the linear or logistic regression. However, it is apparent in the proofs and counterexamples that the failing of the convergence in quadratic mean is due to the exponential explosion of the algorithm on an event of negligible probability. It is then still possible to deduce good quadratic bounds on the convergence of $\theta_n$ even without Assumption **(H1b)**, see for example Corollary 4.1 for such a result for the linear regression.

Anyway, an easy way to enforce Assumption **(H1b)** is to replace the random matrices $A_n$ by

$$\tilde{A}_n = \frac{\min\left\{\|A_n\|_{op}, \beta_{n+1}\right\}}{\|A_n\|_{op}} A_n$$

and one can directly check that $\|\tilde{A}_n\|_{op} \leq \beta_{n+1}$. Similar adjustment can be used to ensure **(H1a)** in the case $\gamma \leq 1/2$. It is worth noting that when the matrix $A_n$ is not diagonal, computing the operator norm can be too costly. In this case, we instead consider the Frobenius norm. This does not affect the results, and the proofs remain strictly analogous.

Let us consider the case of Newton's method, and especially the case where the estimates of the Hessian are of the form $H_n = \frac{1}{n+1}\left(H_0 + \sum_{k=1}^n a_k \Phi_k \Phi_k^T\right)$ and which can be so recursively invert with the help of Riccati/Shermann-Morrisson's formula (see Bercu et al. (2020); Boyer & Godichon-Baggioni (2020); Godichon-Baggioni et al. (2022)), Assumption **(H1b)** can also be enforced by considering the following version of the estimate of the Hessian

$$\tilde{H}_n = H_n + \frac{1}{n+1}\sum_{k=1}^n \frac{\tilde{c}_\beta}{k^\beta} e_k e_k^T$$

where $e_k$ is the $k$-th (modulo $d$) canonical vector (see Bercu et al. (2021); Godichon-Baggioni et al. (2022)).

Regardless, these assumptions remain less restrictive than the standard ones in the literature. For instance, in the literature on stochastic Newton algorithms (see Byrd et al. (2016); Ye et al. (2017); Agarwal et al. (2017), among others), it is often assumed that Hessian estimators are uniformly bounded from above and below, implying that their inverses are also uniformly bounded.

Similarly, in the literature on Adagrad, a penalty term is introduced to upper bound the largest eigenvalue of $A_n$. This corresponds in our framework to setting $a_k = \eta(n+1)$ (see equation 2), where $\eta$ is the penalty : this would amounts to choose the strictest hypothesis $\beta = 0$ in Assumption **(H1b)**. Likewise, it is generally assumed that gradients are bounded in order to ensure a uniform lower bound on $A_n$ (see Défossez et al. (2020) and Duchi et al. (2011), for example) : this would then amounts to choose $v_{n=1} = 0$ in Assumption **(H1a)**. It is worth noting that in Guo et al. (2025), gradients are not assumed to be bounded, yet the "regularization" term is still present.

Finally, one can examine the proofs of Theorems 4.1, 4.2, 5.1, and 5.2 to understand how these assumptions can be verified in practice.

We can now obtain a first rate of convergence of the estimates. For the sake of simplicity, let us now denote the risk error by $V_n := G(\theta_n) - G(\theta)$. Note that since $G$ is $\mu$ strongly convex, one has $\|\theta_n - \theta\|^2 \leq \frac{2}{\mu} V_n$.

**Theorem 3.1.** *Suppose Assumptions **(A1)** to **(A4)** and **(H1)** hold. Then, for all $n \geq 1$ and for any $\lambda < \min\{\gamma - 2\beta, 1 - \gamma\}$,*

$$
\mathbb{E}\left[V_n\right] \leq \exp\left(-c_\gamma \mu \lambda_0 n^{1-(\lambda+\gamma)}(1-\varepsilon(n))\right)\left(K_1^{(1)} + K_{1'}^{(1)} \max_{1 \leq k \leq n+1} k^{\gamma-2\beta-\delta/2-(q/2+1)\lambda}\sqrt{v_k}\right)
$$
$$
+ K_2^{(1)} n^{-(\gamma-2\beta-\lambda)} + K_3^{(1)}\sqrt{v_{\lfloor n/2\rfloor}}\, n^{-(\delta+q\lambda)/2},
$$

*with $\varepsilon(n) = o(1)$ given in equation 19 and $K_1^{(1)}$, $K_{1'}^{(1)}$, $K_2^{(1)}$, $K_3^{(1)}$ constants respectively given in equation 20 and equation 21.*

In the particular case where $\delta/2 \geq \gamma - 2\beta$ (which happens as soon as $\delta \geq 1$), one can simply set $\lambda = 0$ in the above formula : we will see that it is the case for the generalized linear model with the stochastic Newton algorithm. However, for Adagrad algorithms, one can not avoid using first $\lambda > 0$, since $A_n$ depends on $\nabla g(X, \cdot)$ rather than $\nabla^2 g(X, \cdot)$ (while the expectation of the latter is bounded on $\mathbb{R}^d$, the one of the former is generally unbounded). To get rid of this weaker statement, we will need the following equivalent of Theorem 3.1 for higher moments.

**Proposition 3.1.** *Suppose that Assumptions **(A1)** with $p > 2$, **(A2)** to **(A3)** and **(H1)** hold. Then for any $2 \leq p' < p$ and any $\lambda < \min\{\gamma - 2\beta, 1 - \gamma\}$,*

$$
\mathbb{E}\left[V_n^{p'}\right] \leq \exp\left(-c_\gamma \mu \lambda_0 n^{1-(\lambda+\gamma)}(1-\varepsilon'(n))\right)\left(K_1^{(1')} + K_{1'}^{(1')} \max_{1 \leq k \leq n+1} k^{\gamma-2\beta-\lambda-\frac{p-p'}{p}(\delta+q\lambda)} v_k^{\frac{p-p'}{p}}\right)
$$
$$
+ K_2^{(1')} n^{-p'(\gamma-2\beta-\lambda)} + K_3^{(1')} v_{\lfloor n/2\rfloor}^{\frac{p-p'}{p}}(n+1)^{-\frac{p-p'}{p}(\delta+q\lambda)},
$$

*with $\epsilon'(n)$, $K_1^{(1')}$, $K_{1'}^{(1')}$, $K_2^{(1')}$ and $K_3^{(1')}$ constants respectively given in equation 67, equation 68 and equation 70.*

### 3.2.2 Convergence when $A_n$ has bounded moments

In order to get a better rate of convergence, let us now introduce some new assumptions on the sequence of random matrices $(A_n)$:

**(H2a)** The random matrices $A_n$ admit uniformly bounded second order moments. There exists $C_S > 0$ such that for all $n \geq 0$:
$$
\mathbb{E}\left[\|A_n\|^2\right] \leq C_S^2.
$$

**(H2b)** The random matrices $A_n$ admit uniformly bounded fourth order moments. There exists $C_S > 0$ (which can be taken equal to the one of **(H2a)**, up to increasing the latter) such that for all $n \geq 0$:
$$
\mathbb{E}\left[\|A_n\|^4\right] \leq C_S^4.
$$

For a simpler statement, we assume here and in the next paragraph that $q > 0$ in **(H1a)**, although similar bound would hold in full generality.

**Theorem 3.2.** *Suppose Assumptions **(A1)** to **(A4)** for some $p > 2$, **(H1)** and **(H2a)** hold with $\delta > 0$. Then, for all $n \geq 0$,*

$$\mathbb{E}\left[V_n\right] \leq \exp\left(-c_\gamma \mu \lambda_0 n^{1-\gamma}\left(1 - \varepsilon(n)\right)\right) \cdot \left(K_1^{(2)} + K_{1'}^{(2)} \max_{1 \leq k \leq n+1} v_k^{\frac{p-1}{p}} k^{\gamma - 2\beta - \frac{p-1}{p}\delta}\right)$$
$$+ K_2^{(2)} v_{\lfloor n/2 \rfloor}^{\frac{p-1}{p}} n^{-\frac{(p-1)}{p}\delta} + K_3^{(2)} n^{-\gamma},$$

*where $\varepsilon(n) = o(1)$ is given in equation 24 and $K_1^{(2)}, K_{1'}^{(2)}, K_2^{(2)}, K_3^{(2)}$ are constants respectively given in equation 25, equation 26 and equation 27.*

It is worth noting that, using calculations analogous to ours and those in Bach & Moulines (2013), a bound in the case of the online stochastic gradient algorithm would take the form

$$\mathbb{E}\left[V_n\right] \leq K_0 e^{-k_0 \mu n^{1-\gamma}} + K_1 n^{-\gamma}.$$

Thus, we obtain similar bounds, but with the additional presence of the term $\lambda_0$ in the exponent, which arises from the smallest eigenvalue of the conditioning matrix, as well as the term $v_n$, which controls the extent to which our conditioning matrix cannot be smaller than this value.

Similar to the gradient method, we obtain a bound with two phases: (i) an exponential decay for the initialization error, followed by (ii) a convergence rate of order $n^{-\gamma}$ once the algorithm has stabilized. One might be tempted to choose $\gamma = 1$, but this would result in the loss of exponential decay (or an increase in variance). Thus, finding an optimal trade-off for the choice of $\gamma$ is challenging. In any case, a common approach to accelerate convergence is to introduce an averaging step (see Ruppert (1988), Polyak & Juditsky (1992), Boyer & Godichon-Baggioni (2020), among others).

Finally, in order to get the rate of convergence in quadratic mean of stochastic Newton estimates, we now give the $L^2$ rate of convergence of $G(\theta_n)$ when $\gamma > 1/2$.

**Proposition 3.2.** *Suppose Assumptions **(A1)** to **(A4)** for some $p > 2$, **(H1)** and **(H2b)** hold with $\gamma > 1/2, \delta > 0$ and $\beta < \gamma - 1/2$. Then*

$$\mathbb{E}\left[V_n^2\right] \leq \exp\left(-\frac{3}{2}c_\gamma \lambda_0 \mu n^{1-\gamma}\right)\left(K_1^{(2')} + K_{1'}^{(2')} \max_{1 \leq k \leq n+1} v_k^{\frac{p-2}{p}} k^{\gamma - \frac{p-2}{p}\delta}\right)$$
$$+ K_2^{(2')} n^{-2\gamma} + K_3^{(2')} v_{\lfloor n/2 \rfloor}^{(p-2)/p} n^{-\delta(p-2)/p} =: M_n.$$

*with $K_1^{(2')}, K_{1'}^{(2')}, K_2^{(2')}, K_3^{(2')}$ constants respectively given in equation 73, equation 74 and equation 75.*

In other words, one has $M_n = O\left(n^{-\min\left\{2\gamma, \frac{\delta(p-2)}{p}\right\}}\right)$. Hence, for $\delta$ large enough (namely $\delta > \frac{2p}{p-2}\gamma$), the main contribution comes from the second term of the latter bound, i.e we obtain the good rate of convergence $O(n^{-\gamma})$.

### 3.2.3 Convergence results for stochastic Newton algorithms

Let us now focus on the rate of convergence of stochastic Newton algorithm. To this end, let us set $H := \nabla^2 G(\theta)$ and suppose from now that the following assumptions are also fulfilled:

**(A1')** There exists $L_{\nabla g} > 0$ such that for all $h \in \mathbb{R}^d$,

$$\mathbb{E}\left[\left\|\nabla_h g\left(X, h\right) - \nabla_h g\left(X, \theta\right)\right\|^2\right] \leq L_{\nabla g}\left\|h - \theta\right\|^2 \tag{1}$$

**(A5)** There is a non negative constant $L_\delta$ such that for all $h \in \mathbb{R}^d$,

$$\left\|\nabla G(h) - \nabla^2 G(\theta)\left(\theta - h\right)\right\| \leq L_\delta \left\|h - \theta\right\|^2$$

**(H3)** The estimate $A_n$ converges to $H^{-1}$: there is a decreasing positive sequence $(v_{A,n})_{n\geq 0}$ such that for al $n \geq 0$,

$$\mathbb{E}\left[\left\|A_n - H^{-1}\right\|^2\right] \leq v_{A,n}.$$

Observe that assumption **(A1')** is often called expected smoothness in the literature (Bach & Moulines, 2013) and is satisfied in most of examples such as linear and logistic regressions (Bach & Moulines, 2013; Bach, 2014) or the estimation of geometric quantiles and medians (Cardot et al., 2013) among others. Concerning **(A5)**, under **(A3)**, it is satisfied as soon as the Hessian is Lipschitz on a neighborhood of $\theta$. For instance, in the case of the linear regression, $L_\delta = 0$. Finally, Assumption **(H3)** is satisfied if having a first rate of convergence of the estimates of $\theta$ (thanks to Theorem 3.2 or Proposition 3.2 for instance) leads to a first rate of convergence of $A_n$, which is often verified in practice (see Boyer & Godichon-Baggioni (2020) for instance, see also Lemma 7.9 in the specific case of the linear regression).

**Theorem 3.3.** *Suppose Assumptions **(A1')**, **(A1)** to **(A5)**, and **(H1)** to **(H3)** hold with $\gamma > 1/2$, $\delta > 0$ and $\beta < \gamma - 1/2$. Then,*

$$\mathbb{E}\left[\|\theta_n - \theta\|^2\right] \leq e^{-\frac{1}{2}c_\gamma n^{1-\gamma}}\left(K_1^{(3)} + K_{1'}^{(3)}\max_{0\leq k\leq n}(k+1)^\gamma d_k\right)$$
$$+ n^{-\gamma}\left(2^{3+\gamma}c_\gamma\, Tr\left(H^{-1}\Sigma H^{-1}\right) + \frac{K_2^{(3)}}{n^\gamma} + K_{2'}^{(3)}v_{A,n/2}\right) + d_{\lfloor n/2\rfloor}.$$

*where $\Sigma$ is the covariance matrix of $X$, $K_i^{(3)}$, $i = 1, 1', 2, 2'$ are defined in equation 28, equation 29 and equation 30, and $d_k$ only depending on $M_k$ and $v_{A,k}$ is given in equation 29.*

Recall that $M_k$ is given by Proposition 3.2. Remark from equation 29 that $d_k \leq C(v_{A,k} + M_k)$ for some constant $C > 0$. The latter results can be further simplified if we also assume a sufficiently large exponent $\delta$ in **(H1a)**.

**Corollary 3.1.** *Suppose Assumptions **(A1')**, **(A1)** to **(A5)**, and **(H1)** to **(H3)** hold with $\gamma > 1/2$, $\delta > \frac{2\gamma p}{p-2}$ and $\beta < \gamma - 1/2$. Then,*

$$\mathbb{E}\left[\|\theta_n - \theta\|^2\right] \leq n^{-\gamma}\left(2^{3+\gamma}c_\gamma\, Tr\left(H^{-1}\Sigma H^{-1}\right) + \frac{K_2^{(3')}}{n^\gamma} + K_{2'}^{(3')}v_{A,n/2} + K_{2''}^{(3')}\sqrt{v_{A,n/2}}\right)$$
$$+ K_1^{(3')}e^{-\frac{1}{2}c_\gamma n^{1-\gamma}},$$

*with $K_i^{(3')}$, $i = 1...2''$ given in equation 31 and equation 32.*

Then, if $v_{A,n}$ converges to 0, we obtain the usual rate of convergence $\frac{1}{n^\gamma}$. Indeed, under analogous assumptions, a bound for the online stochastic gradient algorithm would take the form (Bach & Moulines, 2013):

$$\mathbb{E}\left[\|\theta_n - \theta\|^2\right] \leq K_0 e^{-\frac{1}{4}c_\gamma\mu n^{1-\gamma}} + \frac{4c_\gamma C_1^{(1)}}{\mu}n^{-\gamma}.$$

However, we can observe two interesting differences compared to standard gradient algorithms: (i) the smallest eigenvalue no longer influences the exponential decay of the first term, and (ii) the variance term is modified. Observation (i) is one of the main advantages motivating the use of a stochastic Newton algorithm.

### 3.2.4 Convergence results for adaptive gradient (Adagrad)

Recall that the Adagrad algorithm amounts to specify $d$ initial parameters $a_1, \ldots, a_d \in \mathbb{R}_+$ and choose $\overline{A_n}$ diagonal with

$$(\overline{A_n})_{kk'} = \delta_{kk'}\frac{1}{\sqrt{\frac{1}{n+1}\left(a_k + \sum_{i=0}^{n-1}n\left(\nabla_h g(X_{i+1}, \theta_i)_k\right)^2\right)}}. \tag{2}$$

The original Adagrad algorithm would then amount to take $\gamma = 1/2$. To guarantee non-degeneracy of the matrices $(\overline{A_n})_{n\geq 0}$, we assume some minimal fluctuation of the gradient at the minimizer $\theta$.

**(A6)** There is $\alpha > 0$ such that for all $1 \le i \le d$,

$$\mathbb{E}\left[(\nabla_h g(X, \theta))_i^2\right] > \alpha. \tag{3}$$

**(A6')** There is $\alpha > 0$ such that for all $h \in \mathbb{R}^d$ and $1 \le i \le d$,

$$\mathbb{E}\left[(\nabla_h g(X, h))_i^2\right] > \alpha. \tag{4}$$

Remark that **(A6')** is much stronger as **(A6)**. However, the former is often satisfied, as it is the case for the linear regression with noise. Then, we consider the following modification of $\overline{A_n}$: $(A_n)_{kk'} = (\overline{A_n})_{kk'}$ for $k \ne k'$ and

$$(A_n)_{kk} = \begin{cases} \min\left\{c_\beta n^\beta, (\overline{A_n})_{kk}\right\}, & \text{if } \gamma > 1/2 \\ \max\left\{\min\left\{c_\beta n^\beta, (\overline{A_n})_{kk}\right\}, \lambda_0' n^{-\lambda'}\right\}, & \text{if } \gamma \le 1/2 \end{cases} \tag{5}$$

for $1 \le k \le d$, where $\beta_n = c_\beta n^\beta$ with $\beta < \min\{\gamma/2, 1/4\}$ and $\lambda' < \gamma$ (where $\lambda_0'$ and $c_\beta > 0$ are chosen arbitrarily).

We then have the following convergence result for the mean quadratic distance. We only state the result for $\gamma \le 1/2$, but a similar statement holds for $1/2 \le \gamma < 1$ with different constants.

**Theorem 3.4.** *Suppose Assumptions **(A1')**, **(A1)** to **(A4)** and **(A6)** are satisfied for $\gamma \le 1/2$ and $\beta < \min\left(\frac{(1-\gamma)\gamma^2 p}{8-4\gamma+2p\gamma(1-\gamma)}, 1/4\right)$. Then, with $(A_n)_{n\ge 1}$ given in equation 5,*

$$\mathbb{E}\left[\|\theta_n - \theta\|^2\right] \le \tilde{K}_1^{(4)} \exp\left(-c_\gamma \mu \tilde{\lambda}_0 n^{1-\gamma}(1 - \tilde{\varepsilon}(n))\right) + \tilde{K}_2^{(4)} \log(n+1)^{\frac{p-1}{p}} n^{-\frac{(p-1)}{p} \min\left\{\frac{2(1-\gamma)\gamma(\gamma-2\beta)p}{2-\gamma}, 1\right\}}$$
$$+ \tilde{K}_3^{(4)} n^{-\gamma},$$

*with $\tilde{\varepsilon}(n)$ given in equation 34, $v_n = v_0 \log(n+1)$, with $v_0$, $C_S^4$ and $\tilde{\lambda}_0$ given in equation 85, equation 86 and equation 84 with $p' = \frac{2(1-\gamma)}{2-\gamma}p$. In addition, $K_1^{(4)}$, $K_2^{(4)}$ and $K_3^{(4)}$ are given in equation 35. If **(A6')** is satisfied, the same conclusion holds for $\beta < 1/4$ with $C_S$ given in equation 87 taking $p' = \frac{2(1-\gamma)}{2-\gamma}p$.*

In the special case where $\gamma = 1/2$, which corresponds to the usual Adagrad algorithm, we get

$$\mathbb{E}\left[\|\theta_n - \theta\|^2\right] \le K_1^{(4)} \exp\left(-c_\gamma \mu \lambda_0 \sqrt{n}(1 - \varepsilon(n))\right)$$
$$+ \frac{1}{\sqrt{n}}\left(K_2^{(4)} \log(n+1) n^{1/2 - \frac{(1-4\beta)(p-1)}{6}} + K_3^{(4)}\right),$$

and we so achieve the usual rate of convergence $\frac{1}{\sqrt{n}}$ as soon as $1/2 - \frac{(1-4\beta)(p-1)}{6} < 0$, i.e as soon as $p > 4\frac{1-\beta}{1-4\beta}$.

Remark that the advantage of using Adagrad algorithm compared to a standard stochastic gradient algorithm does not appear in the bounds of Theorem 3.4. Since Adagrad algorithm amounts to a regularization of the gradient descent by a diagonal matrix, not much can be deduced in full generality. However, one expects better bounds to hold in the case where the Hessian matrix at the minimizer is also diagonal. For example in practice, the parameter $\beta$ should be tuned in such a way that $c_\beta n^\beta >> \lambda_{\min}(H)^{-1}$ at the time $n$ of interest, where $\lambda_{\min}(H)$ is the smallest eigenvalue of the Hessian at the minimizer : the influence of such a choice of the parameter $\beta$ would appear in the first term of the bound of Theorem 3.4 in the case of a diagonal Hessian at the minimizer.

# 4 Application to linear model

Let us now consider the linear model $Y = X^T\theta + \epsilon$ where $X \in \mathbb{R}^d$ and $\epsilon$ is a centered random real variable independent from $X$. We suppose from now on that $\mathbb{E}\left[XX^T\right]$ is positive. Then, $\theta$ is the unique minimizer of the functional $G : \mathbb{R}^d \longrightarrow \mathbb{R}$ defined for all $h \in \mathbb{R}^d$ by

$$G(h) = \frac{1}{2}\mathbb{E}\left[\left(Y - X^T h\right)^2\right].$$

If $X$ admits a second order moment, the function $G$ is twice continuously differentiable with $\nabla G(h) = -\mathbb{E}\left[(Y - X^T h) X\right]$ and $\nabla^2 G(h) = \mathbb{E}\left[X X^T\right]$.

## 4.1 Stochastic Newton algorithm

The stochastic Newton algorithm is defined recursively for all $n \geq 0$ by (Boyer & Godichon-Baggioni, 2020)

$$\theta_{n+1} = \theta_n + \gamma_{n+1} \tilde{S}_n^{-1} \left(Y_{n+1} - X_{n+1}^T \theta_n\right) X_{n+1} \tag{6}$$

where $\tilde{S}_n = \frac{\alpha_n}{n+m}\left(m S_0 + \sum_{i=1}^n X_i X_i^T\right)$ with $S_0$ positive, $m \geq 1$ and $(\alpha_n)_{\geq 0}$ a deterministic modulating sequence satisfying, for some $0 < \alpha_- < \alpha_+$ and $\alpha > 0$,

$$\alpha_- \leq \alpha_n \leq \alpha_+, \quad \text{and} \quad |\alpha_n - 1| \leq \frac{\alpha}{n}, \quad n \geq 0.$$

The parameter $m \geq 1$ reflects the expected quality of the initial approximation of the Hessian at the minimizer by $S_0$. The usual stochastic Newton algorithm corresponds to the choice $m = 1$, $\alpha_n = 1$. Build then a regularized version by setting

$$\overline{S}_n^{-1} = \frac{\min\left(\left\|\tilde{S}_n^{-1}\right\|_{op}, \beta_{n+1}\right)}{\left\|\tilde{S}_n^{-1}\right\|_{op}} \tilde{S}_n^{-1}$$

with $\beta_n = c_\beta n^\beta$. Remark that $\tilde{S}_{n+1}^{-1}$ can be easily updated with only $O\left(d^2\right)$ operations using Sherman Morrison (or Ricatti's) formula. More precisely, considering $S_n = (n+1)\tilde{S}_n$, one has

$$S_{n+1}^{-1} = S_n^{-1} - \left(1 + X_{n+1}^T S_n^{-1} X_n\right)^{-1} S_n^{-1} X_{n+1} X_{n+1}^T S_n^{-1}.$$

Then, one can easily update $\tilde{S}_n$ and $\overline{S}_n$. We call regularized stochastic Newton algorithm the algorithm equation 6 with $\tilde{S}_n$ replaced by $\bar{S}_n$.

In order to avoid singularities in the estimation of the Hessian, we will assume in the sequel that the distribution of $X$ is non-degenerate on $\mathbb{R}^d$. Formally, this amounts to suppose the existence of a constant $L_{MK} > 0$ such that for any $h \in \mathbb{S}^{d-1}$, $\sqrt{\mathbb{E}\left[h X X^T h\right]} \leq L_{MK} \mathbb{E}\left[\left|X^T h\right|\right]$. We can now rewrite Theorem 3.3 for the regularized algorithm as follows:

**Theorem 4.1.** *[Regularized Stochastic Newton] Suppose that there is $p > 2$ such that $X$ and $\epsilon$ respectively admit a moment of order $4p$ and $2p$. Suppose also that there is a positive constant $L_{MK}$ such that for any $h \in \mathbb{S}^{d-1}$, $\sqrt{\mathbb{E}\left[h X X^T h\right]} \leq L_{MK} \mathbb{E}\left[\left|X^T h\right|\right]$. Then, for any $1/2 < \gamma < 1$, the regularized algorithm $(\theta_n)_{n \geq 0}$ satisfies the mean quadratic error*

$$\mathbb{E}\left[\|\theta_n - \theta\|^2\right] \leq e^{-\frac{1}{2} c_\gamma n^{1-\gamma}} \left(K_{1,lin}^{(3)} + K_{1',lin}^{(3)} \max_{0 \leq k \leq n} d_k (k+1)^\gamma\right)$$

$$+ n^{-\gamma} \left(2^{3+\gamma} c_\gamma \mathbb{E}\left[\epsilon^2\right] Tr\left(H^{-1}\right) + \frac{K_{2,lin}^{(3)}}{n^\gamma} + K_{2',lin}^{(3)} v_{H,n/2} + n^\gamma d_{\lfloor n/2 \rfloor}\right),$$

*where $K_{2,lin}, K_{2',lin}^{(3)}, K_{1,lin}^{(3)}, K_{1',lin}^{(3)}, d_n$ are given by equation 44 while $v_{H,n}$ is defined in equation 43.*

Observe that $n^\gamma d_n = O\left(\frac{1}{n^{\max\left\{\frac{p-2}{2}-\gamma, \gamma\right\}}}\right)$ and $v_{H,n} = O\left(n^{-1}\right)$, and since $p > 4$, these terms are both negligible. Using this theorem, it is possible to prove a non-asymptotic quadratic concentration bound for the convergence of the original stochastic Newton algorithm, at the cost of imposing a sub-gaussian decay on the tail of $X$. Namely, following (Vershynin, 2018, Sec. 2.5), we say that $X$ is sub-gaussian if there exists $c > 0$ such that

$$\mathbb{P}\left[|X| > t\right] \leq 2\exp(-t^2/c)$$

for all $t > 0$, and we then define the sub-gaussian norm $\|X\|_{\psi_2}$ of $X$ as

$$\|X\|_{\psi_2} = \inf\left\{t > 0, \mathbb{E}\left[\exp(\|X\|^2/t^2)\right] \leq 2\right\}.$$

Remark that any gaussian distributed or bounded random variable is sub-gaussian. Under a sub-gaussian hypothesis for $X$, we then have the following concentration bound.

**Corollary 4.1** (Original stochastic Newton)**.** *Suppose that $X$ is sub-gaussian with sub-gaussian norm $\|X\|_{\psi_2} > 0$ and $\epsilon$ admits moments of order $2p$, $p > 2$. Suppose also that there is a positive constant $L_{MK}$ such that for any $h \in \mathbb{S}^{d-1}$, $\sqrt{\mathbb{E}\left[hXX^Th\right]} \leq L_{MK}\mathbb{E}\left[\left|X^Th\right|\right]$. Then, for any $1/2 < \gamma < 1$, we have for all $\delta > 0$ and $n \geq c_0$, with $c_0$ only depending on $S_0$, $d$, $\gamma$ and the second moment of $X$,*

$$\mathbb{P}\left(\|\theta_n - \theta\| > \delta\right) \leq \frac{1}{\delta^2}\left[n^{-\gamma}2^{4+\gamma}c_\gamma\mathbb{E}\left[\epsilon^2\right]Tr\left(H^{-1}\right) + C_1\left(e^{-\frac{1}{2}cn^{1-\gamma}}\mathbb{E}[V_0^p]^{2/p} + \frac{1}{n}\right)\right] + C_2 n^{-2\gamma},$$

*with $c, C_1, C_2$ depending on the parameters of the algorithm, $L_{MK}$, the first $4p$ moments of $X$ and $\|X\|_{\psi_2}$, and the first $2p$ moments of $\epsilon$.*

In view of the central limit theorem proven in Boyer & Godichon-Baggioni (2020), this non-asymptotic bound is optimal in the fluctuation regime up to the numerical constant $2^{4+\gamma}$ and the error terms.

### 4.2 Adagrad algorithm

For linear model, we define Adagrad algorithm for all $n \geq 0$ by

$$\theta_{n+1} = \theta_n + \gamma_{n+1}\bar{D}_n^{-1}\left(Y_{n+1} - X_{n+1}^T\theta_n\right)X_{n+1},$$

with $\bar{D}_n$ diagonal with, for $\gamma \leq 1/2$,

$$(\bar{D}_n)_{kk} = \min\left\{\max\left\{\frac{n^{-\beta}}{c_\beta}, \sqrt{\frac{1}{n+1}\left(a_k + \sum_{i=0}^{n-1}\left(\left(Y_{i+1} - X_{i+1}^T\theta_i\right)(X_{i+1})_k\right)^2\right)}\right\}, \frac{n^{\lambda'}}{\lambda_0'}\right\}.$$

where $0 < \beta < (\gamma - \lambda')/2$ for some $a_k > 0$ and if $\gamma > 1/2$,

$$(\bar{D}_n)_{kk} = \max\left\{\frac{n^{-\beta}}{c_\beta}, \sqrt{\frac{1}{n+1}\left(a_k + \sum_{i=0}^{n-1}\left(\left(Y_{i+1} - X_{i+1}^T\theta_i\right)(X_{i+1})_k\right)^2\right)}\right\},$$

for some $0 < \beta < \gamma - 1/2$. The usual Adagrad algorithm is done with $\gamma = 1/2$, which yields for us

$$(\theta_{n+1})_k = (\theta_n)_k + \frac{\left(Y_{n+1} - X_{n+1}^T\theta_n\right)(X_{n+1})_k}{\min\left\{\max\left\{\frac{n^{-\beta+1/2}}{c_\beta}, \sqrt{a_k + \sum_{i=0}^{n-1}\left(\left(Y_{i+1} - X_{i+1}^T\theta_i\right)(X_{i+1})_k\right)^2}\right\}, \frac{n^{\lambda'+1/2}}{\lambda_0'}\right\}}.$$

Note that a first convergence analysis yields that almost surely there exists $n_0 \geq 0$ such that for $n \geq n_0$,

$$(\theta_{n+1})_k = (\theta_n)_k + \frac{\left(Y_{n+1} - X_{n+1}^T\theta_n\right)(X_{n+1})_k}{\sqrt{a_k + \sum_{i=0}^{n-1}\left(\left(Y_{i+1} - X_{i+1}^T\theta_i\right)(X_{i+1})_k\right)^2}},$$

which is the usual Adagrad algorithm. We can then rewrite Theorem 3.4 as follows (remark that we only state the result for $\gamma \leq 1/2$, but a similar statement holds for $1/2 \leq \gamma < 1$ with different constants).

**Theorem 4.2.** *Suppose that there is $p > 2$ such that $X$ and $\epsilon$ admit a moment of order $2p$. Then, for $\gamma \leq 1/2$ and $\beta < 1/4$, we have*

$$\mathbb{E}\left[\|\theta_n - \theta\|^2\right] \leq K_{1,lin}^{ada}\exp\left(-c_\gamma\lambda_{\min}\lambda_{0,lin}^{ada}n^{1-\gamma}\left(1 - \varepsilon_{n,lin}^{ada}\right)\right)$$
$$+ K_{2,lin}^{ada}\log(n+1)^{\frac{p-1}{p}}n^{-\frac{(p-1)}{p}\min\left\{\frac{2(1-\gamma)\gamma(\gamma-2\beta)p}{2-\gamma}, 1\right\}} + K_{3,lin}^{ada}n^{-\gamma},$$

*where $\varepsilon_{n,lin}^{ada} = o(1)$ is given in equation 51 and $K_{1,lin}^{ada}$, $K_{2,lin}^{ada}$ $K_{3,lin}^{ada}$ are given by equation 52, equation 53 and equation 54.*

Observe that in the case where $\gamma = 1/2$, the $\frac{1}{\sqrt{n}}$ rate of convergence is achieved as soon as $(p-1)(1-4\beta)/3 \geq 1/2$, i.e as soon as $p > \frac{5-4\beta}{2(1-4\beta)}$.

# 5 Application to generalized linear models

The framework of the linear regression can be easily generalized to the more general setting of finite dimensional linear models. Let $\ell : \mathcal{Y} \times \mathcal{Y} \to \mathbb{R}$ be a cost function on some domain $\mathcal{Y} \subset \mathbb{R}$. The general learning problem is to solve the minimization problem

$$\underset{f \in \mathcal{F}}{\arg\min} \, \mathbb{E} \left[ \ell(Y, f(X)) \right],$$

with $(X, Y) \sim \mathbb{P}$ and $\mathcal{F}$ is a given class of measurable functions from $\mathcal{X}$ to $\mathcal{Y}$, where $\mathcal{X}$ is a measurable space. In the case of finite dimensional linear models, $\mathcal{Y} = \mathbb{R}$ and $\mathcal{F} = \left\{ h^T \Phi(\cdot), h \in \mathbb{R}^m \right\}$, with $\Phi : \mathcal{X} \to \mathbb{R}^m$ a *known* design function (remark that the setting can be easily generalized to the case $\mathcal{Y} = \mathbb{R}^p$ and $\Phi : \mathcal{X} \to \mathbb{R}^m$ and $h \in M_{m,p}(\mathbb{R})$). Then, assuming that $\ell$ is convex and adding a regularization term on $\theta$, the minimization problem turns into the framework of this paper with

$$G(h) = \mathbb{E} \left[ g \left( \tilde{Z}, h \right) \right],$$

with $\tilde{Z} = (Y, \Phi(X)) := (Y, \tilde{X})$ and for all $h \in M_{m,p}(\mathbb{R})$, $g(\tilde{Z}, h) = \ell(Y, h^T \tilde{X})$. In what follows, let us suppose from now that the cost function $\ell$ is twice differentiable for the second variable and that there is a positive constant $L_{\nabla l}$ such that for all $h \in \mathbb{R}^d$

$$\left| \nabla_h^2 \ell \left( Y, h^T \tilde{X} \right) \right| \leq L_{\nabla l}, \tag{7}$$

where $\nabla_h^2 \ell(.,.)$ is the second order derivative with respect to the second variable. Remark that such a bound is generally assumed if we require that $\| \nabla^2 G(h) \|_{op} \leq L_{\nabla G} < +\infty$ for all $h \in \mathbb{R}^d$. This is for example satisfied when $\ell(y, y') = f(y - y')$ with $\sup_y |f''(y)| < +\infty$. For example, in the simplest case of the logistic regression, we consider a couple of random variables $(X, Y)$ lying in $\mathbb{R}^d \times \{-1, 1\}$, $\Phi = I_d$ and $\ell(y, y') = \log(1 + \exp(-yy'))$, and we indeed have for all $h$ and $Y \in \{-1, 1\}$

$$\nabla_h^2 \ell(Y, h^T X) = \frac{1}{1 + \exp(h^T X)} \cdot \frac{1}{1 + \exp(-h^T X)} \leq 1.$$

There are then two main cases to deal with the convexity of the minimization problem : either assume strong convexity or use a regularization. The first consists in assuming that the functional $h \longmapsto \mathbb{E} \left[ \ell \left( Y, h^T X \right) \right]$ is strongly convex, which is in particular verified when there exists $\alpha > 0$ such that

$$\inf_{y' \in \mathbb{R}} \nabla_h^2 \ell(y, y') > \alpha. \tag{8}$$

and $\mathbb{E} \left[ X X^T \right]$ is positive. This case is called the elliptic case in the sequel and the results are very analogous to the ones for the linear regression and are thus not repeated. We will then focus on the regularized case. Without uniform lower bound on $\nabla_h^2 \ell(y, y')$, one needs a regularization term, yielding the following regularized minimization problem

$$\underset{\theta \in \mathbb{R}^m}{\arg\min} \, \mathbb{E} \left[ \ell(Y, \langle \theta, \theta^T X \rangle) \right] + \frac{\sigma}{2} \|\theta\|^2 \tag{9}$$

for some $\sigma > 0$. In what follows, we suppose that the minimizer exists and we denote it by $\theta_\sigma$.

## 5.1 Stochastic Newton algorithm

The stochastic Newton algorithm is defined recursively for all $n \geq 0$ by

$$\theta_{n+1} = \theta_n - \gamma_{n+1} \overline{S}_n^{-1} \left( \nabla_h l \left( Y_{n+1}, \theta_n^T X_{n+1} \right) X_{n+1} + \sigma \theta_n \right),$$

where, using the trick introduced in Bercu et al. (2021) and developed in Godichon-Baggioni et al. (2022), $\overline{S}_n$ is the natural recursive estimate of the Hessian given by

$$\overline{S}_n = \frac{1}{n+1} \sum_{i=0}^{n-1} \nabla_h^2 \ell(Y_{i+1}, \langle \theta_i, X_{i+1} \rangle) X_{i+1} X_{i+1}^T + \frac{\sigma d}{n+1} \sum_{i=1}^{n} e_{i[d]+1} e_{i[d]+1}^T, \tag{10}$$

with $i[d]$ denoting $i$ modulo $d$. Remark that one can easily update the inverse using the Riccati's formula used twice, i.e considering $S_n = (n+1)\overline{S}_n$ and

$$S_{n+\frac{1}{2}}^{-1} = S_n^{-1} - \nabla_h^2 \ell(Y_{n+1}, \langle \theta_n, X_{n+1} \rangle) \left(1 + \nabla_h^2 \ell(Y_{n+1}, \langle \theta_n, X_{n+1} \rangle) X_{n+1}^T S_n^{-1} X_{n+1}\right)^{-1} S_n^{-1} X_{n+1} X_{n+1}^T S_n^{-1}$$

$$S_{n+1} = S_{n+\frac{1}{2}}^{-1} - \sigma d \left(1 + \sigma d e_{(n+1)[d]+1}^T S_{n+\frac{1}{2}}^{-1} e_{(n+1)[d]+1}\right)^{-1} S_{n+\frac{1}{2}}^{-1} e_{(n+1)[d]+1} e_{(n+1)[d]+1}^T S_{n+\frac{1}{2}}^{-1},$$

one has $\overline{S}_{n+1}^{-1} = (n+2)S_{n+1}^{-1}$. In what follows, let us suppose that the following assumptions hold:

**(GLM1)** There is $L_{\nabla^2 L} \geq 0$ such that the function $h \longmapsto \mathbb{E}\left[\nabla_h^2 \ell\left(Y, h^T X\right) X X^T\right]$ is $L_{\nabla^2 L}$-Lispchitz with respect to the spectral norm.

**(GLM2)** There is $p > 2$ such that $X$ admits a moment of order $2p$ and such that there is a positive constant $L_\sigma$ satisfying for all $0 \leq a \leq 2p$

$$\mathbb{E}\left[\left\|\nabla_h \ell\left(Y, X^T \theta_\sigma\right) X + \sigma \theta_\sigma\right\|^a\right] \leq L_\sigma^a.$$

Remark that Assumption **(GLM1)** is verified when for all $y$, $\nabla_h^2 \ell(y,.)$ is Lipschitz and $X$ admits a third order moment, which can be easily verified for the logistic regression for instance. Assumption **(GLM2)** is verified when the random variable $\nabla_h \ell\left(Y, X^T \theta_\sigma\right) X$ admits a moment of order $2p$.

**Theorem 5.1.** *Suppose Assumptions **(GLM1)** and **(GLM2)** hold. Then,*

$$\mathbb{E}\left[\|\theta_n - \theta_\sigma\|^2\right] \leq e^{-\frac{1}{2}c_\gamma n^{1-\gamma}} \left(K_{1,GLM}^{(3)} + K_{1',GLM}^{(3)} \max_{0 \leq k \leq n}(k+1)^\gamma d_{k,GLM}\right)$$

$$+ n^{-\gamma}\left(2^{3+\gamma} c_\gamma Tr\left(H_\sigma^{-1} \Sigma_\sigma H_\sigma^{-1}\right) + \frac{K_{2,GLM}^{(3)}}{n^\gamma} + K_{2',GLM}^{(3)} v_{l,n/2} + n^\gamma d_{\lfloor n/2 \rfloor, GLM}\right),$$

*where $H_\sigma = \mathbb{E}\left[\nabla_h^2 \ell\left(Y, X^T \theta_\sigma\right) X X^T\right] + \sigma I_d$, $\Sigma_\sigma = \mathbb{E}\left[\left(\nabla_h \ell\left(Y, X^T \theta_\sigma\right) X + \sigma \theta_\sigma\right)\left(\nabla_h \ell\left(Y, X^T \theta_\sigma\right) X + \sigma \theta_\sigma\right)^T\right]$, $K_{1,GLM}^{(3)}, K_{1',GLM}^{(3)}, K_{2,GLM}^{(3)}, K_{2',GLM}^{(3)}, d_{n,GLM}$ are defined in equations equation 60, equation 61 and equation 62, and $v_{l,n}$ is defined in Proposition 7.5.*

Remark that $n^\gamma d_{\lfloor n/2 \rfloor, \text{GLM}} = O(n^{-\min(\gamma, 1-\gamma)})$, see Section 5.

### 5.2 Adagrad algorithm

For generalized linear model, Adagrad algorithm is defined for all $n \geq 0$ by

$$\theta_{n+1} = \theta_n - \gamma_{n+1} \bar{D}_n^{-1} \nabla_h \ell\left(Y_{n+1}, \theta_n^T X_{n+1}\right) X_{n+1},$$

where $\bar{D}_n$ is diagonal and for $\gamma > 1/2$,

$$(\bar{D}_n)_{kk} = \max\left\{\frac{n^{-\beta}}{c_\beta}, \sqrt{\frac{1}{n+1}\left(a_k + \sum_{i=0}^{n-1}\left(\nabla_h \ell\left(Y_{i+1}, \theta_i^T X_{i+1}\right)(X_{i+1})_k + \sigma(\theta_i)_k\right)^2\right)}\right\},$$

for some $0 < \beta < \gamma - 1/2$, and for $\gamma \leq 1/2$,

$$(\bar{D}_n)_{kk} = \min\left\{\max\left\{\frac{n^{-\beta}}{c_\beta}, \sqrt{\frac{1}{n+1}\left(a_k + \sum_{i=0}^{n-1}\left(\nabla_h \ell\left(Y_{i+1}, \theta_i^T X_{i+1}\right)(X_{i+1})_k + \sigma(\theta_i)_k\right)^2\right)}\right\}, \frac{n^{\lambda'}}{\lambda_0'}\right\}.$$

where $0 < \beta < (\gamma - \lambda')/2$ and $a_k > 0$. The usual Adagrad algorithm is done with $\gamma = 1/2$, which yields for us

$$(\theta_{n+1})_k = (\theta_n)_k + \frac{\nabla_h \ell\left(Y_{n+1}, \theta_n^T X_{n+1}\right)(X_{n+1})_k + \sigma(\theta_n)_k}{\min\left\{\max\left\{\frac{n^{-\beta+1/2}}{c_\beta}, \sqrt{a_k + \sum_{i=0}^{n-1}\left(\nabla_h \ell\left(Y_{i+1}, \theta_i^T X_{i+1}\right)(X_{i+1})_k + \sigma(\theta_i)_k\right)^2}\right\}, \frac{n^{\lambda'+1/2}}{\lambda_0'}\right\}}.$$

Like the linear regression, the general linear model needs minimal randomness to ensure the expected rate of convergence of Adagrad. Indeed, in the extreme case of a deterministic sequence $(X_n, Y_n)_{n \geq 0}$, Adagrad algorithm may diverge in the unfortunate situation where $\nabla_h \ell \left( Y_{i+1}, \theta_i^T X_{i+1} \right) (X_{i+1})_k$ vanishes or remains very small. Such behavior can be averted by requiring at the minimizer $\theta_\sigma$ a minimal variance for $\nabla_h \ell \left( Y, \theta_\sigma^T X \right) (X)_k$ for all $1 \leq k \leq d$.

**(GLM3)** There is a positive constant $\alpha_\sigma > 0$ such that for all $1 \leq k \leq d$

$$Var \left[ \nabla_h \ell \left( Y, X^T \theta_\sigma \right) (X)_k \right] > \alpha_\sigma.$$

Remark that

$$Var \left[ \nabla_h \ell \left( Y, X^T \theta_\sigma \right) (X)_k \right] = \mathbb{E} \left[ \left| \nabla_h \ell \left( Y, X^T \theta_\sigma \right) (X)_k + \sigma(\theta_\sigma)_k \right|^2 \right], \tag{11}$$

so that **(GLM3)** can be seen as a mirror assumption to **(GLM2)**. We should stress that the existence of such $\alpha_\sigma$ is almost automatic when a minimal randomness between $X$ and $Y$ is assumed. Indeed, having $\nabla_h l \left( Y, \theta_\sigma^T X \right) X_k$ deterministic would imply an analytic relation between $Y$ and $X$. The main computational issue is to estimate a concrete value of $\alpha_\sigma$. An example dealing with the logistic regression is given in Section E.

When **(GLM3)** is assumed, one can show using Theorem 5.2 that there exists almost surely $n_0 \geq n$ such that for $n \geq n_0$,

$$(\theta_{n+1})_k = (\theta_n)_k + \frac{\nabla_h \ell \left( Y_{n+1}, \theta_n^T X_{n+1} \right) (X_{n+1})_k + \sigma(\theta_n)_k}{\sqrt{a_k + \sum_{i=0}^{n-1} \left( \nabla_h \ell \left( Y_{i+1}, \theta_i^T X_{i+1} \right) (X_{i+1})_k + \sigma(\theta_i)_k \right)^2}},$$

so that we recover the usual Adagrad algorithm for large $n$. We can then rewrite Theorem 3.4 as follows (remark that we only state the result for $\gamma \leq 1/2$, but a similar statement holds for $1/2 \leq \gamma < 1$ with different constants).

**Theorem 5.2.** *Suppose Assumptions* **(GLM1)**, **(GLM2)** *and* **(GLM3)** *hold. Then, assuming that $\gamma \leq 1/2$ and $\beta < \min \left( \frac{(1-\gamma)\gamma^2 p}{8-4\gamma+2p\gamma(1-\gamma)}, 1/4 \right)$, we have*

$$\mathbb{E} \left[ \| \theta_n - \theta_\sigma \|^2 \right] \leq K_{1,GLM}^{ada} \exp \left( -c_\gamma \sigma \tilde{\lambda}_{0,GLM} n^{1-\gamma} (1 - \varepsilon(n)) \right)$$

$$K_{2,GLM}^{ada} \log(n+1)^{\frac{p-1}{p}} n^{-\frac{(p-1)}{p} \min \left\{ \frac{2(1-\gamma)\gamma(\gamma-2\beta)p}{2-\gamma}, 1 \right\}} + K_{3,GLM}^{ada} n^{-\gamma},$$

*where $\varepsilon(n) = o(1)$, $K_{1,GLM}^{ada}$, $K_{2,GLM}^{ada}$ and $K_{3,GLM}^{ada}$ have explicit formulas depending on the parameters of the model.*

We do not specify the exact value of the constants here, since they can easily be obtained along the lines of previous results. Once again, when $\gamma = 1/2$, the $\frac{1}{\sqrt{n}}$ rate of convergence is achieved as soon as $p > \frac{5-4\beta}{2(1-4\beta)}$.

## 6 Simulation study

In this simulation study, we consider the following scenarios:

**Stochastic Newton Algorithm:** We set $c_\gamma = 1$ and initialize $A_n = \frac{1}{10} I_d$ to stabilize the algorithm during the first iterations. Additionally, and again for stabilization purposes, as suggested in Boyer & Godichon-Baggioni (2020), we use a modified step size, taking $\gamma_n = \frac{c_\gamma}{(n+20)^\gamma}$. We consider:

- The choice of $\gamma$: $\gamma = 0.66$ or $\gamma = 0.75$.

- The use of truncation or not, with $c_\beta = 1$ and $\beta = \gamma - 1/2$, while employing the Frobenius norm.

**Adagrad:** We set $c_\gamma = 1$ and initialize $A_n = I_d$. For stabilization purposes, as suggested in Boyer & Godichon-Baggioni (2020), we use a modified step size, taking $\gamma_n = \frac{c_\gamma}{(n+20)^\gamma}$. We consider:

- The choice of $\gamma$: $\gamma = 0.5$ or $\gamma = 0.75$.

- The use of truncation or not, with $c_\beta = 1$, $\lambda'_0 = 1$, $\beta = 0.25$ (resp. 0.125) and $\lambda' = 0.385$ (resp. 0.25) if $\gamma = 0.75$ (resp. if $\gamma = 0.5$).

### 6.1 Linear model

We consider the linear model:

$$Y = X^T\theta + \epsilon, \quad \text{where} \quad X \sim \mathcal{N}(0, \text{diag}(1, \ldots, d)) \quad \text{and} \quad \epsilon \sim \mathcal{N}(0, 1).$$

The parameter $\theta$ is set as $\theta = (-d/2, -(d-1)/2, \ldots, d/2)$. In the following experiments, we set $d = 10$ and generate 50 datasets of size $n = 10^5$. Moreover, we consider random initializations $\theta_0 = \theta + U$, where $U \sim \mathcal{N}(0, I_d)$. In Figures 1 and 2, we analyze the evolution of the quadratic mean error of the estimates

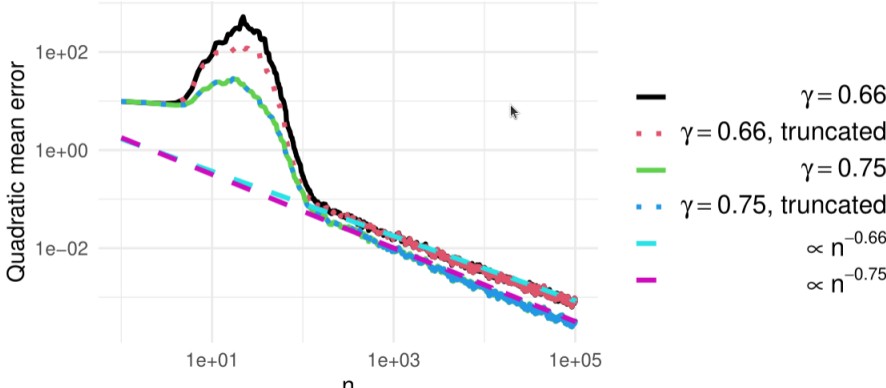

Figure 1: Evolution of the mean squared error of the estimators obtained using the stochastic Newton algorithm in the linear model as a function of the sample size $n$, for $\gamma = 0.66$ or $\gamma = 0.75$, with or without truncation.

as a function of the sample size $n$. Unsurprisingly, we observe that the estimators achieve the expected convergence rates. Although it is possible to construct counterexamples where truncation is essential, we see here that it has little to no impact on the quality of the estimators. This is not surprising, considered Corollary 4.1 obtained for the linear regression : the possible failing of the convergence in quadratic mean would be due to an event of exponentially small probability, and such an event would only appear with reasonable probability on a very large sample of datasets.

### 6.2 Logistic regression

We now consider the logistic regression case:

$$Y|X \sim \mathcal{B}\left(\pi\left(\theta^T X\right)\right), \quad \text{where} \quad X \sim \mathcal{N}(0, \text{diag}(1, \ldots, d)) \quad \text{and} \quad \pi(x) = \frac{e^x}{1 + e^x}.$$

It is well known that $\theta$ is the minimizer of the functional $G : \mathbb{R}^d \longrightarrow \mathbb{R}$ defined for all $h \in \mathbb{R}^d$ by

$$G(h) = \mathbb{E}\left[\log\left(1 + \exp\left(X^T h\right)\right) - Y X^T h\right].$$

However, this function is not strongly convex. To address this, we consider Ridge logistic regression, where we aim to estimate the minimizer of the penalized function $G_\sigma : \mathbb{R}^d \longrightarrow \mathbb{R}$ defined for all $h \in \mathbb{R}^d$ by

$$G_\sigma(h) = \mathbb{E}\left[\log\left(1 + \exp\left(X^T h\right)\right) - Y X^T h\right] + \frac{\sigma}{2}\|h\|^2.$$

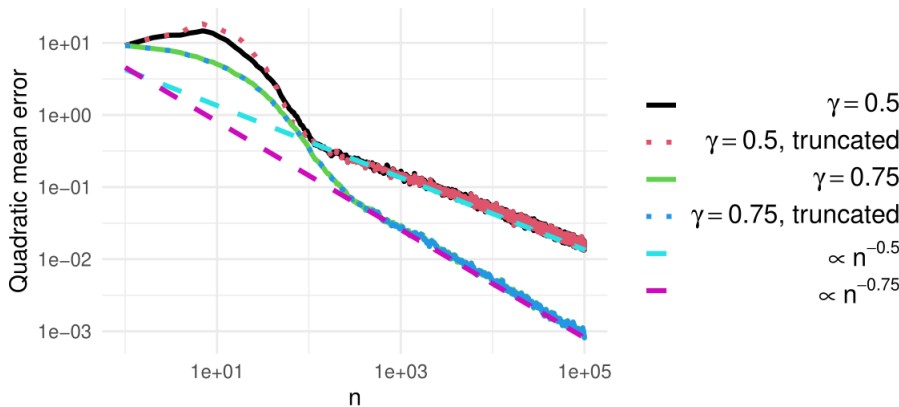

Figure 2: Evolution of the quadratic mean error of the estimates obtained using the Adagrad algorithm in the linear model as a function of the sample size $n$, for $\gamma = 0.5$ or $\gamma = 0.75$, with or without truncation.

In the sequel, the parameter $\theta$ is set as $\theta = (-d/2, -(d-1)/2, \ldots, d/2)$. In addition, we set $\sigma = 0.1$ and denote by $\theta^*$ the minimizer of $G_\sigma$ (in practice, we generate a sample of size $10^7$ and approximate the minimizer using the R function `glmnet`). Moreover, we generate 50 datasets of size $n = 10^5$ and we consider random initializations $\theta_0 = \theta^* + U$, where $U \sim \mathcal{N}(0, I_d)$. In Figures 3 and 4, we analyze the evolution of the

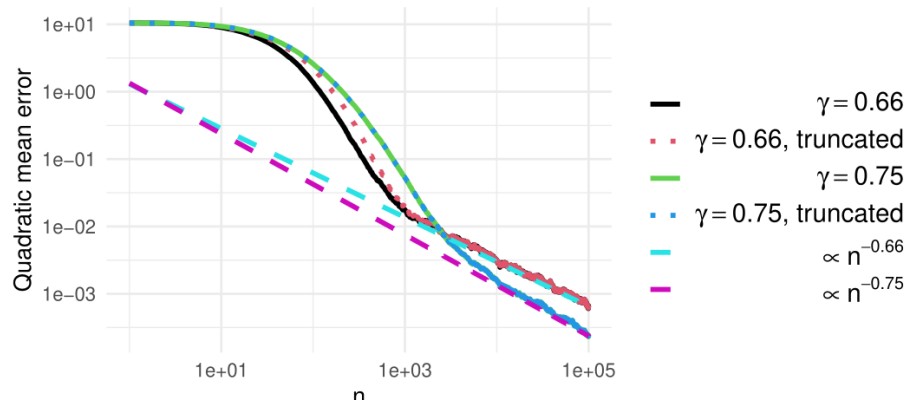

Figure 3: Evolution of the mean squared error of the estimators obtained using the stochastic Newton algorithm in the ridge logistic model as a function of the sample size $n$, for $\gamma = 0.66$ or $\gamma = 0.75$, with or without truncation.

quadratic mean error of the estimates as a function of the sample size $n$. Unsurprisingly, we observe that the estimators achieve the expected convergence rates. Contrary to the linear case, we can see here that there is a little impact of the truncation on the behaviour of the estimates, especially for Adagrad algorithm with $\gamma = 0.5$. Indeed, the truncation seems to have a little impact on the rate of convergence.

## Conclusion

In this paper, we have proposed a relatively simplified framework in which we can establish the convergence rates of adaptive algorithms. More specifically, the relaxation of assumptions primarily concerns the conditions imposed on adaptive step sizes to achieve the desired convergence results.

However, it is now necessary to complement these relaxed assumptions on step sizes with a similar relaxation of assumptions on the studied functions, particularly by providing theoretical guarantees in the non-convex

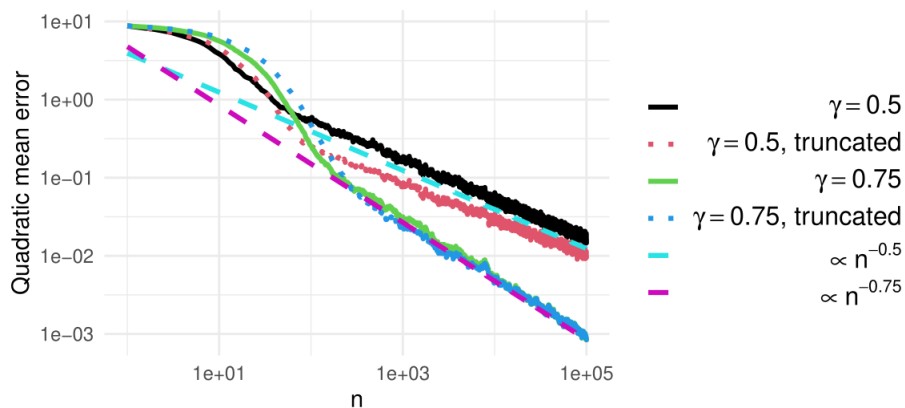

Figure 4: Evolution of the quadratic mean error of the estimates obtained using the Adagrad algorithm in the ridge logistic model as a function of the sample size $n$, for $\gamma = 0.5$ or $\gamma = 0.75$, with or without truncation.

setting. This is especially important since, in most machine learning applications, and particularly in deep learning, the function to be minimized is typically non-convex (typically, for neural networks).

## 7 Proofs

Throughout our proofs, to alleviate notations, we will denote by the same way $\|.\|$ the Euclidean norm of $\mathbb{R}^d$ and the spectral norm for square matrices. Moreover, since we always assume Assumption **(A1)**, we denote by $C_1, C_1', C_2, C_2'$ constants such that for all $h \in \mathbb{R}^d$,

$$\mathbb{E}\left[\|\nabla_h g(X, h)\|^2\right] \le C_1 + C_2 \|h - \theta\|^2, \qquad \mathbb{E}\left[\|\nabla_h g(X, h)\|^4\right] \le C_1' + C_2' \|h - \theta\|^4. \tag{12}$$

### 7.1 Sketch of the proofs and a useful proposition

#### 7.1.1 Proofs of the results of Section 3

The proofs of Theorem 3.1 and Theorem 3.2 rely on the following Taylor's decomposition of the function $G$. Denoting $V_n = G(\theta_n) - G(\theta)$ and $g'_{n+1} = \nabla_h g(X_{n+1}, \theta_n)$,

$$V_{n+1} \le V_n - \gamma_{n+1} \nabla G(\theta_n)^T A_n g'_{n+1} + \frac{L_{\nabla G}}{2} \gamma_{n+1}^2 \|A_n\|^2 \|g'_{n+1}\|^2.$$

Then, taking the conditional expectation, it comes

$$\mathbb{E}[V_{n+1}|\mathcal{F}_n] \le V_n - \gamma_{n+1} \nabla G(\theta_n)^T A_n \nabla G(\theta_n) + \frac{L_{\nabla G}}{2} \gamma_{n+1}^2 \|A_n\|^2 \mathbb{E}\left[\|g'_{n+1}\|^2 |\mathcal{F}_n\right].$$

Furthermore, since Assumption **(A4)** implies that $G(h) - G(\theta) \le \frac{1}{2\mu}\|\nabla G(h)\|^2$ for $h \in \mathbb{R}^d$, and thanks to Assumption **(A1)**, it comes

$$\mathbb{E}[V_{n+1}] \le \left(1 - 2\mu\gamma_{n+1}\lambda_{\min}(A_n) + \frac{C_2 L_{\nabla G}}{\mu}\gamma_{n+1}^2 \|A_n\|^2\right) \mathbb{E}[V_n] + \frac{C_1 L_{\nabla G}}{2}\gamma_{n+1}^2 \|A_n\|^2.$$

Then, the goal of the proofs of both theorems is to control the eigenvalues of $A_n$ in the latter inequality and then to deduce a general bound on $\mathbb{E}[V_n]$ by using Proposition 7.1:

1. the stochastic bound on $\lambda_{\min}(A_n)$ from Assumption **(H1a)** and the deterministic bound on $\|A_n\|$ from Assumption **(H1b)** yield a first bound given in Theorem 3.1.

2. further assuming a bound on the first moments of $\|A_n\|$ (see Assumption **(H2a-H2b)**) yields a refined bound in Theorem 3.2. The exact same pattern of proof also yields a bound on $\mathbb{E}\left[V_n^2\right]$ in Proposition 3.2.

To prove Theorem 3.3, we use the defining recursion relation of $\theta_n$,

$$\theta_{n+1} = \theta_n - \gamma_{n+1} A_n \nabla_h g\left(X_{n+1}, \theta_n\right),$$

and, with the aim of using Assumption **(H3)**, we write $A_n = H^{-1} + \left(A_n - H^{-1}\right)$ and take the square of the latter relation to get

$$\|\theta_{n+1} - \theta\|^2 \le \|\theta_n - \theta\|^2 - 2\gamma_{n+1}\left\langle g'_{n+1}, H^{-1}\left(\theta_n - \theta\right)\right\rangle - 2\gamma_{n+1}\left\langle\left(A_n - H^{-1}\right) g'_{n+1}, \theta_n - \theta\right\rangle$$
$$+ 2\gamma_{n+1}^2\left\|H^{-1} g'_{n+1}\right\|^2 + 2\gamma_{n+1}^2\left\|A_n - H^{-1}\right\|^2 \left\|g'_{n+1}\right\|^2.$$

The last three error terms can be controlled by Assumptions **(A1')** and **(A3)** and by the crude bound on $\mathbb{E}\left[V_n^2\right]$ obtained from Proposition 3.2; remark that the penultimate term is the main contribution to the constant in front of the $n^{-\gamma}$ term in the final bound of Theorem 3.3. The second error term is the one contributing to the exponential term in the final bound. The crucial point is that, after taking conditional expectation on $\mathcal{F}_n$, this second term simplifies into

$$\mathbb{E}\left[2\gamma_{n+1}\left\langle g'_{n+1}, H^{-1}\left(\theta_n - \theta\right)\right\rangle | \mathcal{F}_n\right] = 2\gamma_{n+1}\left\langle \nabla G(\theta_n), H^{-1}\left(\theta_n - \theta\right)\right\rangle$$
$$= 2\gamma_{n+1}\left\langle H^{-1}\left(H(\theta_n - \theta) + o\left(\|\theta_n - \theta\|^2\right)\right), \theta_n - \theta\right\rangle$$
$$= 2\gamma_{n+1}\|\theta_n - \theta\|^2\left(1 + o\left(\|\theta_n - \theta\|^2\right)\right),$$

where $o(\|\theta_n - \theta\|^2)$ is an error term controlled by Assumption **(A5)** and Proposition 3.2.

The proof of Theorem 3.4 consists in checking the hypothesis of Theorem 3.2. The only delicate part is the proof of Assumption **(H2a-H2b)** : in order to get bounds on the moments of the preconditioner $A_n$ in the case of Adagrad, one needs to first use a convergence of $\theta_n$ towards $\theta$ by using Theorem 3.1 and the strong convexity of $G$ (see Lemma 7.4 and Lemma 7.5).

### 7.1.2 Proofs of the results of Section 4 and Section 5

The proof of the results of Section 4 and Section 5 mainly consist in checking all necessary hypotheses to apply Theorem 3.3 and Theorem 3.4. Most computations are straightforward, except when considering the Stochastic Newton algorithm, for which checking Assumption **(H2a-H2b)** in the case of the linear regression and Assumption **(H3)** in the generalized linear case are slightly more delicate.

In the case of the stochastic Newton algorithm for the linear regression, one needs to have a lower bound on $\sigma_{\min}\left(\frac{1}{n+1}(S_0 + \sum_{i=1}^n X_i X_i^T)\right)$ in order to get bounds on the moments of $A_n$ (see Proposition 7.3). Since the bound on the moments has to be independent of $n$, we use a general result from random matrix theory, see Koltchinskii & Mendelson (2015). This only holds for random vectors $X_i$ having a certain non-degeneracy condition which is described by the constant $L_{MK}$ appearing in the statement of Theorem 4.1.

In the case of the generalized linear regression, Assumption **(H2a-H2b)** is directly implied by the regularizing term. However, Assumption **(H3)** is not straightforward, since $A_n$ depends on $(\theta_k)_{1 \le k \le n}$ (which was not the case for the linear regression). In order to prove this assumption, one first applies Theorem 3.2 and the strong convexity to get a first convergence result of $\theta_n$ towards $\theta$. This then allow us to prove Assumption **(H3)** in Proposition 7.7.

### 7.2 A useful inequality

The last step of all the proofs of the main results consists in applying the following technical result from (Godichon-Baggioni et al., 2021, Proposition A.5) to adequate sequences.

**Proposition 7.1.** *Let $(\gamma_t)_{t \ge 1}$, $(\eta_t)_{t \ge 1}$, and $(\nu_t)_{t \ge 1}$ be some positive and decreasing sequences and let $(\delta_t)_{t \ge 0}$, satisfying the following:*

- *The sequence $\delta_t$ follows the recursive relation:*

$$\delta_t \leq (1 - 2\omega\gamma_t + \eta_t\gamma_t)\,\delta_{t-1} + \nu_t\gamma_t, \tag{13}$$

  *with $\delta_0 \geq 0$ and $\omega > 0$.*

- *Let $\gamma_t$ and $\eta_t$ converge to $0$.*

- *Let $t_0 = \inf\{t \geq 1 : \eta_t \leq \omega\}$, and let us suppose that for all $t \geq t_0 + 1$, one has $\omega\gamma_t \leq 1$.*

*Then, for all $t \in \mathbb{N}$, we have the upper bound:*

$$\delta_t \leq \exp\left(-\omega \sum_{j=t/2}^{t} \gamma_j\right) \exp\left(2 \sum_{i=1}^{t} \eta_i\gamma_i\right) \left(\delta_0 + 2 \max_{1 \leq i \leq t} \frac{\nu_i}{\eta_i}\right) + \frac{1}{\omega} \max_{t/2 \leq i \leq t} \nu_i.$$

*with the convention that $\sum_{t_0}^{t/2} = 0$ if $t/2 < t_0$.*

### 7.3 Proof of Theorem 3.1

Remark that thanks to a Taylor's expansion of the gradient, denoting $V_n = G(\theta_n) - G(\theta)$ and $g'_{n+1} = \nabla_h g(X_{n+1}, \theta_n)$,

$$V_{n+1} \leq V_n - \gamma_{n+1} \nabla G(\theta_n)^T A_n g'_{n+1} + \frac{L_{\nabla G}}{2} \gamma_{n+1}^2 \|A_n\|^2 \|g'_{n+1}\|^2$$

$$\leq V_n - \gamma_{n+1} \nabla G(\theta_n)^T A_n g'_{n+1} + \frac{L_{\nabla G}}{2} \gamma_{n+1}^2 \beta_{n+1}^2 \|g'_{n+1}\|^2, \tag{14}$$

where we used Hypothesis **(H1b)** on the last line. Then, taking the conditional expectation, thanks to assumption **(A1)**, and since $\|\theta_n - \theta\|^2 \leq \frac{2}{\mu} V_n$,

$$\mathbb{E}[V_{n+1}|\mathcal{F}_n] \leq \left(1 + \frac{C_2 L_{\nabla G}}{\mu} \beta_{n+1}^2 \gamma_{n+1}^2\right) V_n - \gamma_{n+1} \nabla G(\theta_n)^T A_n \nabla G(\theta_n) + \frac{C_1 L_{\nabla G}}{2} \gamma_{n+1}^2 \beta_{n+1}^2$$

Furthermore, since Assumption **(A4)** implies that $G(h) - G(\theta) \leq \frac{1}{2\mu} \|\nabla G(h)\|^2$ for $h \in \mathbb{R}^d$, it comes

$$\nabla G(\theta_n)^T A_n \nabla G(\theta_n) \geq \lambda_{\min}(A_n) \|\nabla G(\theta_n)\|^2$$

$$\geq 2\lambda_n \mu V_n \mathbf{1}_{\lambda_{\min}(A_n) \geq \lambda_n}$$

$$= 2\lambda_n \mu V_n - \mathbf{1}_{\lambda_{\min}(A_n) < \lambda_n} 2\lambda_n \mu V_n, \tag{15}$$

with $\lambda_n = \lambda_0 (n+1)^\lambda$ with $0 \leq \lambda < 1 - \gamma$. Applying Cauchy-Schwarz yields

$$\mathbb{E}\left[\nabla G(\theta_n)^T A_n \nabla G(\theta_n)\right] \geq 2\lambda_n \mu \mathbb{E}[V_n] - 2\lambda_n \mu \sqrt{\mathbb{E}[V_n^2]} \sqrt{\mathbb{P}[\lambda_{\min}(A_n) < \lambda_n]}$$

$$\geq 2\lambda_n \mu \mathbb{E}[V_n] - 2\lambda_n \mu V \sqrt{\mathbb{P}[\lambda_{\min}(A_n) < \lambda_n]},$$

with $V^2 \geq \sup_{n \geq 0} \mathbb{E}[V_n^2]$ calculated later. Then, Assumption **(H1a)** gives $\mathbb{P}[\lambda_{\min}(A_n) < \lambda_n] \leq v_{n+1}(n+1)^{-\delta - q\lambda} := \bar{v}_n$, so that

$$\mathbb{E}[V_{n+1}] \leq \left(1 - 2\mu\lambda_0(n+1)^{-\lambda}\gamma_{n+1} + \frac{C_2 L_{\nabla G}}{\mu} \beta_{n+1}^2 \gamma_{n+1}^2\right) \mathbb{E}[V_n]$$

$$+ 2\lambda_0(n+1)^{-\lambda} \mu V \gamma_{n+1} \sqrt{\bar{v}_n} + \frac{C_1 L_{\nabla G}}{2} \gamma_{n+1}^2 \beta_{n+1}^2.$$

In order to apply Proposition 7.1, let us denote

$$C_M = \max\left\{\frac{C_2 L_{\nabla G} c_\beta^2 c_\gamma}{\mu}, (\mu\lambda_0)^{\frac{2\gamma-2\beta}{\gamma+\lambda}} c_\gamma^{\frac{\gamma-2\beta-\lambda}{\gamma+\lambda}}\right\}, \tag{16}$$

the last upper bound being added so that the terms of equation 17 below satisfy the third condition of Proposition 7.1. Set $\tilde{\gamma}_n = c_\gamma n^{-(\lambda+\gamma)}$, and remark that

$$\mathbb{E}[V_{n+1}] \leq \left(1 - 2\mu\lambda_0\tilde{\gamma}_{n+1} + C_M(n+1)^{2\beta+\lambda-\gamma}\tilde{\gamma}_{n+1}\right)\mathbb{E}[V_n] + 2\lambda_0\mu V\sqrt{v_n}\tilde{\gamma}_{n+1}$$
$$+ \frac{C_1 L_{\nabla G}}{2}(n+1)^\lambda \gamma_{n+1}\beta_{n+1}^2\tilde{\gamma}_{n+1}. \tag{17}$$

Then, since $2\gamma - 2\beta - 1 \neq 1$, with the help of Proposition 7.1 and an integral test for convergence to get $\sum_{k=1}^n k^{2\beta-2\gamma} \leq 1 + \frac{n^{(1+2\beta-2\gamma)^+}}{|2\gamma-2\beta-1|}$ and $\sum_{t=\lfloor n/2\rfloor}^n t^{-\gamma} \geq \frac{1-2^{\gamma-1}}{1-\gamma}n^{1-\gamma} \geq n^{1-\gamma}$ for $\gamma \in (0,1)$,

$$\mathbb{E}[V_n] \leq \exp\left(-c_\gamma\mu\lambda_0 n^{1-(\lambda+\gamma)}\right)\exp\left(2C_M c_\gamma\left(1 + \frac{n^{(1+2\beta-2\gamma)^+}}{|2\gamma-2\beta-1|}\right)\right) \cdot$$
$$\left(\mathbb{E}[V_0] + 4\frac{\lambda_0\mu V}{C_M}\max_{1\leq k\leq n} k^{\gamma-2\beta-\lambda}\sqrt{\bar{v}_k} + \frac{C_1 L_{\nabla G}c_\gamma c_\beta^2}{C_M}\right) + 2V\sqrt{\bar{v}_{n/2}} + \frac{C_1 L_{\nabla G}}{2^{1+\lambda}\mu\lambda_0}n^\lambda \beta_{n/2}^2\gamma_{n/2}, \tag{18}$$

where we recall that $\bar{v}_n = v_{n+1}(n+1)^{-\delta-q\lambda} \geq \mathbb{P}[\lambda_{\min}(A_n) < \lambda_n]$. Remark that

$$k^{\gamma-2\beta-\lambda}\sqrt{\bar{v}_k} = \sqrt{v_{k+1}}(k+1)^{\gamma-2\beta-\lambda}(k+1)^{-(\delta+q\lambda)/2} = \sqrt{v_{k+1}}(k+1)^{\gamma-2\beta-\delta/2-(q/2+1)\lambda},$$

so that $\max_{0\leq k\leq n}(k+1)^{\gamma-2\beta-\lambda}\sqrt{\bar{v}_k} = \max_{1\leq k\leq n+1} k^{\gamma-2\beta-\delta/2-(q/2+1)\lambda}\sqrt{v_k}$. Hence, we get

$$\mathbb{E}[V_n] \leq \exp\left(-c_\gamma\mu\lambda_0 n^{1-(\lambda+\gamma)}\right)\exp\left(2C_M c_\gamma\left(1 + \frac{n^{(1+2\beta-2\gamma)^+}}{|2\gamma-2\beta-1|}\right)\right)$$
$$\cdot\left(\mathbb{E}[V_0] + 4\frac{\lambda_0\mu V}{C_M}\max_{1\leq k\leq n+1} k^{\gamma-2\beta-\delta/2-(q/2+1)\lambda}\sqrt{v_k} + \frac{C_1 L_{\nabla G}c_\gamma c_\beta^2}{C_M}\right)$$
$$+ 2^{1+(\delta+q\lambda)/2}V\sqrt{v_{\lfloor n/2\rfloor}}n^{-(\delta+q\lambda)/2} + 2^{\gamma-2\beta-\lambda-1}\frac{C_1 L_{\nabla G}c_\gamma c_\beta^2}{\mu\lambda_0}n^{2\beta+\lambda-\gamma}$$

where $V$ is defined in Lemma 7.1. Hence, as long as $\gamma + \lambda + (1+2\beta-2\gamma)^+ < 1$, which is satisfied since $\lambda < \min\{\gamma - 2\beta, 1-\gamma\}$, we have

$$\mathbb{E}[V_n] \leq \exp\left(-c_\gamma\mu\lambda_0 n^{1-(\lambda+\gamma)}(1-\varepsilon'(n))\right)\left(K_1^{(1)} + K_{1'}^{(1)}\max_{1\leq k\leq n+1} k^{\gamma-2\beta-\delta/2-(q/2+1)\lambda}\sqrt{v_k}\right)$$
$$+ K_2^{(1)}n^{-(\gamma-2\beta-\lambda)} + K_3^{(1)}\sqrt{v_{\lfloor n/2\rfloor}}n^{-(\delta+q\lambda)/2},$$

with

$$\varepsilon'(n) = \frac{2C_M n^{-1+\lambda+\gamma}}{\mu\lambda_0}\left(1 + \frac{n^{(1+2\beta-2\gamma)^+}}{|2\gamma-2\beta-1|}\right), \tag{19}$$

$$K_1^{(1)} = \left(\mathbb{E}[V_0] + \frac{C_1 L_{\nabla G}c_\gamma c_\beta^2}{C_M}\right), \quad K_1^{(1')} = 4\frac{\lambda_0\mu V}{C_M}, \tag{20}$$

where $C_M$ is given in equation 16 and $V$ in Lemma 7.1 and

$$K_2^{(1)} = 2^{\gamma-2\beta-\lambda-1}\frac{C_1 L_{\nabla G}c_\gamma c_\beta^2}{\mu\lambda_0}, \quad K_3^{(1)} = 2^{1+(\delta+q\lambda)/2}V. \tag{21}$$

**Lemma 7.1.** *Suppose Assumption **(A1)** for $p \geq 2$ and **(H1b)** hold. Then, for all $n \geq 0$, if $\gamma > 1/2$ then*

$$\mathbb{E}\left[V_n^p\right] \leq e^{a_p c_\gamma^2 c_\beta^2 \frac{2\gamma - 2\beta}{2\gamma - 2\beta - 1}} \max\left\{1, \mathbb{E}\left[V_0^2\right]\right\} := V_n^p$$

*and if $\gamma \leq 1/2$ then*

$$\mathbb{E}\left[V_n^p\right] \leq \exp\left(-p\mu\lambda_0' c_\gamma \left(1 + \frac{1 + \left(\frac{c_\gamma c_\beta^2 a_p}{p\mu\lambda_0'}\right)^{\frac{1-\gamma-\lambda'}{\gamma - 2\beta - \lambda'}}}{1 - \gamma - \lambda'}\right) + c_\gamma^2 c_\beta^2 a_p \left(1 + \frac{1 + \left(\frac{c_\gamma c_\beta^2 a_p}{p\mu\lambda_0'}\right)^{\frac{1-2\gamma+2\beta}{\gamma - 2\beta - \lambda'}}}{1 - 2\gamma + 2\beta}\right)\right) =: V_p^p$$

*with $a_2$ given in equation 79 and $a_p$ is given by equation 78 for $p > 2$.*

The proof of this Lemma is given in Section B.

### 7.4 Proof of Theorem 3.2

Remark that thanks to Assumption **(H1b)**, one has

$$\mathbb{E}\left[\|A_n\|^2 \left\|g_{n+1}'\right\|^2 |\mathcal{F}_n\right] \leq C_1 \|A_n\|^2 + \frac{C_2 L_{\nabla G}}{\mu} \|A_n\|^2 V_n \leq C_1 \|A_n\|^2 + \beta_{n+1}^2 \frac{C_2 L_{\nabla G}}{\mu} V_n.$$

Moreover, with the help of Assumption **(H2a)**,

$$\mathbb{E}\left[\|A_n\|^2 \left\|g_{n+1}'\right\|^2\right] \leq C_1 C_S^2 + \beta_{n+1}^2 \frac{C_2 L_{\nabla G}}{\mu} V_n$$

leading as in the proof of Theorem 3.1 to

$$\mathbb{E}\left[V_{n+1}\right] \leq \left(1 - 2\mu\lambda_0\gamma_{n+1} + \frac{C_2 L_{\nabla G}}{\mu}\beta_{n+1}^2\gamma_{n+1}^2\right)\mathbb{E}\left[V_n\right] + 2\lambda_0\gamma_{n+1}\mu\mathbb{E}\left[\mathbf{1}_{\lambda_{\min}(A_n)<\lambda_n} V_n\right]$$
$$+ \frac{C_1 L_{\nabla G} C_S^2}{2}\gamma_{n+1}^2.$$

Using Hölder inequality with $p$ yields then

$$\mathbb{E}\left[\mathbf{1}_{\lambda_{\min}(A_n)<\lambda_n} V_n\right] \leq \left(\mathbb{P}\left[\mathbf{1}_{\lambda_{\min}(A_n)<\lambda_n}\right]\right)^{\frac{p-1}{p}} \mathbb{E}\left[V_n^p\right]^{1/p} \leq \bar{v}_n^{\frac{p-1}{p}} V_p$$

with $\bar{v}_n = v_{n+1}(n+1)^{-\delta}$ and $V_p$ given in Lemma 7.1. Considering $C_M$ defined by

$$C_M = \max\left\{\frac{C_2 L_{\nabla G} c_\beta^2 c_\gamma}{\mu}, (\mu\lambda_0)^{\frac{2\gamma - 2\beta}{\gamma}} c_\gamma^{\frac{\gamma - 2\beta}{\gamma}}\right\}, \tag{22}$$

one has

$$\mathbb{E}\left[V_{n+1}\right] \leq \left(1 - 2\mu\lambda_0\gamma_{n+1} + C_M(n+1)^{2\beta - \gamma}\gamma_{n+1}\right)\mathbb{E}\left[V_n\right] + 2\lambda_0\mu V_p \bar{v}_n^{\frac{p-1}{p}}\gamma_{n+1}$$
$$+ \frac{C_1 L_{\nabla G} C_S^2}{2}\gamma_{n+1}^2.$$

Then, applying Proposition 7.1 and with the help of integral tests for convergence, it comes

$$\mathbb{E}\left[V_n\right] \leq \exp\left(-c_\gamma\mu\lambda_0 n^{1-\gamma}\right)\exp\left(2C_M c_\gamma\left(1 + \frac{n^{(1+2\beta-2\gamma)^+}}{|2\gamma - 2\beta - 1|}\right)\right) \cdot$$
$$\left(\mathbb{E}\left[V_0\right] + 4\frac{\lambda_0\mu V_p \max_{1\leq k\leq n} k^{\gamma - 2\beta}\bar{v}_k^{\frac{p-1}{p}}}{C_M} + \frac{C_1 L_{\nabla G} c_\gamma C_S^2}{C_M}\right) + 2V_p \bar{v}_{n/2}^{\frac{p-1}{p}}$$
$$+ 2^{\gamma-1}\frac{C_1 L_{\nabla G} c_\gamma C_S^2}{\mu\lambda_0} n^{-\gamma}. \tag{23}$$

Concluding as in the proof of Theorem 3.1, we get

$$\mathbb{E}\left[V_n\right] \le \exp\left(-c_\gamma \mu \lambda_0 n^{1-\gamma}\left(1 - \varepsilon(n)\right)\right) \cdot \left(K_1^{(2)} + K_{1'}^{(2)} \max_{1 \le j \le n+1} v_k^{\frac{p-1}{p}} k^{\gamma - 2\beta - \frac{p-1}{p}\delta}\right)$$
$$+ K_2^{(2)} v_{\lfloor n/2 \rfloor}^{\frac{p-1}{p}} n^{-\frac{(p-1)}{p}\delta} + K_3^{(2)} n^{-\gamma},$$

with

$$\varepsilon(n) = \frac{2C_M n^{-1+\gamma}}{\mu \lambda_0}\left(1 + \frac{n^{(1+2\beta-2\gamma)^+}}{|2\gamma - 2\beta - 1|}\right), \tag{24}$$

where $C_M$ is defined by equation 22 and

$$K_1^{(2)} = \left(\mathbb{E}\left[V_0\right] + \frac{C_1 L_{\nabla G} c_\gamma C_S^2}{C_M}\right), \quad K_{1'}^{(2)} = 4\frac{\lambda_0 \mu V_p}{C_M}, \tag{25}$$

$$K_2^{(2)} = 2^{1+\delta\frac{p-1}{p}} V_p, \tag{26}$$

$$K_3^{(2)} = 2^{\gamma-1} \frac{C_1 L_{\nabla G} c_\gamma C_S^2}{\mu \lambda_0}. \tag{27}$$

## 7.5 Proofs of Theorem 3.3 and Corollary 3.1

*Proof of Theorem 3.3.* Remark that one can rewrite

$$\theta_{n+1} - \theta = \theta_n - \theta - \gamma_{n+1} H^{-1} g'_{n+1} - \gamma_{n+1}\left(A_n - H^{-1}\right) g'_{n+1}$$

leading, since $H$ is symmetric, to

$$\|\theta_{n+1} - \theta\|^2 \le \|\theta_n - \theta\|^2 - 2\gamma_{n+1}\left\langle g'_{n+1}, H^{-1}\left(\theta_n - \theta\right)\right\rangle - 2\gamma_{n+1}\left\langle\left(A_n - H^{-1}\right) g'_{n+1}, \theta_n - \theta\right\rangle$$
$$+ 2\gamma_{n+1}^2 \left\|H^{-1} g'_{n+1}\right\|^2 + 2\gamma_{n+1}^2 \left\|A_n - H^{-1}\right\|^2 \left\|g'_{n+1}\right\|^2$$

First, thanks to Assumption **(A3)** and by Cauchy-Schwarz inequality,

$$(*) := \left|\mathbb{E}\left[2\gamma_{n+1}\left\langle\left(A_n - H^{-1}\right) g'_{n+1}, \theta_n - \theta\right\rangle | \mathcal{F}_n\right]\right| = 2\gamma_{n+1}\left|\left\langle\left(A_n - H^{-1}\right)\nabla G\left(\theta_n\right), \theta_n - \theta\right\rangle\right|$$
$$\le 2L_{\nabla G}\gamma_{n+1}\left\|A_n - H^{-1}\right\|\left\|\theta_n - \theta\right\|^2.$$

Then, using Assumption **(A1')**, one has

$$(**) := \mathbb{E}\left[2\gamma_{n+1}^2 \left\|H^{-1} g'_{n+1}\right\|^2 | \mathcal{F}_n\right] \le 4\gamma_{n+1}^2 \mathrm{Tr}\left(H^{-1}\Sigma H^{-1}\right) + 4\gamma_{n+1}^2 \left\|H^{-1}\right\|^2 L_{\nabla g}\left\|\theta_n - \theta\right\|^2$$

Finally, one has

$$(***) = \mathbb{E}\left[-2\gamma_{n+1}\left\langle g'_{n+1}, H^{-1}\left(\theta_n - \theta\right)\right\rangle | \mathcal{F}_n\right] \le -2\gamma_{n+1}\left\|\theta_n - \theta\right\|^2 + 2\gamma_{n+1}\left\|H^{-1}\right\|\left\|\delta_n\right\|\left\|\theta_n - \theta\right\|$$

with, using Assumption **(A5)**, $\|\delta_n\| := \|\nabla G\left(\theta_n\right) - H\left(\theta_n - \theta\right)\| \le L_\delta \|\theta_n - \theta\|^2$. Hence,

$$(***) \le -2\gamma_{n+1}\left\|\theta_n - \theta\right\|^2 + 2\gamma_{n+1}\left\|H^{-1}\right\| L_\delta \left\|\theta_n - \theta\right\|^3,$$

which yields, using that $\|\theta_n - \theta\|^3 \le \frac{1}{2a}\|\theta_n - \theta\|^2 + \frac{a}{2}\|\theta_n - \theta\|^4$ with $a = \left\|H^{-1}\right\| L_\delta$,

$$(***) \le -\gamma_{n+1}\left\|\theta_n - \theta\right\|^2 + \gamma_{n+1}\left\|H^{-1}\right\|^2 L_\delta^2 \left\|\theta_n - \theta\right\|^4.$$

Furthermore,

$$(****) := \mathbb{E}\left[2\gamma_{n+1}^2 \left\|A_n - H^{-1}\right\|^2 \left\|g'_{n+1}\right\|^2 | \mathcal{F}_n\right]$$
$$\le 2\gamma_{n+1}^2 \left\|A_n - H^{-1}\right\|^2 C_1 + 2\gamma_{n+1}^2 C_2 \left\|A_n - H^{-1}\right\|^2 \left\|\theta_n - \theta\right\|^2$$
$$\le 2\gamma_{n+1}^2 \left\|A_n - H^{-1}\right\|^2 C_1 + C_2\gamma_{n+1}\left\|\theta_n - \theta\right\|^4 + C_2\gamma_{n+1}^3 \left\|A_n - H^{-1}\right\|^4.$$

As a conclusion, one has (after using Cauchy-Schwartz inequality on $(*)$),

$$\mathbb{E}\left[\|\theta_{n+1} - \theta\|^2\right] \le \left(1 - \gamma_{n+1} + 4\left\|H^{-1}\right\|^2 \gamma_{n+1}^2 L_{\nabla g}\right)\mathbb{E}\left[\|\theta_n - \theta\|^2\right] + 4\gamma_{n+1}^2 \operatorname{Tr}\left(H^{-1}\Sigma H^{-1}\right)$$

$$+ \gamma_{n+1}\left(\left\|H^{-1}\right\|^2 L_\delta^2 + C_2\right)\mathbb{E}\left[\|\theta_n - \theta\|^4\right] + C_2\gamma_{n+1}^3\mathbb{E}\left[\left\|A_n - H^{-1}\right\|^4\right]$$

$$+ 2C_1\gamma_{n+1}^2\mathbb{E}\left[\left\|A_n - H^{-1}\right\|^2\right] + 2\gamma_{n+1}L_{\nabla G}\sqrt{\mathbb{E}\left[\|\theta_n - \theta\|^4\right]\mathbb{E}\left[\left\|A_n - H^{-1}\right\|^2\right]},$$

leading, using Proposition 3.2 with the fact that $\mathbb{E}\left[\|\theta_n - \theta\|^4\right] \le \frac{4}{\mu^2}\mathbb{E}\left[V_n^2\right]$ by **(A2)**, and **(H2b)** and **(H3)**, to

$$\mathbb{E}\left[\|\theta_{n+1} - \theta\|^2\right] \le \left(1 - \gamma_{n+1} + 4\left\|H^{-1}\right\|^2 \gamma_{n+1}^2 L_{\nabla g}\right)\mathbb{E}\left[\|\theta_n - \theta\|^2\right] + 4\gamma_{n+1}^2 \operatorname{Tr}\left(H^{-1}\Sigma H^{-1}\right)$$

$$+ \gamma_{n+1}\left(\left\|H^{-1}\right\|^2 L_\delta^2 + C_2\right)\frac{M_n}{\mu^2} + C_2\gamma_{n+1}^3 2^3\left(C_S^4 + \frac{1}{\mu^4}\right)$$

$$+ 2C_1\gamma_{n+1}^2 v_{A,n} + 2\gamma_{n+1}\frac{L_{\nabla G}}{\mu}\sqrt{M_n v_{A,n}}$$

$$\le \left(1 - \gamma_{n+1} + 4\left\|H^{-1}\right\|^2 \gamma_{n+1}^2 L_{\nabla g}\right)\mathbb{E}\left[\|\theta_n - \theta\|^2\right]$$

$$+ \gamma_{n+1}\cdot\left[4\gamma_{n+1}\operatorname{Tr}\left(H^{-1}\Sigma H^{-1}\right) + \left(\frac{L_\delta^2}{\mu^2} + C_2\right)\frac{4M_n}{\mu^2}\right.$$

$$\left. + C_2\gamma_{n+1}^2 2^3\left(C_S^4 + \frac{1}{\mu^4}\right) + 2C_1\gamma_{n+1}v_{A,n} + 4\frac{L_{\nabla G}}{\mu}\sqrt{M_n v_{A,n}}\right].$$

Finally, let us denote $C_A = c_\gamma \max\left\{4\left\|H^{-1}\right\|^2 L_{\nabla g}, \frac{1}{4}\right\}$. Then, with the help of Proposition 7.1, one has

$$\mathbb{E}\left[\|\theta_n - \theta\|^2\right] \le e^{-\frac{1}{2}c_\gamma n^{1-\gamma}}e^{2C_A c_\gamma \frac{2\gamma}{2\gamma-1}}\left(\mathbb{E}\left[\|\theta_0 - \theta\|^2\right] + \frac{8\operatorname{Tr}\left(H^{-1}\Sigma H^{-1}\right)}{C_A} + c_\gamma\frac{16C_2\left(\mu^{-4} + C_S^4\right)}{C_A} + \frac{4C_1 v_{A,0}}{C_A}\right)$$

$$+ e^{-\frac{1}{2}c_\gamma n^{1-\gamma}}e^{2C_A c_\gamma \frac{2\gamma}{2\gamma-1}}\max_{1\le k\le n}(k+1)^\gamma\cdot\left(8\frac{L_\delta^2\mu^{-2} + C_2}{\mu^2 C_A}M_{k-1} + 8\frac{L_{\nabla G}}{C_A\mu}\sqrt{M_{k-1}v_{A,k-1}}\right)$$

$$+ \frac{2^{3+\gamma}c_\gamma\operatorname{Tr}\left(H^{-1}\Sigma H^{-1}\right)}{n^\gamma} + \frac{8\left(\frac{L_\delta^2}{\mu^2} + C_2\right)}{\mu^2}M_{n/2} + \frac{8L_{\nabla G}}{\mu}\sqrt{M_{n/2}v_{A,n/2}}$$

$$+ \frac{2^{4+2\gamma}C_2 c_\gamma\left(\mu^{-4} + C_S^4\right)c_\gamma^2}{n^{2\gamma}} + \frac{2^{2+\gamma}C_1 c_\gamma v_{A,n/2}}{n^\gamma}.$$

Finally,

$$\mathbb{E}\left[\|\theta_n - \theta\|^2\right] \le e^{-\frac{1}{2}c_\gamma n^{1-\gamma}}\left(K_1^{(3)} + K_{1'}^{(3)}\max_{0\le k\le n}d_k(k+1)^\gamma\right)$$

$$+ n^{-\gamma}\left(2^{3+\gamma}c_\gamma\operatorname{Tr}\left(H^{-1}\Sigma H^{-1}\right) + \frac{K_2^{(3)}}{n^\gamma} + K_{2'}^{(3)}v_{A,n/2}\right) + d_{\lfloor n/2\rfloor}.$$

with

$$K_1^{(3)} = e^{2C_A c_\gamma^2 \frac{2\gamma}{2\gamma-1}}\left(\mathbb{E}\left[\|\theta_0 - \theta\|^2\right] + \frac{8\operatorname{Tr}\left(H^{-1}\Sigma H^{-1}\right)}{C_A} + c_\gamma\frac{16C_2\left(\mu^{-4} + C_S^4\right)}{C_A} + \frac{4C_1 v_{A,0}}{C_A}\right), \qquad (28)$$

$$K_{1'}^{(3)} = \frac{1}{C_A}e^{2C_A c_\gamma^2 \frac{2\gamma}{2\gamma-1}}, \quad d_n = 8L_{\nabla G}\sqrt{M_n v_{A,n}} + 8\frac{L_\delta^2\mu^{-2} + C_2}{\mu^2}M_n, \qquad (29)$$

where we recall that $C_A = c_\gamma \max\left\{4\left\|H^{-1}\right\|^2 L_{\nabla g}, \frac{1}{4}\right\}$, and

$$K_2^{(3)} = 2^{4+2\gamma}C_2 c_\gamma\left(\mu^{-4} + C_S^4\right)c_\gamma^2, \quad K_{2'}^{(3)} = 2^{2+\gamma}C_1 c_\gamma. \qquad (30)$$

$\square$

*Proof of Corollary 3.1.* Remark that as long as $\delta \frac{p-2}{p} \geq 2\gamma$, by Proposition 3.2 and the following discussion,

$$
\begin{aligned}
\max_{0 \leq k \leq n} d_k (k+1)^\gamma = \max_{0 \leq k \leq n} \left( (k+1)^\gamma 8 L_{\nabla G} \sqrt{M_k v_{A,k}} + 8 \frac{L_\delta^2 \mu^{-2} + C_2}{\mu^2} M_k \right) \\
\leq \frac{8 L_{\nabla G} \sqrt{v_A, 0}}{c_\gamma} \sqrt{w_\infty(2\gamma)} + 8 \frac{L_\delta^2 \mu^{-2} + C_2}{\mu^2} w_\infty(\gamma).
\end{aligned}
$$

Likewise,

$$
M_{n/2} \leq \frac{2^{2\gamma} w_\infty(2\gamma)}{n^{2\gamma}}.
$$

Hence, plugging these inequalities into Theorem 3.3 yields

$$
\mathbb{E} \left[ \|\theta_n - \theta\|^2 \right] \leq n^{-\gamma} \left( 2^{3+\gamma} c_\gamma \operatorname{Tr} \left( H^{-1} \Sigma H^{-1} \right) + \frac{K_2^{(3')}}{n^\gamma} + K_{2'}^{(3')} v_{A,n/2} + K_{2''}^{(3')} \sqrt{v_{A,n/2}} \right)
$$
$$
+ K_1^{(3')} e^{-\frac{1}{2} c_\gamma n^{1-\gamma}},
$$

with

$$
K_1^{(3')} = K_1^{(3)} + K_1^{(3')} \left( \frac{8 L_{\nabla G} \sqrt{v_A, 0}}{c_\gamma} \sqrt{w_\infty(2\gamma)} + 8 \frac{L_\delta^2 \mu^{-2} + C_2}{\mu^2} w_\infty(\gamma) \right), \tag{31}
$$

$$
K_2^{(3')} = K_2^{(3)} + 2 \frac{L_\delta^2 \mu^{-2} + C_2}{\mu^2} 2^{2\gamma} w_\infty(2\gamma), \quad K_{2'}^{(3')} = K_{2'}^{(3)}, K_{2''}^{(3')} = 2^{2+\gamma} L_{\nabla G} \sqrt{w_\infty(2\gamma)}. \tag{32}
$$

$\square$

## 7.6 Proof of Theorem 3.4

To prove this theorem, we will apply Theorem 3.2. We first need to check that $(A_n)_{n \geq 0}$ satisfies Assumptions **(H1a)**, **(H1b)** and **(H2)**. Assumption **(H1b)** is given by construction (see equation 5) while **(H1a)** is given by the following lemma:

**Lemma 7.2.** *Assume (A1) is satisfied for some $p > 2$. Then, for all $0 < t < 1$,*

$$
\mathbb{P} \left[ \lambda_{\min} (A_n) < c_\beta t \right] \leq v_n t^{2p},
$$

*with*

$$
v_n = c_\beta^p \left( \left( \frac{1}{n} \sum_{i=1}^d a_k \right)^p + C_1'' + \frac{2^p C_2'' V_p^p}{\mu^p} \right).
$$

The proof is given in Appendix B. Remark that $\mathbb{E}[V_n^p] < +\infty$ by Lemma 7.1 with **(A1)**. Assume from now that $p > 2$ and let $p' = \frac{2(1-\gamma)}{2-\gamma} p$ and $\lambda = (1-\gamma)(\gamma - 2\beta)$. Remark that $\lambda < 1 - \gamma$, $\lambda < \gamma - 2\beta$ and $p' < p$. Hence, applying Proposition 3.1 with $\lambda_0 = c_\beta$, $\delta = 0$, $q = 2p$,

$$
\mathbb{E} \left[ V_n^{p'} \right] \leq \exp \left( -c_\gamma \mu \lambda_0 n^{1-(\lambda+\gamma)}(1 - \varepsilon'(n)) \right) \left( K_1^{(1')} + K_{1'}^{(1')} \max_{1 \leq k \leq n+1} k^{\gamma - 2\beta - \lambda - 2(p-p')\lambda} v_0^{\frac{p-p'}{p}} \right)
$$
$$
+ K_2^{(1')} n^{-p'(\gamma - 2\beta - \lambda)} + K_3^{(1')} v_0^{\frac{p-p'}{p}} (n+1)^{-2(p-p')\lambda},
$$

with $\epsilon'(n)$, $K_1^{(1')}$, $K_{1'}^{(1')}$, $K_2^{(1')}$ and $K_3^{(1')}$ respectively given in equation 67, equation 68 and equation 70 with $\lambda_0 = c_\beta$. By the choice of $\lambda, p'$ one has

$$
p'(\gamma - 2\beta - \lambda) = p \frac{2(1-\gamma)}{2-\gamma} \gamma (\gamma - 2\beta) = 2(p - p')\lambda,
$$

so that

$$\mathbb{E}\left[V_n^{p'}\right] \leq \tilde{K}_1 \exp\left(-c_\gamma \mu c_\beta n^{1-((1-\gamma)(\gamma-2\beta)+\gamma)}(1-\varepsilon'(n))\right) + \tilde{K}_2(n+1)^{-\frac{2(1-\gamma)\gamma(\gamma-2\beta)}{2-\gamma}p} := c_n \qquad (33)$$

with

$$\tilde{K}_1 = K_1^{(1')} + K_{1'}^{(1')} v_0^{\frac{\gamma}{2-\gamma}}, \quad \tilde{K}_2 = K_2^{(1')} + K_3^{(1')} v_0^{\frac{\gamma}{2-\gamma}}.$$

By strong convexity, one can so obtain a first rate of convergence of the estimates. The following lemma enables to ensure that **(H1a)** is satisfied, but with a possibly better rate than with Lemma 7.2.

**Lemma 7.3.** *Assume **(A1)** is satisfied for some $p > 2$. Then,*

$$\mathbb{P}[\lambda_{\min}(A_n) < \tilde{\lambda}_0] \leq \frac{v_0 \log(n+1)}{(n+1)^{\frac{2(1-\gamma)\gamma(\gamma-2\beta)}{2-\gamma}p\wedge 1}},$$

*with $\tilde{\lambda}_0 = \left[\frac{2(1-\gamma)}{2-\gamma}p\left(C_{\left(\frac{2(1-\gamma)}{2-\gamma}\right)}+1\right)\right]^{-\frac{2-\gamma}{2(1-\gamma)p}}$ and $v_0$ is given in equation 85 with $p' = \frac{2(1-\gamma)}{2-\gamma}p$.*

The proof is given in Appendix B. We can also deduce from equation 33 a bound on $\mathbb{E}\left[\|A_n\|^4\right]$ in case only **(A6)** holds.

**Lemma 7.4.** *Assume Assumptions **(A1)-(A6)** and **(A1')** hold for some $p > 2$. Then, for $\beta < \min\left\{\frac{(1-\gamma)\gamma(\gamma-2\beta)p}{4(2-\gamma)}, 1/4\right\}$, the sequence of random matrices $(A_n)$ defined by equation 2 verifies*

$$\mathbb{E}\left[\|A_n\|^4\right] \leq C_S^4,$$

*with $C_S^4$ given in equation 86.*

The proof is given in Appendix B. If the stronger hypothesis **(A6')** holds, an improved and simpler bound on $\mathbb{E}\left[\|A_n\|^4\right]$ can be reached, as next lemma shows.

**Lemma 7.5.** *Assume Assumptions **(A1)-(A6')** and **(A1')** hold for some $p > 2$. Then, for $\beta < \min\{\gamma/2 \wedge 1/4\}$, the sequence of random matrices $(A_n)$ defined by equation 2 verifies*

$$\mathbb{E}\left[\|A_n\|^4\right] \leq C_S^4,$$

*with $C_S^4$ given in equation 87.*

The proof is given in Appendix B. Theorem 3.4 is then a consequence of Theorem 3.2 whose hypotheses are satisfied thanks to Lemma 7.2, 7.3 and 7.4 (or 7.5). We then have

$$\mathbb{E}\left[V_n\right] \leq \exp\left(-c_\gamma \mu \tilde{\lambda}_0 n^{1-\gamma}(1-\varepsilon(n))\right) \cdot \left(K_1^{(2)} + K_{1'}^{(2)} \max_{1 \leq j \leq n+1} v_k^{\frac{p-1}{p}} k^{\gamma-2\beta-\frac{p-1}{p}\delta}\right)$$

$$+ K_2^{(2)} v_{\lfloor n/2 \rfloor}^{\frac{p-1}{p}} n^{-\frac{(p-1)}{p}\min\left\{\frac{2(1-\gamma)\gamma(\gamma-2\beta)p}{2-\gamma},1\right\}} + K_3^{(2)} n^{-\gamma}$$

with $K_1^{(2)}, K_{1'}^{(2)}, K_2^{(2)}$ and $K_3^{(2)}$ respectively given in equation 25, equation 26 and equation 27 with $\delta = \min\left\{\frac{2(1-\gamma)\gamma(\gamma-2\beta)p}{2-\gamma}, 1\right\}$, $\lambda_0$ given in equation 84, $v_n = v_0 \log(n+1)$ with $v_0$ given in equation 85 and $C_S$ given in equation 86 or equation 87 depending on whether **(A6)** or **(A6')** holds. By strong convexity

$$\mathbb{E}\left[\|\theta_n - \theta\|^2\right] \leq \tilde{K}_1^{(4)} \exp\left(-c_\gamma \mu \tilde{\lambda}_0 n^{1-\gamma}(1-\tilde{\varepsilon}(n))\right) + \tilde{K}_2^{(4)}\left(v_0 \log(n+1)\right)^{\frac{p-1}{p}} n^{-\frac{(p-1)}{p}\min\left\{\frac{2(1-\gamma)\gamma(\gamma-2\beta)p}{2-\gamma},1\right\}}$$

$$+ \tilde{K}_3^{(4)} n^{-\gamma},$$

with $\tilde{\lambda}_0$ defined in equation 84

$$\tilde{\varepsilon}(n) = \frac{2C_M n^{-1+(1-\gamma)(2\gamma-\beta)+\gamma}}{\mu\tilde{\lambda}_0}\left(1 + \frac{n^{(1+2\beta-2\gamma)^+}}{|2\gamma-2\beta-1|}\right), \qquad (34)$$

with

$$\tilde{K}_1^{(4)} = \frac{2}{\mu}\left(K_1^{(2)} + K_{1'}^{(2)} v_0\right), \quad \tilde{K}_2^{(4)} = \frac{2K_2^{(4)}}{\mu}, \quad \tilde{K}_3^{(4)} = \frac{2K_3^{(4)}}{\mu}. \qquad (35)$$

where $v_0$ is given in equation 85.

### 7.7 Proofs of Theorem 4.1, Corollary 4.1 and Theorem 4.2

The proof relies on the verification of each assumption needed in Theorem 3.3.

**Verifying Assumptions (A1), (A1') to (A6).** First, remark that

$$\left\| \nabla_h g\left(X, Y, h\right) \right\| \leq \left\| \left(X^T h - X^T \theta - \epsilon\right) X \right\| \leq |\epsilon| \left\| X \right\| + \left\| X \right\|^2 \left\| h - \theta \right\|.$$

Then, if $X$ and $\epsilon$ respectively admit moments of order $4p$ and $2p$, since $\epsilon$ and $X$ are independent,

$$\mathbb{E}\left[ \left\| \nabla_h g\left(X, Y, h\right) \right\|^{2p} \right] \leq \sigma_{(2p)} + C_{(2p)} \left\| h - \theta \right\|^{2p}$$

with $\sigma_{(t)} = 2^{t-1} \mathbb{E}\left[ |\epsilon|^t \right] \mathbb{E}\left[ \|X\|^t \right]$ and $C_{(t)} = 2^{t-1} \mathbb{E}\left[ \|X\|^{2t} \right]$. In a particular case, if $p \geq 2$, Assumption **(A1)** is verified. Furthermore, since for all $h$, $\nabla^2 G(h) = \mathbb{E}\left[ XX^T \right]$ is positive, **(A2)** to **(A4)** hold with $\mu = \lambda_{\min}\left( \mathbb{E}\left[ XX^T \right] \right) =: \lambda_{\min}$, $L_{\nabla G} = \lambda_{\max}\left( \mathbb{E}\left[ XX^T \right] \right) =: \lambda_{\max}$ and **(A5)** holds with $L_\delta = 0$. Finally Assumption **(A1')** is verified since

$$\mathbb{E}\left[ \left\| \nabla_h g\left(X, Y, h\right) - \nabla_h g\left(X, Y, \theta\right) \right\|^2 \right] = \mathbb{E}\left[ \left\| X^T(h - \theta)X \right\|^2 \right]$$
$$\leq \underbrace{\mathbb{E}\left[ \|X\|^4 \right]}_{=:L_{\nabla g}} \left\| h - \theta \right\|^2.$$

We can now prove Theorem 4.1

*Proof of Theorem 4.1.* **Verifying Assumption (H1) for stochastic Newton algorithm.** Let us first check Assumption **(H1)** for $\widetilde{S}_n = \frac{\alpha_n}{n+m}\left[ mS_0 + \sum_{i=1}^n X_i X_i^T \right]$.

**Lemma 7.6.** *Suppose that $X$ admits $4p$-moments, with $p > 2$. Then, for $\lambda_0 = \frac{1}{2\alpha_+ \mathbb{E}\|X\|^2}$, we have*

$$\mathbb{P}\left[ \lambda_{\min}\left( \widetilde{S}_n^{-1} \right) < \lambda_0 \right] \leq \tilde{v}_n$$

*with*

$$\tilde{v}_n = \frac{2^{p-1}}{\left( \mathbb{E}\left[ \|X\|^2 \right] \right)^p} \left( C_1(p) n^{1-p} \mathbb{E}\left[ |Z|^p \right] + C_2(p) n^{-p/2} \left( \mathbb{E}\left[ |Z|^2 \right] \right)^{p/2} + m^p \|S_0\|^p n^{-p} \right),$$

*where $Z = \|X\|^2 - \mathbb{E}\left[ \|X\|^2 \right]$ and $C_1(p), C_2(p)$ are numerical constants given in Rosenthal inequality, see Pinelis (1994).*

*If moreover $X$ is subgaussian with subgaussian norm $\|X\|_{\psi_2}$ and $m \leq \frac{n \mathbb{E}[\|X\|^2]}{2\|S_0\|}$, then one can set*

$$\bar{v}_n = 2 \exp\left( -\frac{cn \mathbb{E}[\|X\|^2]}{\|X\|_{\psi_2}} \right),$$

*with $c$ numeric.*

The proof is given in Section C. To deal with $\overline{S}_n = \frac{\|\widetilde{S}_n^{-1}\|}{\min(n^\beta, \|\widetilde{S}_n^{-1}\|)} \widetilde{S}_n$, one first needs the following control on the behavior of $\lambda_{\min}(\widetilde{S}_n)$. Set $H = \mathbb{E}\left[ XX^T \right]$.

**Proposition 7.2** (See Koltchinskii & Mendelson (2015), Theorem 1.5 and Theorem 3.3)**.** *Suppose that $0 < \lambda_{\min} I_d \leq H := \mathbb{E}\left[ XX^T \right] \leq \lambda_{\max} I_d$ and that there exists $L_{MK} > 0$ such that $\mathbb{E}\left[ \langle X, t \rangle^2 \right] \leq L_{MK} \mathbb{E}\left[ |\langle X, t \rangle| \right]$ for all $t \in \mathbb{S}^{d-1}$. Then, for $n \geq c_1 d$,*

$$\mathbb{P}\left[ \lambda_{\min}\left( \frac{1}{n} \sum_{i=1}^n X_i X_i^T \right) \leq c_2 \right] \leq 2 \exp\left( -c_3 n \right),$$

*with $c_1 = \frac{\lambda_{\max}^2 (16 L_{MK})^4}{\lambda_{\min}^2}$, $c_2 = \frac{\lambda_{\min}}{8\sqrt{2} L_{MK}^2}$ and $c_3 = \frac{1}{128 L_{MK}^4}$.*

Remark that the constant $c_1$, $c_2$ and $c_3$ are fairly explicit in terms of $L_{MK}$ and $\lambda_{\min}$. For the latter result and Lemma 7.6 and Proposition 7.2 we deduce Hypothesis (H1) for $\overline{S}_n$. We will need several times the threshold

$$n_0 = \max \left\{ c_1 d, \left( \frac{1}{c_\beta \alpha_- c_2} \left( 1 + \frac{1}{c_1 d} \right) \right)^{1/\beta} m, \frac{2\|S_0\|}{\mathbb{E}[\|X\|^2]} m \right\}. \tag{36}$$

**Lemma 7.7.** *Suppose that $X$ satisfies hypothesis of Proposition 7.2 and admits $4p$-moments, with $p > 2$. Then, for $\lambda_0 = \frac{1}{2\alpha_+ \mathbb{E}[\|X\|^2]}$, we have*

$$\mathbb{P}\left[\lambda_{\min}\left(\overline{S}_n^{-1}\right) < \lambda_0\right] \leq v_{n+1}(n+1)^{-p/2}$$

*with $\delta = p/2$, $v_{n+1} = (n+1)^\delta$ for $n \leq n_0$ and, for $n > n_0$,*

$$v_n = 2\exp(-c_3 n)n^{p/2} + \frac{2^{p-1}\left(C_2(p)\mathbb{E}\left[|Z|^2\right]^{p/2} + C_1(p)n^{1-p/2}\mathbb{E}\left[|Z|^p\right] + m^p \|S_0\|^p n^{-p/2}\right)}{\mathbb{E}\left[\|X\|^2\right]^p}, \tag{37}$$

*where $c_1$, $c_2$, $c_3$ are given in Proposition 7.2, $C_1(p)$ and $C_2(p)$ are numerical constants depending on $p$ and $Z = \|X\|^2 - \mathbb{E}\left[\|X\|^2\right]$.*

*In the case $X$ is subgaussian with subgaussian norm $\|X\|_{\psi_2}$, for $n \geq n_0$, one has instead*

$$v_n = 2\left[\exp(-c_3 n) + \exp\left(-\frac{cn(\mathbb{E}[\|X\|^2])}{\|X\|_{\psi_2}}\right)\right] n^{p/2}, \tag{38}$$

*with $c > 0$ same as in Lemma 7.6.*

The proof is given in Section C. In particular, in the subgaussian case and for $n \geq n_0$

$$v_n = O\left(\exp(-c'n)\right) \tag{39}$$

for some constant $c'$ only depending on $\|X\|_{\psi_2}$ and $\mathbb{E}[\|X\|^2]$. As a particular case, Assumption **(H1a)** is verified with a rate $\delta = p/2$ when $\gamma > 1/2$.

**Verifying Assumption (H2) for stochastic Newton algorithm.** A straightforward deduction of the above lemma is the following.

**Lemma 7.8.** *Suppose that hypothesis of Proposition 7.2 holds and that $X$ admits a moment of order $4p$ with $p > 2$. Then, for all $\kappa > 0$, we have*

$$\mathbb{E}\left[\|\bar{S}_n^{-1}\|^\kappa\right] \leq 2\beta_{n+1}^\kappa \exp(-c_3 n) + (\alpha_- c_2/2)^{-\kappa}$$

*for $n \geq c_1 d \vee m$ and*

$$\mathbb{E}\left[\|\bar{S}_n^{-1}\|^\kappa\right] \leq \left[\frac{c_1 d + 2}{\alpha_-}\|S_0^{-1}\|\right]^\kappa$$

*for $n \leq c_1 d \vee m$, with $c_1$, $c_2$, $c_3$ given in Proposition 7.2.*

The proof is given in Section C. Finally, the following proposition gives a precise bound for Assumption **(H2)**.

**Proposition 7.3.** *Suppose that hypothesis of Proposition 7.2 hold and that $X$ admits a moment of order $4p$ with $p > 2$. Then*

$$\mathbb{E}\left[\|\bar{S}_n^{-1}\|^2\right] \leq \max\left\{2c_\beta^2\left(\frac{2\beta}{ec_3}\right)^{2\beta} + (\alpha_- c_2/2)^{-2}, \left[\frac{c_1 d + 2}{\alpha_-}\|S_0^{-1}\|\right]^2\right\} \leq C_S^2$$

*and*

$$\mathbb{E}\left[\|\bar{S}_n^{-1}\|^4\right] \le \max\left\{2c_\beta^4\left(\frac{4\beta}{ec_3}\right)^{4\beta} + (\alpha_- c_2/2)^{-4}, \left[\frac{c_1 d + 2}{\alpha_-}\left\|S_0^{-1}\right\|\right]^4\right\} \le C_S^4$$

*for all $n \ge 0$, with $C_S := \max\left\{\left(2c_\beta^2\left(\frac{4\beta}{ec3}\right)^{2\beta} + (\alpha_- c_2/2)^{-2}\right)^2, \left[\frac{c_1 d + 2}{\alpha_-}\left\|S_0^{-1}\right\|\right]^4\right\}^{1/4}$*

The proof is given in Section C. Remark that $C_S = O(d)$.

**A first convergence result.** Since in the case of the linear model, one as $C_1 = \sigma_{(2)}, C_1' = \sigma_{(4)}, C_2 = C_{(2)}, C_2' = C_{(4)}, L_{\nabla G} = \lambda_{\max}, \mu = \lambda_{\min}, \lambda_0 = \frac{1}{2\mathbb{E}[\|X\|^2]} \delta = p/2$, Proposition 3.2 can now be written as follows:

**Proposition 7.4.** *Suppose that there is $p > 2$ such that $X, \epsilon$ respectively admit moments of orders $4p$ and $2p$. Suppose also that there is a positive constant $L_{MK}$ such that for any $h \in \mathbb{S}^{d-1}$, $\sqrt{\mathbb{E}[hXX^T h]} \le L_{MK}\mathbb{E}\left[|X^T h|\right]$. Then, denoting $\lambda_{\min}$ and $\lambda_{\max}$ the smallest and largest eigenvalues of $\mathbb{E}\left[XX^T\right]$,*

$$\mathbb{E}\left[V_n^2\right] \le \exp\left(-\frac{3c_\gamma\lambda_{\min}}{4\mathbb{E}\left[\|X\|^4\right]}n^{1-\gamma}\right)\left(K_{1,lin}^{(2')} + K_{1',lin}^{(2')}\max_{1\le k\le n+1}v_k^{\frac{p-2}{p}}k^{\gamma-\frac{p-2}{p}}\right)$$

$$+ K_{2,lin}^{(2')}n^{-2\gamma} + K_{3,lin}^{(2')}v_{\lfloor n/2\rfloor}^{(p-2)/p}n^{-(p-2)/2} := c_{n,lin}.$$

*with $v_n$ given in equation 37 in the general case and in equation 38 in the subgaussian case, and*

$$K_{1,lin}^{(2')} = e^{2a_{M,lin}\frac{2\gamma-2\beta}{2\gamma-2\beta-1}}\left(\mathbb{E}\left[V_0^2\right] + \frac{2a_{1,lin}c_\gamma^2}{a_{M,lin}}\right), \quad K_{1',lin}^{(2')} = e^{2a_{M,lin}\frac{2\gamma-2\beta}{2\gamma-2\beta-1}}\frac{4\lambda_{\min}V_{p,lin}^2}{a_{M,lin}\mathbb{E}\left[\|X\|^2\right]},$$

$$K_{2,lin}^{(2')} = \frac{2^{1+2\gamma}a_{1,lin}c_\gamma^2\mathbb{E}\left[\|X\|^2\right]}{3\lambda_{\min}}, \quad K_{3,lin}^{(2')} = \frac{2^{p/2+1}}{3}V_{p,lin}^2,$$

*where, recalling the notations $\sigma_{(t)} = 2^{t-1}\mathbb{E}\left[|\epsilon|^t\right]\mathbb{E}\left[\|X\|^t\right]$ and $C_{(t)} = 2^{t-1}\mathbb{E}\left[\|X\|^{2t}\right]$,*

$$a_{M,lin} := \max\left\{\left(\frac{2\lambda_{\max}C_{(2)}}{\lambda_{\min}} + \frac{2\lambda_{\max}^2}{\lambda_{\min}^2}\left(4C_{(2)} + C_{(4)}c_\gamma^2 c_\beta^2\right)\right)c_\gamma c_\beta^2, \left(\frac{3\lambda_{\min}}{4\mathbb{E}\left[\|X\|^2\right]}\right)^{\frac{2\gamma-2\beta}{\gamma}}c_\gamma^{\frac{\gamma-2\beta}{\gamma}}\right\},$$

*with $C_S$ given by Proposition 7.3, $a_{1,lin} := C_S^4\lambda_{\max}^2\left(\frac{16\lambda_{\max}^2\sigma_{(2)}^2\mathbb{E}[\|X\|^2]}{\lambda_{\min}^3} + \frac{\sigma_{(4)}c_\gamma}{2} + \frac{2C_{(2)}^2\mathbb{E}[\|X\|^2]}{\lambda_{\min}}\right)$ and*

$$\mathbb{E}\left[V_n^p\right] \le e^{a_{p,lin}c_\gamma^2 c_\beta^2\frac{2\gamma-2\beta}{2\gamma-2\beta-1}}\max\left\{1, \mathbb{E}\left[V_0^p\right]\right\} := V_{p,lin}^p$$

*where*

$$a_{p,lin} := p\left(\frac{C_{(2)}}{\lambda_{\min}} + \frac{\sigma_{(2)}}{2}\right) + 2^{p-2}(p-1)p\lambda_{\max}^2\left(c_\gamma^2 c_\beta^2\left(\sigma_{(4)} + \frac{4C_{(4)}}{\lambda_{\min}^2}\right) + \frac{2\sigma_{(2)}}{\lambda_{\min}} + \frac{4C_{(2)}}{\lambda_{\min}^2}\right)$$

$$+ 2^{p-2}(p-1)p\lambda_{\max}^p\left(c_\gamma^{2p-2}c_\beta^{2p-2}\left(\sigma_{(2p)} + \frac{2^p C_{(2p)}}{\lambda_{\min}^2}\right) + c_\gamma^{p-2}c_\beta^{p-2}\left(\frac{1}{2}\sigma_{(2p)} + \frac{2p}{\lambda_{\min}^2}\left(\frac{1}{2} + \sqrt{C_{(2p)}}\right)\right)\right). \quad (40)$$

Remark that putting together the above expressions yields that, in the subgaussian case and for $n \ge 2n_0$ for $n_0$ defined in equation 36,

$$c_{n,lin} = O\left(\exp(-Cn^{1-\gamma})\left(\mathbb{E}\left[V_0^2\right] + m^{\gamma+(\delta-1)(p-2)/p}V_{p,lin}^2\right) + n^{-2\gamma} + V_{p,lin}^2\exp\left(-\frac{p-2}{2p}c'n\right)\right). \quad (41)$$

**Verifying Assumption (H3) for stochastic Newton algorithm.** Hypothesis **(H3)** is then a straightforward combination of the convergence of $\overline{S}_n$ towards $H$, together with Hypothesis **(H2)**.

**Lemma 7.9.** *Suppose that $X$ admits moments of order $2p$ with $p > 4$, and let suppose as well that the distribution of $X$ satisfies hypothesis of Proposition 7.2. Then, for $n \geq n_0$ (with $n_0$ defined in equation 36),*

$$\mathbb{E}\left[\left\|\overline{S}_n^{-1} - H^{-1}\right\|^2\right] \leq \frac{16\alpha_+ \left(\|S_0\| + \left(\mathbb{E}\left[\|X\|^{2p}\right]\right)^{2/p}\right)}{\left(\lambda_{\min}\beta_n\right)^2} e^{-c_3(p-2)n/p} + \frac{2\mathbb{E}\left[\|X\|^4\right]}{(n+m)\left(\lambda_{\min}\alpha_- c_2/2\right)^2} \tag{42}$$

$$+ \frac{2\|mS_0 - H\|_F^2}{(n+m)^2\left(\lambda_{\min}\alpha_- c_2/2\right)^2} + \frac{2C_\alpha}{n^2(\lambda_{\min}\alpha_- c_2/2)^2}\left(\|S_0\|^2 + \mathbb{E}[\|X\|^2]\right) =: v_{H,n}. \tag{43}$$

For $n < n_0$, we simply bound

$$\mathbb{E}\left[\left\|\overline{S}_n^{-1} - H^{-1}\right\|^2\right] \leq \max\left\{\frac{2}{\lambda_{\min}^2} + 2C_S^2, v_{H,n_0}\right\} := v_{H,n}.$$

Remark that $v_{H,n} = O\left(\frac{1}{n}\right)$ uniformly on $m \geq 1$. By Lemma 7.7, **(H1a)** is satisfied with $\delta = p/2$. Applying Theorem 3.3 with the constants computed in the previous lemmas and proposition, we get finally,

$$\mathbb{E}\left[\|\theta_n - \theta\|^2\right] \leq e^{-\frac{1}{2}c_\gamma n^{1-\gamma}}\left(K_{1,\mathrm{lin}}^{(3)} + K_{1',\mathrm{lin}}^{(3)} \max_{0 \leq k \leq n} d_k(k+1)^\gamma\right)$$

$$+ n^{-\gamma}\left(2^{3+\gamma}c_\gamma \mathbb{E}\left[\epsilon^2\right]\mathrm{Tr}\left(H^{-1}\right) + \frac{K_{2,\mathrm{lin}}^{(3)}}{n^\gamma} + K_{2',\mathrm{lin}}^{(3)}v_{H,n/2}\right) + d_{\lfloor n/2\rfloor}.$$

with $v_{H,n}$ defined by equation 43, recalling that $\lambda_{\min}$ and $\lambda_{\max}$ are the smallest and largest eigenvalues of $\mathbb{E}\left[XX^T\right]$, and since for the linear case one has $C_A = 4c_\gamma \frac{\mathbb{E}[\|X\|^4]}{\lambda_{\min}^2} \geq 4c_\gamma$,

$$K_{1,\mathrm{lin}}^{(3)} = e^{8\frac{\mathbb{E}[\|X\|^4]}{\lambda_{\min}^2}c_\gamma^3 \frac{2\gamma}{2\gamma-1}}\left(\mathbb{E}\left[\|\theta_0 - \theta\|^2\right] + \frac{2\mathbb{E}\left[\epsilon^2\right]\mathrm{Tr}\left(H^{-1}\right)}{c_\gamma} + 4C_{(2)}\left(\lambda_{\min}^4 + C_S^4\right) + \frac{\sigma_{(2)}v_{H,0}}{c_\gamma}\right),$$

$$K_{1',\mathrm{lin}}^{(3)} = \frac{1}{4c_\gamma}e^{8\frac{\mathbb{E}[\|X\|^4]}{\lambda_{\min}^2}c_\gamma^3 \frac{2\gamma}{2\gamma-1}}, \quad d_n = 8\lambda_{\max}\sqrt{c_{n,\mathrm{lin}}v_{H,n}} + 8\frac{C_{(2)}}{\lambda_{\min}^2}c_{n,\mathrm{lin}},$$

$$K_{2,\mathrm{lin}}^{(3)} = 2^{4+2\gamma}C_{(2)}c_\gamma\left(\lambda_{\min}^{-4} + C_S^4\right)c_\gamma^2, \quad K_{2',\mathrm{lin}}^{(3)} = 2^{2+\gamma}\sigma_{(2)}c_\gamma, \tag{44}$$

and $c_{n,\mathrm{lin}}$ and $C_S^4$ are respectively defined in Propositions 7.4 and 7.3. □

From the asymptotic behavior of $v_{H,n}$ and $c_{n,lin}$ (see equation 41) and the bound on $V_{p,lin}$ in terms of $\mathbb{E}\left[V_0^p\right]$ given in Proposition 7.4, we deduce that, for $n \geq 2n_0$,

$$d_n = O\Bigg(\exp(-Cn^{1-\gamma})\left(\mathbb{E}\left[V_0^2\right] + m^{\gamma+(\delta-1)(p-2)/p}\mathbb{E}[V_0^p]^{2/p}\right)$$

$$+ \mathbb{E}[V_0^p]^{2/p}\exp\left(-\frac{p-2}{2p}c'n\right) + \frac{1}{n}\Bigg). \tag{45}$$

We can deduce Corollary 4.1 from Theorem 4.1. Remark first that we have the following rough bound on $\mathbb{E}\left[V_m^2\right]$ for the usual stochastic Newton algorithm with adaptive matrix $A_n = \tilde{S}_n^{-1}$ with $\alpha_n = 1$, $m = 1$.

**Lemma 7.10.** *Let $V_n = G(\theta_n)$ for the stochastic Newton algorithm with $A_n = \tilde{S}_n^{-1}$. Then,*

$$\mathbb{E}\left[V_n^p\right] \leq \exp\left((p-1)\log(2)n + C_{lin,1}\frac{n^{3-2\gamma}}{3-2\gamma}\right)\left(1 + C_{lin,2}\frac{n^{(p+1)-p\gamma}}{(p+1)-p\gamma}\right)\mathbb{E}\left[V_0^p\right],$$

*with $C_{lin,1}$ and $C_{lin,2}$ given in equation 91.*

*Proof of Corollary 4.1.* Set $m_0 = \lfloor R \log(n) \rfloor + 1$ with $R > 0$ to tune later, and let $(\theta_n)_{n \geq 0}$ be the sequence induced by the stochastic Newton algorithm with $m = 1, \alpha_n = 1$ for $n \geq 1$, initial estimated Hessian matrix $S_0$ and starting point $\theta_0$. Set $\tilde{\theta}_0 = \theta_{m_0}$, and set $\tilde{\theta}_n$ for the $n$-step of the regularized stochastic Newton algorithm with respect to the sequence of random variables $(X_{m_0+k}, \epsilon_{m_0+k})_{k \geq 1}$ parameters $m = m_0 + 1$, initial estimator of the Hessian $\tilde{S}_0 = \frac{1}{m_0+1}(S_0 + \sum_{k=1}^{m_0} X_k X_k^T)$, $\alpha_n = \frac{(n+1)^\gamma}{(n+1+m_0)^\gamma}$ and $\beta = 0$, $c_\beta = 2c_2^{-1}$ (with $c_2$ given in Proposition 7.2). Then, remark that on the event $B_{m_0} = \{\sigma_{\inf}(\tilde{S}_n) \geq c_2, n \geq 0\}$, $\tilde{\theta}_n = \theta_{m_0+n}$ : indeed on this event, by the choice of the previous parameters and a simple recursion,

$$\tilde{\theta}_{n+1} = \tilde{\theta}_n + \alpha_n c_\gamma n^{-\gamma} \tilde{A}_n \nabla_h g((X_{m_0+n+1}, \epsilon_{m_0+k+1}), \tilde{\theta}_n)$$

$$= \theta_{m_0+n} + c_\gamma (m_0 + n + 1)^{-\gamma} \left( \frac{1}{m_0 + n + 1} \left( S_0 + \sum_{k=1}^{m_0+n} X_k X_k^T \right) \right)^{-1} \nabla_h g((X_{m_0+n+1}, \epsilon_{m_0+n+1}), \theta_{m_0+n})$$

$$= \theta_{m_0+n+1}.$$

Hence, for $n \geq m_0 + 1$, by Markov's inequality

$$\mathbb{P}\left[\|\theta_n - \theta\| > \epsilon\right] = \mathbb{P}\left(\{\|\theta_n - \theta\| > \epsilon\} \cap B_{m_0}\right) + \mathbb{P}\left[\{\|\theta_n - \theta\| > \epsilon\} \cap B_{m_0}^c\right]$$

$$\leq \mathbb{P}\left[\{\|\tilde{\theta}_{n-m_0} - \theta\| > \epsilon\} \cap B_{m_0}\right] + \mathbb{P}\left[B_{m_0}^c\right]$$

$$\leq \frac{\mathbb{E}\left[\|\tilde{\theta}_{n-m_0} - \theta\|^2\right]}{\epsilon^2} + \mathbb{P}\left[B_{m_0}^c\right]. \tag{46}$$

Suppose that $n \geq 2\max(n_0, m_0)$, with $n_0$ given in equation 36 for $m = m_0$. By Theorem 4.1 and the fact that $n \geq 2m_0$,

$$\mathbb{E}\left[\|\tilde{\theta}_{n-m_0} - \theta\|^2\right] = (n - m_0)^{-\gamma} 2^{3+\gamma} c_\gamma \mathbb{E}\left[\epsilon^2\right] \text{Tr}\left(H^{-1}\right)$$

$$+ O\left(e^{-\frac{1}{2}c_\gamma(n-m_0)^{1-\gamma}} \left(\mathbb{E}[\|\tilde{\theta}_0 - \theta\|^2] + d_{n-m}(n+1)^\gamma) + \frac{1}{n} + d_{\lfloor n/2 \rfloor}\right). \tag{47}$$

Set $\tilde{V}_0 = G(\tilde{\theta}_0) - G(\theta)$ and let us bound $d_{n/2}$ and $d_{n-m_0}$. By Lemma 7.10 and the fact that $\tilde{V}_0 = V_{m_0}$, with $V_{m_0} = G(\theta_{m_0}) - G(\theta)$, and $m_0 = R \log n$,

$$\mathbb{E}[\tilde{V}_0^p] = O\left(\exp(Cm_0^{3-2\gamma}))\mathbb{E}[V_0^p]\right) = O\left(\exp\left(C' \log(n)^{3-2\gamma}\right) \mathbb{E}[V_0^p]\right), \tag{48}$$

for some constant $C, C' > 0$. Hence, by equation 45,

$$d_n = O\left(\exp\left(-Cn^{1-\gamma}\right) \mathbb{E}[V_0^p]^{2/p} + \frac{1}{n}\right)$$

for some constant $C > 0$ only depending on the first moments of $X$ and the parameters of the algorithm. Therefore, using the strong convexity and Lemma 7.10 to bound $\mathbb{E}[\|\tilde{\theta}_0 - \theta\|^2]$ as in equation 48, we finally get, for $n \geq 2\max(n_0, m_0)$,

$$O\left(e^{-\frac{1}{2}c_\gamma(n-m_0)^{1-\gamma}} \left(\mathbb{E}[\|\tilde{\theta}_0 - \theta\|^2] + d_{n-m}(n+1)^\gamma) + \frac{1}{n} + d_{\lfloor n/2 \rfloor}\right)$$

$$= O\left(e^{-\frac{1}{2}cn^{1-\gamma}} \left(\mathbb{E}[V_0^p]^{2/p}\right) + \frac{1}{n}\right), \tag{49}$$

with $c > 0$ only depending on the first moments of $X$ and the parameters of the algorithm. Finally, by Proposition 7.2 and choosing $R = \frac{2\gamma}{c_3}$ yields

$$\mathbb{P}\left[B_{m_0}^c\right] \leq \sum_{n \geq m_0} \mathbb{P}\left[\lambda_{\min}\left(\frac{1}{n}\sum_{i=1}^n X_i X_i^T\right) \leq c_2\right] \leq 2\sum_{n \geq m_0} \exp\left(-c_3 n\right)$$

$$\leq \frac{2}{1 - \exp(-c_3)} \exp(-c_3 m_0) \leq \frac{C}{n^{2\gamma}} \tag{50}$$

for some $C > 0$. Putting equation 50 and equation 47 together with equation 49 in equation 46 yields, for $n \geq 2\max(n_0, m_0)$,

$$\mathbb{P}\left[\|\theta_n - \theta\| > \delta\right] \leq \frac{1}{\delta^2}\left[n^{-\gamma}2^{4+\gamma}c_\gamma\mathbb{E}\left[\epsilon^2\right]\operatorname{Tr}\left(H^{-1}\right) + O\left(e^{-\frac{1}{2}cn^{1-\gamma}}\mathbb{E}[\tilde{V}_0^p]^{2/p} + \frac{1}{n}\right)\right] + O(n^{-2\gamma}).$$

Finally, since $m_0 = \lfloor R\log n \rfloor + 1$ with $R = \frac{2\gamma}{c_3}$ and $n_0 = \max(c_1 d, Cm_0)$ for some constant $C > 0$ depending on the second moment of $X$, $d$ and $S_0$ (see equation 36), we deduce that $n \geq 2\max(n_0, m_0)$ as long as $n \geq c_0$, where $c_0$ is a threshold only depending on $\gamma, d, S_0$ and the second moment of $X$. □

*Proof of Theorem 4.2.* Let us first prove that Assumption **(A6')** is fulfilled. For all $h$,

$$\mathbb{E}\left[\nabla_h g\left(X, Y, h\right)\nabla_h g\left(X, Y, h\right)^T\right] = \mathbb{E}\left[\left(Y - X^T h\right)^2 XX^T\right]$$
$$= \mathbb{E}\left[\epsilon^2\right]\mathbb{E}\left[XX^T\right] + \mathbb{E}\left[\left(X^T h - X^T\theta\right)^2 XX^T\right]$$

and **(A6')** is satisfied with $\alpha = \mathbb{E}\left[\epsilon^2\right]\lambda_{\min}$. Hence, we have by equation 87

$$\mathbb{E}\left[\|A_n\|^4\right] \leq \frac{4d\left(1 + \sigma_{(4)} + C_{(4)}\frac{4V_{2,ada}^2}{\lambda_{\min}^2}\right)}{\mathbb{E}\left[\epsilon^2\right]^2\lambda_{\min}^2} := C_{S,ada}^4,$$

with $V_2$ given by Lemma 7.1 for $p = 2$. Then, applying Theorem 3.4,

$$\mathbb{E}\left[\|\theta_n - \theta\|^2\right] \leq K_{1,lin}^{ada}\exp\left(-c_\gamma\lambda_{\min}\lambda_{0,lin}^{ada}n^{1-\gamma}\left(1 - \varepsilon_{n,lin}^{ada}\right)\right)$$
$$+ K_{2,lin}^{ada}\left(v_{0,lin}^{ada}\log(n+1)\right)^{\frac{p-1}{p}}n^{-\frac{(p-1)}{p}\min\left\{\frac{2(1-\gamma)\gamma(\gamma-2\beta)p}{2-\gamma}, 1\right\}} + K_{3,lin}^{ada}n^{-\gamma},$$

with $\lambda_{0,lin}^{ada} = \left[\frac{4(1-\gamma)p}{2-\gamma}\left(C_{\left(\frac{4p(1-\gamma)}{2-\gamma}\right)} + 1\right)\right]^{-\frac{2-\gamma}{4p(1-\gamma)}}$ and, recalling that $\lambda_{\min}$ and $\lambda_{\max}$ are the smallest and largest eigenvalues of $\mathbb{E}\left[XX^T\right]$,

$$\varepsilon_{n,lin}^{ada} = \frac{2C_{M,lin}^{ada}n^{-1+(1-\gamma)(2\gamma-\beta)+\gamma}}{\lambda_{\min}\lambda_{0,lin}^{ada}}\left(1 + \frac{n^{(1+2\beta-2\gamma)^+}}{|2\gamma - 2\beta - 1|}\right), \tag{51}$$

$$K_{1,lin}^{ada} = \frac{2}{\lambda_{\min}}\left(\mathbb{E}\left[V_0\right] + \frac{c_\gamma\lambda_{\max}\sigma_{(2)}C_{S,ada}^2}{C_{M,lin}^{ada}} + \frac{4\lambda_{\min}\lambda_{0,lin}^{ada}V_{p,lin}^{ada}}{C_{M,lin}^{ada}}\right), \tag{52}$$

$$K_{2,lin}^{ada} = \frac{1}{\lambda_{\min}}2^{p/2+3/2}V_{p,lin}^{ada} \tag{53}$$

$$K_{3,lin}^{ada} = \frac{2^\gamma c_\gamma\lambda_{\max}\sigma_{(2)}C_{S,ada}^2}{\lambda_{\min}^2\lambda_{0,lin}^{ada}}. \tag{54}$$

where $v_0 = dM(\beta) + \dfrac{d2^{\frac{2(1-\gamma)}{2-\gamma}p}\left(\sigma_{\left(\frac{4(1-\gamma)}{2-\gamma}p\right)} + 2^{\frac{2(1-\gamma)}{2-\gamma}p}C_{\left(\frac{4(1-\gamma)}{2-\gamma}p\right)}\frac{V_{p,ada}^{\frac{2(1-\gamma)}{2-\gamma}p}}{\lambda_{min}^{\frac{2(1-\gamma)}{2-\gamma}p}}\right)}{\sigma_{\left(\frac{4(1-\gamma)}{2-\gamma}p\right)} + 1}$.

$$C_{M,lin}^{ada} = \max\left\{\frac{C_{(2)}\lambda_{\max}c_\beta^2 c_\gamma}{\lambda_{\min}}, \left(\lambda_{\min}\lambda_{0,lin}^{ada}\right)^{\frac{2\gamma-2\beta}{\gamma}}c_\gamma^{\frac{\gamma-2\beta}{\gamma}}\right\}$$

and

$$V_{p,ada}^p = e^{-p\lambda_{\min}\lambda_0' c_\gamma\left(1 + \frac{1 + \left(\frac{c_\gamma c_\beta^2 a_{p,lin}^{ada}}{p\lambda_{\min}\lambda_0'}\right)^{\frac{1-\gamma-\lambda'}{\gamma-2\beta-\lambda'}}}{1-\gamma-\lambda'}\right) + c_\gamma^2 c_\beta^2 a_p\left(1 + \frac{1 + \left(\frac{c_\gamma c_\beta^2 a_{p,lin}^{ada}}{p\lambda_{\min}\lambda_0'}\right)^{\frac{1-2\gamma+2\beta}{\gamma-2\beta-\lambda'}}}{1-2\gamma+2\beta}\right)}$$

where

$$a_{p,lin}^{ada} = p\left(\frac{C_{(2)}}{\lambda_{\min}} + \frac{\sigma_{(2)}}{2}\right) + 2^{p-2}(p-1)p\lambda_{\max}^2\left(c_\gamma^2 c_\beta^2\left(\sigma_{(4)} + \frac{4C_{(4)}}{\lambda_{\min}^2}\right) + \frac{2\sigma_{(2)}}{\mu} + \frac{4C_{(2)}}{\lambda_{\min}^2}\right)$$
$$+ 2^{p-2}(p-1)p\lambda_{\max}^p\left(c_\gamma^{2p-2}c_\beta^{2p-2}\left(\sigma_{(2p)} + \frac{2^P C_{(2p)}}{\lambda_{\min}^2}\right) + c_\gamma^{p-2}c_\beta^{p-2}\left(\frac{1}{2}\sigma_{(2p)} + \frac{2p}{\lambda_{\min}^2}\left(\frac{1}{2} + \sqrt{C_{(2p)}}\right)\right)\right), \quad (55)$$

and

$$a_{2,lin}^{ada} = \sigma_{(2)} + \frac{2C_{(2)}}{\lambda_{\min}} + \frac{4\lambda_{\max}^2}{\lambda_{\min}}\sigma_{(2)} + \frac{8\lambda_{\max}^2 C_{(2)}}{\lambda_{\min}^2} + 2\lambda_{\max}^2\sigma_{(4)}c_\gamma^2 c_\beta^2 + \frac{8\lambda_{\max}^2 C_{(4)}}{\lambda_{\min}^2}c_\gamma^2 c_\beta^2 \quad (56)$$

$\square$

## 7.8 Proof of Theorem 5.1

The proof relies on the verification of each Assumption in Theorem 3.3.

**Verifying Assumptions (A1), (A1') to (A6).** First, remark that taking for all $0 \le a \le 2p$, one has

$$\mathbb{E}\left[\left\|\nabla_h l\left(Y, X^T h\right)X + \sigma h\right\|^a\right] \le 2^{a-1}\mathbb{E}\left[\left\|\nabla_h l\left(Y, X^T\theta_\sigma\right)X + \sigma\theta_\sigma\right\|^a\right]$$
$$+ 2^{a-1}\mathbb{E}\left[\left\|\nabla_h l\left(Y, X^T h\right)X - \nabla_h \ell\left(Y, X^T\theta_\sigma\right)X + \sigma\left(h - \theta_\sigma\right)\right\|^a\right]$$
$$\le 2^{a-1}L_\sigma^a + 2^{a-1}\underbrace{\mathbb{E}\left[\left(L_{\nabla l}\|X\| + \sigma\right)^a\right]}_{=:C_{\text{GLM}}^{(a)}}\|h - \theta_\sigma\|^a \quad (57)$$

and Assumption **(A1)** is so verified. In a same way,

$$\mathbb{E}\left[\left\|\left(\nabla_h g\left(X, h\right) - \nabla_h g\left(X, \theta_\sigma\right)\right)\right\|^2\right] \le \mathbb{E}\left[\left(L_{\nabla l}\|X\| + \sigma\right)^2\right]\|h - \theta_\sigma\|^2 \le C_{\text{GLM}}^{(2)}\|h - \theta_\sigma\|^2$$

and **(A1')** is so verified. Remark that **(A2)** and **(A4)** are satisfied by hypothesis. For **(A3)**, one has

$$\left\|\mathbb{E}\left[\nabla_h^2\ell\left(Y, X^T h\right)XX^T + \sigma I_d\right]\right\|_{op} \le L_{\nabla l}\mathbb{E}\left[\|X\|^2\right] + \sigma =: C_{\text{GLM}}. \quad (58)$$

Observe that Assumption **(A5)** is given by **(GLM1)** while for Assumption **(A6)**, **(GLM3)** together equation 11, which yields

$$\mathbb{E}\left[\left(\nabla_h g(X, \theta_v)\right)_k^2\right] = \mathbb{E}\left[\left|\nabla_h l\left(Y, X^T\theta_\sigma\right)X_k + \sigma(\theta_\sigma)_k\right|^2\right] > \alpha_\sigma$$

for all $1 \le k \le d$.

**Verifying Assumption (H1).** The following lemma ensures that Assumption **(H1)** is fulfilled.

**Lemma 7.11.** *Assume first equation 7 and that $X$ admits a moment of order $2p$ for some $p > 2$. In the regularized case defined by equation 9, denoting $\lambda_0 = \frac{1}{2L_{\nabla l}\mathbb{E}[\|X\|^2] + 2\lambda}$, we have*

$$\mathbb{P}\left[\lambda_{\min}\left(\overline{S}_n^{-1}\right) < \lambda_0\right] \le v_{n+1}(n+1)^{-p/2}$$

*with*

$$v_{n+1} = \frac{2^{p-1}}{\left(L_{\nabla l}\mathbb{E}\left[\|X\|^2\right] + \sigma\right)^p}\left(\frac{(n+1)^{p/2}}{n^p}\|S_0\|^p + C_1(p)\frac{(n+1)^{p/2}}{n^{p-1}}\mathbb{E}\left[|T|^p\right]\right.$$
$$\left. + C_2(p)\left(\frac{n+1}{n}\right)^{p/2}\left(\mathbb{E}\left[|T|^2\right]\right)^{p/2}\right),$$

*where $T = L_{\nabla l}\left(\|X\|^2 - \mathbb{E}\left[\|X\|^2\right]\right) + \sigma\left(\|Z\|^2 - 1\right)$ and $Z$ being a standard $d$-dimensional random variable independent of $X$. In addition, $C_1(p)$ and $C_2(p)$ are given in Pinelis (1994).*

The proof is given in Appendix D.

**Verifying Assumption (H2).** The following proposition ensures that **(H2)** is fulfilled.

**Proposition 7.5.** *Considering from the regularized problem given by equation 9, one has for all $n \geq 0$,*

$$\left\| \bar{S}_n^{-1} \right\| \leq 2d \max \left\{ \frac{1}{\sigma}, \left\| S_0^{-1} \right\| \right\} =: C_{S,\sigma}$$

**Remark 7.1.** *Remark that if equation 8 holds for some constant $\alpha > 0$ and if $\mathbb{E}\left[XX^T\right]$ is positive, under hypothesis of Proposition 7.2, for all $n \geq 0$ and for $\sigma = 0$, one has*

$$\mathbb{E}\left[\|\bar{S}_n^{-1}\|^2\right] \leq \frac{1}{\alpha^2} \max \left\{ 2c_\beta^2 \left(\frac{2\beta}{ec_3}\right)^{2\beta} + c_2^{-2}, \left((c_1 d + 1)\left\|S_0^{-1}\right\|\right)^2 \right\} \leq C_{S,0}^2,$$

$$\mathbb{E}\left[\|\bar{S}_n^{-1}\|^4\right] \leq \frac{1}{\alpha^4} \max \left\{ 2c_\beta^4 \left(\frac{2\beta}{ec_3}\right)^{4\beta} + c_2^{-4}, \left((c_1 d + 1)\left\|S_0^{-1}\right\|\right)^4 \right\} \leq C_{S,0}^4$$

*with $C_{S,0}^4 = \frac{1}{\alpha^4} \max \left\{ \left(2c_\beta^2 \left(\frac{2\beta}{ec_3}\right)^{2\beta} + c_2^{-2}\right)^2, \left((c_1 d + 1)\left\|S_0^{-1}\right\|\right)^4 \right\}$.*

**A first result**

Remark that one can rewrite Proposition 3.2 as follows:

**Proposition 7.6.** *Suppose there exists $p > 2$ such that $X$ admits a $2p$-th order moment and that there is $L_\sigma$ verifying*

$$\mathbb{E}\left[\left|\nabla_h l\left(Y, X^T\theta_\sigma\right)\right|^p \|X\|^p\right] + \sigma\theta_\sigma \leq L_\sigma^p. \tag{59}$$

*Then,*

$$\mathbb{E}\left[V_n^2\right] \leq \exp\left(-\frac{3c_\gamma\sigma}{4C_{GLM}}n^{1-\gamma}\right)\left(K_{1,GLM}^{(2')} + K_{1',GLM}^{(2')} \max_{1 \leq k \leq n+1} v_k^{\frac{p-2}{p}} k^{\gamma - \frac{p-2}{2}}\right)$$
$$+ K_{2,GLM}^{(2')} n^{-2\gamma} + K_{3,GLM}^{(2')} v_{\lfloor n/2 \rfloor}^{(p-2)/p} n^{-(p-2)/2} =: v_{n,GLM},$$

*with $v_n$ defined in Lemma 7.11, $C_{S,\sigma}$ defined in Lemma 7.5, $C_{GLM}$ and $C_{GLM}^{(a)}$ defined in equations equation 58 and equation 57,*

$$a_{1,GLM} = C_{S,\sigma}^4 C_{GLM}^2 \left(\frac{64L_\sigma^4 C_{GLM}^5}{\sigma^3} + 4c_\gamma L_\sigma^4 + \frac{4L_\sigma^4 C_{GLM}}{\sigma}\right)$$

$$a_{M,GLM} = \max \left\{ \left(\frac{4C_{GLM}C_{GLM}^{(2)}}{\sigma} + \frac{2C_{GLM}^2}{\sigma^2}\left(8C_{GLM}^{(2)} + 8C_{GLM}^{(4)}c_\gamma^2 C_{S,\sigma}^2\right)\right)c_\gamma C_{S,\sigma}^2, \left(\frac{3\sigma}{4C_{GLM}}\right)^2 c_\gamma \right\}$$

$$K_{1,GLM}^{(2')} = \exp\left(2a_{M,GLM}\frac{2\gamma}{2\gamma - 1}\right)\left(\mathbb{E}\left[V_0^2\right] + \frac{2a_{1,GLM}c_\gamma^2}{a_{M,GLM}}\right)$$

$$K_{1',GLM}^{(2')} = \exp\left(2a_{M,GLM}\frac{2\gamma}{2\gamma - 1}\right) \cdot \frac{4\sigma V_{p,GLM}^2}{a_{M,GLM}C_{GLM}}$$

$$K_{2,GLM}^{(2')} = \frac{2^{2\gamma + 1}a_{1,GLM}C_{GLM}c_\gamma^2}{3\sigma}$$

$$K_{3,GLM}^{(2')} = \frac{2^{2+(p-2)/2}}{3}V_{p,GLM}^2,$$

*with $V_{p,GLM}^p = e^{a_{p,GLM}c_\gamma^2 C_{S,\sigma}^2 \frac{2\gamma}{2\gamma - 1}} \max\{1, \mathbb{E}\left[V_0^p\right]\}$ where*

$$a_{p,GLM} := p\left(\frac{2C_{GLM}^{(2)}}{\sigma} + L_\sigma^2\right) + 2^{p-2}(p-1)pC_{GLM}^2\left(c_\gamma^2 C_{S,\sigma}^2\left(8L_\sigma^4 + \frac{32C_{GLM}^{(4)}}{\sigma^2}\right) + \frac{4L_\sigma^2}{\sigma} + \frac{8C_{GLM}^{(2)}}{\sigma^2}\right)$$
$$+ 2^{p-2}(p-1)pC_{GLM}^p\left(c_\gamma^{2p-2}C_{S,\sigma}^{2p-2}\left(2^{2p-1}L_\sigma^{2p} + \frac{2^{3p-1}C_{GLM}^{(2p)}}{\sigma^2}\right) + c_\gamma^{p-2}C_{S,\sigma}^{p-2}\left(2^{2p-2}L_\sigma^{2p} + \frac{2p}{\sigma^2}\left(\frac{1}{2} + 2^{p-1/2}\sqrt{C_{GLM}^{(2p)}}\right)\right)\right).$$

Remark that for $p > 2$, $v_{n,GLM} = O(n^{-\min(1,2\gamma)})$.

**Verifying Assumption (H3).** We prove here that **(H3)** holds for general linear models. We now denote

$$H_\sigma =: \mathbb{E}\left[\nabla_h^2 \ell\left(Y, \theta_\sigma^T X\right) X X^T\right] + \sigma I_d.$$

**Proposition 7.7.** *Suppose Assumptions **(GLM1)** and **(GLM2)** hold, then for all $n \geq 0$,*

$$\mathbb{E}\left[\left\|\bar{S}_n^{-1} - H_\sigma^{-1}\right\|^2\right] \leq \frac{4C_{S,\sigma}^2}{\sigma^2 n}\left(L_{\nabla l}^2 \mathbb{E}\left[\|X\|^4\right] + \frac{L_{\nabla^2 L}^2}{\sigma}\sum_{i=0}^{n-1} v_{i,GLM} + \frac{1}{n}\|S_0 - H(\theta_\sigma)\|^2\right) + \frac{16d^4 C_{S,\sigma}^2}{n^2} =: v_{\ell,n}$$

*with $v_{i,GLM}$ defined in Proposition 7.6.*

We can now finish the proof of Theorem 5.1. In this aim, let us first remark that for all $h, h'$,

$$\mathbb{E}\left[\left\|\nabla_h \ell\left(y, X^T h\right) X + \sigma h - \nabla_h \ell\left(y, X^T h'\right) X - \sigma h'\right\|^2\right] \leq 2\left(L_{\nabla l}^2 \mathbb{E}\left[\|X\|^2\right] + \sigma^2\right)\|h - h'\|^2.$$

Then, with the help of Theorem 3.3, one has

$$\mathbb{E}\left[\|\theta_n - \theta_\sigma\|^2\right] \leq e^{-\frac{1}{2}c_\gamma n^{1-\gamma}}\left(K_{1,\mathrm{GLM}}^{(3)} + K_{1',\mathrm{GLM}}^{(3)}\max_{0 \leq k \leq n}(k+1)^\gamma d_{k,\mathrm{GLM}}\right)$$

$$+ n^{-\gamma}\left(2^{3+\gamma}c_\gamma \mathrm{Tr}\left(H_\sigma^{-1}\Sigma_\sigma H_\sigma^{-1}\right) + \frac{K_{2,\mathrm{GLM}}^{(3)}}{n^\gamma} + K_{2',\mathrm{GLM}}^{(3)}v_{l,n/2}\right) + d_{\lfloor n/2\rfloor,\mathrm{GLM}},$$

with $\Sigma_\sigma := \mathbb{E}\left[\left(\nabla_h \ell\left(y, X^T \theta_\sigma\right) X + \sigma \theta_\sigma\right)\left(\nabla_h \ell\left(y, X^T \theta_\sigma\right) X + \sigma \theta_\sigma\right)^T\right]$ and since $c_\gamma 4 \frac{C_{\mathrm{GLM}}^{(2)}}{\sigma^2} \geq C_{A,\mathrm{GLM}} =\geq 4c_\gamma$,

$$K_{1,\mathrm{GLM}}^{(3)} = e^{8\frac{C_{\mathrm{GLM}}^{(2)}}{\sigma^2}c_\gamma^3 \frac{2\gamma}{2\gamma-1}}\left(\mathbb{E}\left[\|\theta_0 - \theta_\sigma\|^2\right] + \frac{2\mathrm{Tr}\left(H_\sigma^{-1}\Sigma_\sigma H_\sigma^{-1}\right)}{c_\gamma} + 8\sigma^2\left(\sigma^{-4} + C_{S,\sigma}^4\right) + \frac{2L_\sigma^2 v_{l,0}}{c_\gamma}\right), \quad (60)$$

$$K_{1',\mathrm{GLM}}^{(3)} = \frac{1}{4c_\gamma}e^{8\frac{C_{\mathrm{GLM}}^{(2)}}{\sigma^2}c_\gamma^3 \frac{2\gamma}{2\gamma-1}}, \quad d_{n,\mathrm{GLM}} = 8C_{\mathrm{GLM}}\sqrt{v_{n,\mathrm{GLM}}v_{l,n}} + 8\frac{L_{\nabla^2 L}^2\sigma^{-2} + 2C_{\mathrm{GLM}}^{(2)}}{\sigma^2}v_{n,\mathrm{GLM}}, \quad (61)$$

$$K_{2,\mathrm{GLM}}^{(3)} = 2^{5+2\gamma}C_{\mathrm{GLM}}^{(2)}c_\gamma\left(\sigma^{-4} + C_{S,\sigma}^4\right)c_\gamma^2, \quad K_{2',\mathrm{GLM}}^{(3)} = 2^{3+\gamma}L_\sigma^2 c_\gamma. \quad (62)$$

*Proof of Theorem 5.2.* The proof follows exactly the same pattern as the proof of Theorem 4.2, using Assumption **(A6)** together with Lemma 7.4 to compute the constant $C_S$ such that **(H2)** is satisfied. □

# A Proofs of technical proposition

## A.1 Proof of Proposition 3.1

Let us recall that

$$V_{n+1} = V_n \underbrace{-\gamma_{n+1}\left(g_{n+1}'\right)^T A_n \int_0^1 \nabla G\left(\theta_n + t\left(\theta_{n+1} - \theta_n\right)\right)dt}_{=:U_{n+1}}$$

Remark that for $a \geq 2$ and $x, h \in \mathbb{R}$ such that $x \geq 0$ and $x + h \geq 0$, we have by Taylor's expansion

$$(x + h)^a \leq x^a + ax^{a-1}h + 2^{a-2}a(a-1)(x^{a-2}|h|^2 + |h|^a). \quad (63)$$

This yields for $a = p'$, $x = V_n$ and $h = U_{n+1}$ and after conditioning on $\mathcal{F}_n$

$$\mathbb{E}\left[V_{n+1}^{p'}|\mathcal{F}_n\right] \leq V_n^{p'} + p'V_n^{p'-1}\mathbb{E}\left[U_{n+1}|\mathcal{F}_n\right]$$

$$+ 2^{p'-2}p'(p'-1)\left(\mathbb{E}\left[|U_{n+1}|^2|\mathcal{F}_n\right]V_n^{p'-2} + \mathbb{E}\left[|U_{n+1}|^{p'}|\mathcal{F}_n\right]\right). \quad (64)$$

Since $G$ is convex and $\nabla G$ is Lipschitz,

$$
\begin{aligned}
\mathbb{E}\left[U_{n+1}V_n^{p'-1}|\mathcal{F}_n\right] \leq &-\mathbb{E}\left[\gamma_{n+1}\left(g'_{n+1}\right)^T A_n \int_0^1 \nabla G\left(\theta_n\right)dt|\mathcal{F}_n\right]V_n^{p'-1}\\
&+\mathbb{E}\left[\gamma_{n+1}\left(g'_{n+1}\right)^T A_n \int_0^1 \left(\nabla G\left(\theta_n\right)-\nabla G\left(\theta_n+t\left(\theta_{n+1}-\theta_n\right)\right)\right)dt|\mathcal{F}_n\right]V_n^{p'-1}\\
\leq &-\gamma_{n+1}\nabla G\left(\theta_n\right)^T A_n \nabla G\left(\theta_n\right)V_n^{p'-1}+\frac{L_{\nabla G}}{2}\gamma_{n+1}^2\mathbb{E}\left[\left\|g'_{n+1}\right\|^2|\mathcal{F}_n\right]\|A_n\|^2 V_n^{p'-1}.\\
\leq &-\gamma_{n+1}\nabla G\left(\theta_n\right)^T A_n \nabla G\left(\theta_n\right)V_n^{p'-1}+\gamma_{n+1}^2\beta_{n+1}^2\frac{L_{\nabla G}}{2}\left(C_1 V_n^{p'-1}+\frac{2C_2}{\mu}V_n^{p'}\right).
\end{aligned}
$$

By strong convexity, we have

$$
\begin{aligned}
\nabla G\left(\theta_n\right)^T A_n \nabla G\left(\theta_n\right)V_n^{p'-1} &\geq \lambda_{\min}\left(A_n\right)\left\|\nabla G\left(\theta_n\right)\right\|^2 V_n^{p'-1}\\
&\geq 2\lambda_n\mu V_n^{p'}\mathbf{1}_{\lambda_{\min}(A_n)\geq\lambda_n}\\
&= 2\lambda_n\mu V_n^{p'}-2\mathbf{1}_{\lambda_{\min}(A_n)<\lambda_n}\lambda_n\mu V_n^{p'},
\end{aligned}
$$

where $\lambda_n=\lambda_0(n+1)^\lambda$ with $0\leq\lambda<\min\{\gamma-2\beta,1-\gamma\}$. Applying Hölder inequality yields then

$$
\begin{aligned}
\mathbb{E}\left[\nabla G\left(\theta_n\right)^T A_n \nabla G\left(\theta_n\right)\right] \geq &2\lambda_n\mu\mathbb{E}\left[V_n^{p'}\right]-2\lambda_n\mu\mathbb{E}[V_n^p]^{p'/p}\left(\mathbb{P}\left[\lambda_{\min}(A_n)<\lambda_n\right]\right)^{\frac{p-p'}{p}}\\
\geq &2\lambda_n\mu\mathbb{E}\left[V_n^{p'}\right]-2\lambda_n\mu V_p^{p'}\left(\mathbb{P}\left[\lambda_{\min}(A_n)<\lambda_n\right]\right)^{\frac{p-p'}{p}},
\end{aligned}
$$

with $V_p^p\geq\sup_{n\geq0}\mathbb{E}[V_n^p]$ given by Lemma 7.1. Then, Assumption **(H1a)** gives $\mathbb{P}\left[\lambda_{\min}\left(A_n\right)<\lambda_n\right]\leq v_{n+1}(n+1)^{-\delta-q\lambda}:=\bar{v}_n$, so that finally

$$
\begin{aligned}
&\mathbb{E}\left[U_{n+1}V_n^{p'-1}\right]\\
&\leq -2\gamma_{n+1}\lambda_n\mathbb{E}\left[\mu V_n^{p'}\right]+2\lambda_n\gamma_{n+1}\mu V_p^{p'}\bar{v}_n^{\frac{p-p'}{p}}+\gamma_{n+1}^2\beta_{n+1}^2\frac{L_{\nabla G}}{2}\left(C_1\mathbb{E}\left[V_n^{p'-1}\right]+\frac{2C_2}{\mu}\mathbb{E}\left[V_n^{p'}\right]\right). \quad (65)
\end{aligned}
$$

Furthermore, since $\nabla G$ is $L_{\nabla G}$-Lipschitz, one has

$$
\begin{aligned}
\left\|\int_0^1 \nabla G\left(\theta_n+t\left(\theta_{n+1}-\theta_n\right)\right)dt\right\| &\leq L_{\nabla G}\int_0^1\left(\left\|\theta_n-\theta\right\|+t\left\|\theta_{n+1}-\theta_n\right\|\right)dt\\
&\leq L_{\nabla G}\left(\left\|\theta_n-\theta\right\|+\frac{1}{2}\gamma_{n+1}\left\|A_n\right\|\left\|g'_{n+1}\right\|\right). \quad (66)
\end{aligned}
$$

Hence, using **(H1b)** and the strong convexity of $G$ yields

$$
\begin{aligned}
\mathbb{E}\left[\left|U_{n+1}\right|^{p'}|\mathcal{F}_n\right] \leq &L_{\nabla G}^{p'}\left\|A_n\right\|^{p'}\gamma_{n+1}^{p'}\mathbb{E}\left[\left\|g'_{n+1}\right\|^{p'}\left(2^{p'-1}\left\|\theta_n-\theta\right\|^{p'}+2^{-1}\gamma_{n+1}^{p'}\left\|A_n\right\|^{p'}\left\|g'_{n+1}\right\|^{p'}\right)|\mathcal{F}_n\right]\\
\leq &\frac{L_{\nabla G}^{p'}}{2}\gamma_{n+1}^{p'}\beta_{n+1}^{p'}\left(2^{p'}\left(C_1^{(p'/2)}\frac{2^{p'/2}V_n^{p'/2}}{\mu^{p'/2}}+C_2^{(p'/2)}\frac{2^{p'}V_n^{p'}}{\mu^{p'}}\right)\right.\\
&\left.+\gamma_{n+1}^{p'}\beta_{n+1}^{p'}\left(C_1^{(p')}+C_2^{(p')}\frac{2^{p'}V_n^{p'}}{\mu^{p'}}\right)\right)
\end{aligned}
$$

Specializing the latter inequality with $p'=2$ yields then (recalling inequalities equation 12)

$$
\begin{aligned}
&\mathbb{E}\left[\left|U_{n+1}\right|^2|\mathcal{F}_n\right]V_n^{p'-2}\\
&\leq \frac{L_{\nabla G}^2}{2}\gamma_{n+1}^2\beta_{n+1}^2\left(2^{p'}\left(C_1\frac{2V_n}{\mu}+C_2\frac{2^2V_n^2}{\mu^2}\right)+\gamma_{n+1}^2\beta_{n+1}^2\left(C_1'+C_2'\frac{4V_n^2}{\mu^2}\right)\right)V_n^{p'-2},
\end{aligned}
$$

so that

$$\mathbb{E}\left[\left|U_{n+1}\right|^{p'}|\mathcal{F}_n\right] + \mathbb{E}\left[\left|U_{n+1}\right|^2|\mathcal{F}_n\right]V_n^{p'-2}$$

$$\leq \frac{L_{\nabla G}^{p'}C_1^{(p')}}{2}\gamma_{n+1}^{2p'}\beta_{n+1}^{2p'} + \frac{2^{3p'/2-1}L_{\nabla G}^{p'}C_1^{(p'/2)}}{\mu^{p'/2}}\gamma_{n+1}^{p'}\beta_{n+1}^{p'}V_n^{p'/2} + \frac{2^{p'}L_{\nabla G}^2C_1}{\mu}\gamma_{n+1}^2\beta_{n+1}^2V_n^{p'-1}$$

$$+ \frac{L_{\nabla G}^2C_1'}{2}\gamma_{n+1}^4\beta_{n+1}^4V_n^{p'-2} + V_n^{p'}\left(\frac{2^{2p'-1}L_{\nabla G}^{p'}C_2^{(p'/2)}}{\mu^{p'}}\gamma_{n+1}^{p'}\beta_{n+1}^{p'} + \frac{2^{p'-1}L_{\nabla G}^{p'}C_2^{(p')}}{\mu^{p'}}\gamma_{n+1}^{2p'}\beta_{n+1}^{2p'}\right.$$

$$\left. + \frac{2^{p'+1}C_2L_{\nabla G}^2}{\mu^2}\gamma_{n+1}^2\beta_{n+1}^2 + \frac{2C_2'L_{\nabla G}^2}{\mu^2}\gamma_{n+1}^4\beta_{n+1}^4\right).$$

Using the latter inequality with equation 65 in equation 64 yields then

$$\mathbb{E}\left[V_{n+1}^{p'}\right] \leq \mathbb{E}\left[V_n^{p'}\right] - 2p'\mu\gamma_{n+1}\lambda_n\mathbb{E}\left[V_n^{p'}\right] + 2p'\lambda_n\gamma_{n+1}\mu V_p^{p'}\bar{v}_n^{\frac{p-p'}{p}} + \mathbb{E}\left[P\left(\gamma_{n+1}^2\beta_{n+1}^2, V_n\right)\right]$$

with $P(x,y) = A_0x^{p'} + A_{p'/2}x^{p'/2}y^{p'/2} + A_{p'-1}xy^{p'-1} + A_{p'-2}x^2y^{p'-2} + A_{p'}xy^{p'}$, where

$$A_0 = 2^{p'-3}p'(p'-1)L_{\nabla G}^{p'}C_1^{(p')}, \ A_{p'/2} = \frac{2^{5p'/2-3}p'(p'-1)L_{\nabla G}^{p'}C_1^{(p'/2)}}{\mu^{p'/2}},$$

$$A_{p'-1} = p'\frac{L_{\nabla G}}{2} + \frac{2^{2p'-2}p'(p'-1)L_{\nabla G}^2C_1}{\mu}, \ A_{p'-2} = 2^{p'-3}p'(p'-1)L_{\nabla G}^2C_1',$$

and

$$A_{p'} = \frac{p'L_{\nabla G}C_2}{\mu} + p'(p'-1)\left(\frac{2^{3p'-3}L_{\nabla G}^{p'}C_2^{(p'/2)}}{\mu^{p'}}c_\gamma^{p'-2}c_\beta^{p'-2} + \frac{2^{2p'-3}L_{\nabla G}^{p'}C_2^{(p')}}{\mu^{p'}}c_\gamma^{2p'-2}c_\beta^{2p'-2}\right.$$

$$\left. + \frac{2^{2p'-1}C_2L_{\nabla G}^2}{\mu^2} + \frac{2^{p'-1}C_2'L_{\nabla G}^2}{\mu^2}c_\gamma^2c_\beta^2\right).$$

Applying now Young's inequality, which implies $a^ib^{p'-i} \leq \frac{ia^{p'}}{p'} + \frac{(p'-i)b^{p'}}{p'}$ for $0 < i < p'$ and $a, b \geq 0$, yields for any $t > 0$ and $i \in \{1, 2, p'/2\}$

$$A_{p-i}x^iy^{p'-i} = \left(\frac{A_i^{1/i}x}{(t\lambda_n\gamma_n)^{\frac{p'-i}{p'}}}\right)^i\left((t\lambda_n\gamma_n)^{\frac{i}{p'}}y\right)^{p'-i} \leq \frac{iA_i^{\frac{p'}{i}}x^{p'}}{(t\lambda_n\gamma_n)^{\frac{p'-i}{i}}} + \frac{(p'-i)t\lambda_n\gamma_ny^{p'}}{p'},$$

so that using the latter inequality with $t = \frac{p'^2\mu}{3(p'-i)\mu}$ for $i \in \{1, 2, p'/2\}$ and using that

$$\frac{\gamma_{n+1}^{2p'}\beta_n^{2p'}}{(\gamma_{n+1}\lambda)^{\frac{p'}{i}-1}} = (\gamma_{n+1}\lambda_n)c_\gamma^{\frac{2i-1}{i}p'}c_\beta^{2p'}\lambda_0^{-\frac{p'}{i}}(n+1)^{-\frac{(2i-1)p'}{i}\gamma+2p'\beta+\frac{p'}{i}\lambda}$$

$$\leq (\gamma_{n+1}\lambda_n)c_\gamma^{\frac{2i-1}{i}p'}c_\beta^{2p'}\lambda_0^{-\frac{p'}{i}}(n+1)^{-p'(\gamma-2\beta-\lambda)}$$

gives

$$\mathbb{E}\left[P\left(\gamma_{n+1}^2\beta_{n+1}^2, V_n\right)\right] \leq L\left(\frac{p'\mu}{2}\lambda_n\gamma_{n+1}\right)(n+1)^{-p'(\gamma-2\beta-\lambda)} + \left(p'\mu(\lambda_n\gamma_{n+1}) + A_{p'}(\gamma_{n+1}\beta_{n+1})^2\right)\mathbb{E}\left[V_n^{p'}\right]$$

with

$$L = \frac{c_\gamma^{2p'-1}c_\beta^{2p'}}{\lambda_0}A_0 + \frac{3c_\gamma^{2(p'-1)}c_\beta^{2p'}\lambda_0^{-2}\sqrt{A_{p'}}}{4\mu}A_{p'/2} + \frac{2c_\gamma^{\frac{3}{2}p'}c_\beta^{2p'}\lambda_0^{-\frac{p'}{2}}}{\left(\frac{p'^2\mu}{3(p'-2)}\right)^{\frac{p'-2}{2}}}A_2^{p'/2} + \frac{c_\gamma^{p'}c_\beta^{2p'}\lambda_0^{-p'}}{\left(\frac{p'^2\mu}{3(p'-1)}\right)^{p'-1}}A_1.$$

Putting together the previous inequalities and taking the expectation yield then

$$\mathbb{E}\left[V_{n+1}^{p'}\right] \leq \left(1 - p'\mu\gamma_{n+1}\lambda_n + \frac{A_{p'}c_\gamma c_\beta^2}{\lambda_0}(n+1)^{-\gamma+2\beta+\lambda}\gamma_{n+1}\lambda_n\right)\mathbb{E}\left[V_n^{p'}\right]$$
$$+ \lambda_n\gamma_{n+1}\left(2p'\mu V_p^{p'}\bar{v}_n^{\frac{p-p'}{p}} + \frac{Lp'\mu}{2}(n+1)^{-p'(\gamma-2\beta-\lambda)}\right).$$

Then, recalling that $\bar{v}_n = v_{n+1}(n+1)^{-\delta-q\lambda}$ and using Proposition 7.1 yields

$$\mathbb{E}\left[V_n^{p'}\right] \leq \exp\left(-\frac{c_\gamma p'\mu\lambda_0}{2}n^{1-(\lambda+\gamma)}(1-\varepsilon(n))\right)\left(K_1^{(1')} + K_{1'}^{(1')}\max_{1\leq k\leq n+1}k^{\gamma-2\beta-\lambda-\frac{p-p'}{p}(\delta+q\lambda)}v_k^{\frac{p-p'}{p}}\right)$$
$$+ K_2^{(1')}n^{-p'(\gamma-2\beta-\lambda)} + K_3^{(1')}v_{\lfloor n/2\rfloor}^{\frac{p-p'}{p}}(n+1)^{-\frac{p-p'}{p}(\delta+q\lambda)},$$

with

$$\varepsilon(n) = \frac{4C_M'n^{-1+\lambda+\gamma}}{\mu p'\lambda_0}\left(1 + \frac{n^{(1+2\beta-2\gamma)^+}}{|2\gamma-2\beta-1|}\right), \tag{67}$$

and

$$K_1^{(1')} = \left(\mathbb{E}\left[V_0\right] + \frac{p'\mu L}{C_M'}\right), \quad K_{1'}^{(1')} = \frac{4p'\mu V_p^{p'}}{C_M'}, \tag{68}$$

where

$$C_M' = \max\left\{\frac{A_{p'}c_\gamma c_\beta^2}{\lambda_0}, \left(\frac{\mu p'\lambda_0}{8}\right)^{\frac{2\gamma-2\beta}{\gamma+\lambda}}c_\gamma^{\frac{\gamma-2\beta-\lambda}{\gamma+\lambda}}\right\}, \tag{69}$$

and

$$K_2^{(1')} = 2^{p'(\gamma-2\beta-\lambda)}L, \; K_3^{(1')} = 2^{2+\frac{p-p'}{p}(\delta+q\lambda)}V_p^{p'}. \tag{70}$$

where $V_p$ is given in Lemma 7.1.

### A.2   Proof of Proposition 3.2

Remark that with the help of a Taylor's expansion of $G$, one has

$$V_{n+1} = V_n + (\theta_{n+1} - \theta_n)^T\int_0^1\nabla G\left(\theta_n + t\left(\theta_{n+1} - \theta_n\right)\right)dt$$
$$= V_n - \gamma_{n+1}\left(g_{n+1}'\right)^T A_n\int_0^1\nabla G\left(\theta_n + t\left(\theta_{n+1} - \theta_n\right)\right)dt.$$

Then, using equation 66 one has

$$V_{n+1}^2 \leq V_n^2 \overbrace{- 2\gamma_{n+1}V_n\left(g_{n+1}'\right)^T A_n\int_0^1\nabla G\left(\theta_n + t\left(\theta_{n+1} - \theta_n\right)\right)dt}^{:=(\star)}$$
$$+ \underbrace{L_{\nabla G}^2\left\|A_n\right\|^2\left\|g_{n+1}'\right\|^2\gamma_{n+1}^2\left(2\left\|\theta_n - \theta\right\|^2 + \frac{1}{2}\gamma_{n+1}^2\left\|A_n\right\|^2\left\|g_{n+1}'\right\|^2\right)}_{:=(\star\star)}$$

We now bound $(\star)$ and $(\star\star)$. First, thanks to Assumption **(H1)** and since $\|\theta_n - \theta\|^2 \leq \frac{2}{\mu} V_n$, one has

$$
\mathbb{E}\left[(\star\star)|\mathcal{F}_n\right] \leq \frac{4L_{\nabla G}^2 C_1}{\mu}\gamma_{n+1}^2 \|A_n\|^2 V_n + \frac{8L_{\nabla G}^2 C_2}{\mu^2}\|A_n\|^2 \gamma_{n+1}^2 V_n^2
$$
$$
+ \frac{1}{2}L_{\nabla G}^2 C_1' \gamma_{n+1}^4 \|A_n\|^4 + \frac{2L_{\nabla G}^2 C_2'}{\mu^2}\gamma_{n+1}^4 \|A_n\|^4 V_n^2
$$
$$
\leq \frac{8L_{\nabla G}^4 C_1^2}{\mu^3 \lambda_0}\gamma_{n+1}^3 \|A_n\|^4 + \frac{1}{2}\mu\lambda_0 \gamma_{n+1} V_n^2 + \frac{L_{\nabla G}^2 C_1'}{2}\gamma_{n+1}^4 \|A_n\|^4
$$
$$
+ \frac{2L_{\nabla G}^2}{\mu^2}\left(4C_2 + C_2' c_\gamma^2 c_\beta^2\right)\gamma_{n+1}^2 \beta_{n+1}^2 V_n^2.
$$

Then, taking the expectation with Assumption **(H2b)**,

$$
\mathbb{E}\left[(\star\star)\right] \leq \frac{8L_{\nabla G}^4 C_1^2}{\mu^3 \lambda_0}\gamma_{n+1}^3 C_S^4 + \frac{\mu\lambda_0}{2}\gamma_{n+1}\mathbb{E}\left[V_n^2\right] + \frac{L_{\nabla G}^2 C_1'}{2}\gamma_{n+1}^4 C_S^4
$$
$$
+ \frac{2L_{\nabla G}^2}{\mu^2}\left(4C_2 + C_2' c_\gamma^2 c_\beta^2\right)\gamma_{n+1}^2 \beta_{n+1}^2 \mathbb{E}\left[V_n^2\right].
$$

Moreover, since $\nabla G$ is $L_{\nabla G}$-Lipschitz, one can check that

$$
\left\|\int_0^1 \nabla G\left(\theta_n + t\left(\theta_{n+1} - \theta_n\right)\right) - \nabla G\left(\theta_n\right)dt\right\| \leq L_{\nabla G}\int_0^1 tdt\gamma_{n+1}\|A_n\|\left\|g_{n+1}'\right\|
$$
$$
\leq \frac{L_{\nabla G}}{2}\gamma_{n+1}\|A_n\|\left\|g_{n+1}'\right\|.
$$

Then, one has

$$
\mathbb{E}\left[(\star)|\mathcal{F}_n\right] \geq 2\gamma_{n+1}\nabla G\left(\theta_n\right)^T A_n \nabla G\left(\theta_n\right)V_n - L_{\nabla G}\gamma_{n+1}^2 \|A_n\|^2 \mathbb{E}\left[\left\|g_{n+1}'\right\|^2 |\mathcal{F}_n\right]V_n
$$
$$
\geq 2\gamma_{n+1}\nabla G\left(\theta_n\right)^T A_n \nabla G\left(\theta_n\right)V_n - L_{\nabla G}\gamma_{n+1}^2 \|A_n\|^2 C_1 V_n - \frac{2L_{\nabla G}C_2}{\mu}\gamma_{n+1}^2 \|A_n\|^2 V_n^2
$$
$$
\geq 2\gamma_{n+1}\nabla G\left(\theta_n\right)^T A_n \nabla G\left(\theta_n\right)V_n - \frac{C_1^2 L_{\nabla G}^2}{2\mu\lambda_0}\gamma_{n+1}^3 \|A_n\|^4 - \frac{\mu\lambda_0 \gamma_{n+1}}{2}V_n^2 - \frac{2L_{\nabla G}C_2}{\mu}\gamma_{n+1}^2 \beta_{n+1}^2 V_n^2.
$$

Furtermore, with the help of inequality equation 15 it comes

$$
\gamma_{n+1}\nabla G\left(\theta_n\right)^T A_n \nabla G\left(\theta_n\right)V_n \geq 2\lambda_0 \mu\gamma_{n+1}V_n^2 - 2\lambda_0 \mu\gamma_{n+1}\mathbf{1}_{A_n < \lambda_0}V_n^2.
$$

Then, with the help of Holder's inequality, coupled with **(H1a)** for $t = 1$, one has

$$
\mathbb{E}\left[(\star)\right] \geq \frac{7}{2}\lambda_0 \mu\gamma_{n+1}V_n^2 - 4\lambda_0 \mu\gamma_{n+1}\bar{v}_n^{(p-2)/p}V_p^2 - \frac{C_1^2 L_{\nabla G}^2}{2\mu\lambda_0}\gamma_{n+1}^3 C_S^4 - \frac{2L_{\nabla G}C_2}{\mu}\gamma_{n+1}^2 \beta_{n+1}^2 \mathbb{E}\left[V_n^2\right]
$$

with $V_p$ defined in Lemma 7.1 and $\bar{v}_n := v_n(n+1)^{-\delta}$ is the upper bound from **(H1a)** on $\mathbb{P}\left[\lambda_{\min}\left(A_n\right) \leq \lambda_0\right]$. Let

$$
a_M := \max\left\{\left(\frac{2L_{\nabla G}C_2}{\mu} + \frac{2L_{\nabla G}^2}{\mu^2}\left(4C_2 + C_2' c_\gamma^2 c_\beta^2\right)\right)c_\gamma c_\beta^2, \left(\frac{3\lambda_0 \mu}{2}\right)^{\frac{2\gamma-2\beta}{\gamma}}c_\gamma^{\frac{\gamma-2\beta}{\gamma}}\right\}, \tag{71}
$$

one has

$$
\mathbb{E}\left[V_{n+1}^2\right] \leq \left(1 - 3\lambda_0 \mu\gamma_{n+1} + a_M n^{2\beta-\gamma}\gamma_{n+1}\right)\mathbb{E}\left[V_n\right] + 4\lambda_0 \mu\gamma_{n+1}\bar{v}_n^{(p-2)/p}V_p^2
$$
$$
+ \underbrace{C_S^4 L_{\nabla G}^2 \left(\frac{8L_{\nabla G}^4 C_1^2}{\mu^3 \lambda_0} + \frac{C_1' c_\gamma}{2} + \frac{C_1^2}{2\mu\lambda_0}\right)}_{=:a_1}\gamma_{n+1}^3 \tag{72}
$$

Applying Proposition 7.1, it comes (with analogous calculus to the ones in the proof of Theorem 3.1)

$$\mathbb{E}\left[V_n^2\right] \leq \exp\left(-\frac{3}{2}c_\gamma\lambda_0\mu n^{1-\gamma}\right)\exp\left(2a_M\frac{2\gamma-2\beta}{2\gamma-2\beta-1}\right)$$
$$\cdot\left(\mathbb{E}\left[V_0^2\right] + \frac{2a_1c_\gamma^2}{a_M} + \frac{8\lambda_0\mu c_\gamma V_p^{2/p}}{a_M}\max_{1\leq k\leq n+1}v_k^{\frac{p-2}{p}}k^{\gamma-\frac{(p-2)}{p}\delta}\right) + \frac{2^{2\gamma}a_1c_\gamma^2}{3\lambda_0\mu}n^{-2\gamma} + \frac{4}{3}V_p^2\bar{v}_{\lfloor n/2\rfloor}^{(p-2)/p}.$$

where $V_p$ is given by Lemma 7.1 and $\bar{v}_{\lfloor n/2\rfloor} \leq v_{n/2}2^\delta(n+1)^{-\delta}$. Setting

$$K_1^{(2')} = \exp\left(2a_M\frac{2\gamma-2\beta}{2\gamma-2\beta-1}\right)\left(\mathbb{E}\left[V_0^2\right] + \frac{2a_1c_\gamma^2}{a_M}\right), \tag{73}$$

$$K_{1'}^{(2')} = \exp\left(2a_M\frac{2\gamma-2\beta}{2\gamma-2\beta-1}\right)\cdot\frac{8\lambda_0\mu V_p^2}{a_M}, \tag{74}$$

with $a_M$ given in equation 71, $a_1$ given in equation 72 and $V_p$ given in Lemma 7.1, and

$$K_2^{(2')} = \frac{2^{2\gamma}a_1c_\gamma^2}{3\lambda_0\mu}, \quad K_3^{(2')} = \frac{2^{2+(p-2)\delta/p}}{3}V_p^2, \tag{75}$$

we finally get

$$\mathbb{E}\left[V_n^2\right] \leq \exp\left(-\frac{3}{2}c_\gamma\lambda_0\mu n^{1-\gamma}\right)\left(K_1^{(2')} + K_{1'}^{(2')}\max_{1\leq k\leq n+1}v_k^{\frac{p-2}{p}}k^{\gamma-\delta\frac{p-2}{p}}\right)$$
$$+ K_2^{(2')}n^{-2\gamma} + K_3^{(2')}v_{\lfloor n/2\rfloor}^{(p-2)/p}n^{-\delta(p-2)/p} =: M_n.$$

Then, for any $0 \leq \gamma' \leq \min\left\{2\gamma, \frac{\delta(p-2)}{p}\right\}$, only depending on $v_n$ and $\gamma$, we have

$$w_\infty(\gamma') := \sup_{n\geq 1}M_n n^{\gamma'} < +\infty. \tag{76}$$

The function $w_\infty : \left[0, \min\left\{2\gamma, \frac{\delta(p-2)}{p}\right\}\right] \to \mathbb{R}$ can be computed numerically, but in any case note that $w_\infty(\gamma') \leq K_1^{(2')}\sup_{t\geq 1}\left\{t^{\gamma'}\exp\left(-\frac{1}{2}\lambda_0\mu t^{1-\gamma}\right)\right\} + K_2^{(2')} + K_3^{(2')}$, so that a function analysis yields, for $\gamma' \in \left[0, \min\left\{2\gamma, \frac{\delta(p-2)}{p}\right\}\right]$,

$$w_\infty(\gamma') \leq K_1^{(2')}\left(\frac{2\gamma'}{\lambda_0\mu e(1-\gamma)}\right)^{\frac{\gamma'}{1-\gamma}} + K_2^{(2')} + K_3^{(2')}. \tag{77}$$

We will see in most applications that under suitable assumptions, $\gamma'$ can be equal to $2\gamma$ (namely when $\delta \geq \frac{2p}{p-2}\gamma$).

## B  Proofs of technical lemmas

### B.1  Proof of Lemma 7.1

Observe that since the proofs are analogous, we only make the proof for $p > 2$, and for the case where $p = 2$, if there are some differences in the proof, it will be indicated with the help of remarks.

With the help of a Taylor expansion of the functional $G$, one has

$$V_{n+1} = V_n - \gamma_{n+1}\left(g'_{n+1}\right)^T A_n\int_0^1 \nabla G\left(\theta_n + t\left(\theta_{n+1} - \theta_n\right)\right)dt.$$

Then, applying the inequality

$$(a+h)^p \le a^p + pa^{p-1}h + \frac{p(p-1)h^2}{2}\max(1, 2^{p-3})(a^{p-2} + |h|^{p-2})$$
$$\le a^p + pa^{p-1}h + p(p-1)2^{p-3}h^2(a^{p-2} + |h|^{p-2})$$

for $a, a+h \ge 0$ to $a = V_n$ and $h = -\gamma_{n+1}\left(g'_{n+1}\right)^T A_n \int_0^1 \nabla G\left(\theta_n + t\left(\theta_{n+1} - \theta_n\right)\right) dt$, one has

$$V_{n+1}^p \le V_n^p - p\gamma_{n+1}\left(g'_{n+1}\right)^T A_n \int_0^1 \nabla G\left(\theta_n + t\left(\theta_{n+1} - \theta_n\right)\right) dt V_n^{p-1}$$

$$+ 2^{p-3}p(p-1)\left\|\gamma_{n+1}\left(g'_{n+1}\right)^T A_n \int_0^1 \nabla G\left(\theta_n + t\left(\theta_{n+1} - \theta_n\right)\right) dt\right\|^2 V_n^{p-2}$$

$$+ 2^{p-3}p(p-1)\left\|\gamma_{n+1}\left(g'_{n+1}\right)^T A_n \int_0^1 \nabla G\left(\theta_n + t\left(\theta_{n+1} - \theta_n\right)\right) dt\right\|^p$$

**Remark B.1.** *Observe that in the case where $p = 2$, one has*

$$(a+h)^2 = a^2 + 2ah + h^2 = a^p + 2a^{p-1}h + p(p-1)2^{p-3}h^2|h|^{p-2}$$

*the last term on the right hand-side of previous inequality can be considered equal to $0$.*

Recalling that since $\nabla G$ is $L_{\nabla G}$-Lipschitz, one has

$$\left\|\int_0^1 \nabla G\left(\theta_n + t\left(\theta_{n+1} - \theta_n\right)\right) dt\right\| \le L_{\nabla G}\left(\|\theta_n - \theta\| + \gamma_{n+1}\|A_n\|\left\|g'_{n+1}\right\|\right),$$

which implies

$$\left.\begin{aligned}V_{n+1}^p &\le V_n^p - p\gamma_{n+1}\left(g'_{n+1}\right)^T A_n \int_0^1 \nabla G\left(\theta_n + t\left(\theta_{n+1} - \theta_n\right)\right) dt V_n^{p-1}\end{aligned}\right\} =: (*)$$

$$+ 2^{p-2}p(p-1)L_{\nabla G}^2\gamma_{n+1}^2\left\|g'_{n+1}\right\|^2\|A_n\|^2\left(\|\theta_n - \theta\|^2 + \gamma_{n+1}^2\|A_n\|^2\left\|g'_{n+1}\right\|^2\right)V_n^{p-2}\Bigg\} =: (**)$$

$$+ 2^{p-2}p(p-1)L_{\nabla G}^p\gamma_{n+1}^p\left\|g'_{n+1}\right\|^p\|A_n\|^p\left(\|\theta_n - \theta\|^p + \gamma_{n+1}^p\|A_n\|^p\left\|g'_{n+1}\right\|^p\right)\Bigg\} =: (***)$$

Furthermore, one has

$$(*) = -p\gamma_{n+1}\left(g'_{n+1}\right)^T A_n \int_0^1 \nabla G\left(\theta_n + t\left(\theta_{n+1} - \theta_n\right)\right) dt V_n^{p-1}$$

$$= -p\gamma_{n+1}\left(g'_{n+1}\right)^T A_n \nabla G\left(\theta_n\right) V_n^{p-1}$$

$$- p\gamma_{n+1}\left(g'_{n+1}\right)^T A_n \int_0^1 \left(\nabla G\left(\theta_n + t\left(\theta_{n+1} - \theta_n\right)\right) - \nabla G\left(\theta_n\right)\right) dt V_n^{p-1}$$

Since $A_n$ is positive and since $\nabla G$ is $L_{\nabla G}$-lipschitz, taking the conditional expectation, it comes, since for all $a, b \ge 0$, $ab \le \frac{1}{p}a^p + \frac{p-1}{p}b^{p/(p-1)}$ and with the help of Assumption **(H1a)**,

$$\mathbb{E}\left[(*)|\mathcal{F}_n\right] \le -p\gamma_{n+1}\nabla G\left(\theta_n\right)^T A_n \nabla G\left(\theta_n\right) V_n^{p-1} + \frac{p}{2}\gamma_{n+1}^2\|A_n\|^2 \mathbb{E}\left[\left\|g'_{n+1}\right\|^2 |\mathcal{F}_n\right]V_n^{p-1}$$

$$\le -p\gamma_{n+1}\lambda_{\min}\left(A_n\right)\|\nabla G\left(\theta_n\right)\|^2 V_n^{p-1} + \frac{p}{2}\beta_{n+1}^2\gamma_{n+1}^2\left(C_1 + C_2\|\theta_n - \theta\|^2\right)V_n^2$$

$$\le -p\mu\gamma_{n+1}\lambda_{\min}\left(A_n\right)V_n^p + \frac{pC_2}{\mu}\beta_{n+1}^2\gamma_{n+1}^2 V_n^p + \frac{pC_1}{2}\beta_{n+1}^2\gamma_{n+1}^2 V_n^{p-1}$$

$$\le -p\mu\gamma_{n+1}\lambda'_{n+1}\mathbf{1}_{\gamma \le 1/2}V_n^p + \left(\frac{pC_2}{\mu} + \frac{C_1(p-1)}{2}\right)\beta_{n+1}^2\gamma_{n+1}^2 V_n^p + \frac{C_1}{2}\beta_{n+1}^2\gamma_{n+1}^2,$$

with $\lambda'_n = \lambda'_0 n^{-\lambda'}$. We also used Assumptions **(A1)** on the first inequality and the fact that, by $\mu$-strong convexity, $\|\theta_n - \theta\|^2 \leq \frac{2}{\mu} V_n \leq \frac{2}{\mu^2} \|\nabla G(\theta_n)\|^2$ on the third inequality. For the same reasons, one has

$$
\begin{aligned}
\mathbb{E}\left[(**)|\mathcal{F}_n\right] &\leq 2^{p-2} p(p-1) L_{\nabla G}^2 \left(\gamma_{n+1}^4 \beta_{n+1}^4 \left(C_1' + \frac{4C_2'}{\mu^2} V_n^2\right) + \gamma_{n+1}^2 \beta_{n+1}^2 \left(\frac{2C_1}{\mu} V_n + \frac{4C_2}{\mu^2} V_n^2\right)\right) V_n^{p-2} \\
&\leq 2^{p-2}(p-1) L_{\nabla G}^2 \gamma_{n+1}^4 \beta_{n+1}^4 \left(2C_1' + \left((p-2)C_1' + \frac{4pC_2'}{\mu^2}\right) V_n^p\right) \\
&\quad + 2^{p-2}(p-1) L_{\nabla G}^2 \gamma_{n+1}^2 \beta_{n+1}^2 \left(\frac{2C_1}{\mu} + \left(\frac{2(p-1)C_1}{\mu} + \frac{4pC_2}{\mu^2}\right) V_n^p\right)
\end{aligned}
$$

In a same way, thanks to Assumptions **(A1")** and **(H1)**, one has

$$
\begin{aligned}
\mathbb{E}\left[(***)|\mathcal{F}_n\right] &\leq 2^{p-2} p(p-1) L_{\nabla G}^p \gamma_{n+1}^{2p} \beta_{n+1}^{2p} \left(C_1^{(p)} + \frac{2^p C_2^{(p)}}{\mu^p} V_n^p\right) \\
&\quad + 2^{p-2} p(p-1) L_{\nabla G}^p \gamma_{n+1}^p \beta_{n+1}^p \left(\frac{1}{2} C_1^{(p)} + \frac{2^p}{\mu^p}\left(\frac{1}{2} + \sqrt{C_2^{(p)}}\right) V_n^p\right)
\end{aligned}
$$

Taking the expectation on $\mathbb{E}\left[(*)|\mathcal{F}_n\right] + \mathbb{E}\left[(**)|\mathcal{F}_n\right] + \mathbb{E}\left[(***)|\mathcal{F}_n\right]$, applying the latter inequalities, it comes

$$
\mathbb{E}\left[V_{n+1}^p\right] \leq \max\left\{\mathbb{E}\left[V_n^p\right], 1\right\} \left(1 - p\mu\lambda'_{n+1}\gamma_{n+1}\mathbf{1}_{\gamma \leq 1/2} + a_p \gamma_{n+1}^2 \beta_{n+1}^2\right)
$$

with

$$
\begin{aligned}
a_p := {}& p\left(\frac{C_2}{\mu} + \frac{C_1}{2}\right) + 2^{p-2}(p-1)pL_{\nabla G}^2 \left(c_\gamma^2 c_\beta^2 \left(C_1' + \frac{4C_2'}{\mu^2}\right) + \frac{2C_1}{\mu} + \frac{4C_2}{\mu^2}\right) \\
&+ 2^{p-2}(p-1)pL_{\nabla G}^p \left(c_\gamma^{2p-2} c_\beta^{2p-2} \left(C_1^{(p)} + \frac{2^p C_2^{(p)}}{\mu^2}\right) + c_\gamma^{p-2} c_\beta^{p-2} \left(\frac{1}{2} C_1^{(p)} + \frac{2^p}{\mu^2}\left(\frac{1}{2} + \sqrt{C_2^{(p)}}\right)\right)\right). \quad (78)
\end{aligned}
$$

**Remark B.2.** *Observe that in the case where $p = 2$, one has*

$$
a_2 = C_1 + \frac{2C_2}{\mu} + \frac{4L_{\nabla G}^2}{\mu} C_1 + \frac{8L_{\nabla G}^2 C_2}{\mu^2} + 2L_{\nabla G}^2 C_1' c_\gamma^2 c_\beta^2 + \frac{8L_{\nabla G}^2 C_2'}{\mu^2} c_\gamma^2 c_\beta^2 \tag{79}
$$

If $\gamma > 1/2$, by summation,

$$
\mathbb{E}\left[V_n^p\right] \leq e^{a_p c_\gamma^2 c_\beta^2 \frac{2\gamma - 2\beta}{2\gamma - 2\beta - 1}} \max\left\{1, \mathbb{E}\left[V_0^p\right]\right\} =: V_p^p.
$$

If $\gamma \leq 1/2$, let $n_0$ be the smallest integer such that $\gamma_{n+1}^2 \beta_{n+1}^2 a_p > p\mu\lambda'_n \gamma_{n+1}$. Recording that $\lambda'_n = \lambda'_0(n+1)^{-\lambda'}$, we have $n_0 = \left\lfloor \left(\frac{c_\gamma c_\beta^2 a_p}{p\mu\lambda'_0}\right)^{\frac{1}{\gamma - 2\beta - \lambda'}} \right\rfloor$. Then,

$$
\begin{aligned}
\mathbb{E}\left[V_n^p\right] &\leq \exp\left(\sum_{n=0}^{n_0} -p\mu\lambda'_n \gamma_{n+1} + a_p \gamma_{n+1}^2 \beta_{n+1}^2\right) \max\left\{1, \mathbb{E}\left[V_0^p\right]\right\} \\
&\leq \exp\left(-p\mu\lambda'_0 c_\gamma \left(1 + \frac{1 + \left(\frac{c_\gamma c_\beta^2 a_p}{p\mu\lambda'_0}\right)^{\frac{1-\gamma-\lambda'}{\gamma-2\beta-\lambda'}}}{1 - \gamma - \lambda'}\right) + c_\gamma^2 c_\beta^2 a_p \left(1 + \frac{1 + \left(\frac{c_\gamma c_\beta^2 a_p}{p\mu\lambda'_0}\right)^{\frac{1-2\gamma+2\beta}{\gamma-2\beta-\lambda'}}}{1 - 2\gamma + 2\beta}\right)\right) =: V_p^p.
\end{aligned}
$$

## B.2 Proof of Lemma 7.2

Recall that $(A_n)_{kk'} = \max\left\{\min\left\{c_\beta n^\beta, \overline{(A_n)}_{kk'}\right\}, \lambda'_0 n^{-\lambda'} \mathbf{1}_{\gamma \leq 1/2}\right\}$ with $\overline{(A_n)}_{kk'} = \frac{\delta_{kk'}}{\sqrt{\frac{1}{n+1}\left(a_k + \sum_{i=0}^{n-1}(\nabla_h g(X_{i+1}, \theta_i)_k)^2\right)}}$. Since $\lambda_{\min}(A_n) \geq \lambda_{\min}(\overline{A_n})$ on the event $\left\{\lambda_{\min}(\overline{A_n}) < c_\beta\right\}$, we

have for $0 < t < 1$

$$
\begin{aligned}
\mathbb{P}\left[\lambda_{\min}\left(A_n\right) < tc_\beta\right] &\leq \mathbb{P}\left[\lambda_{\min}\left(\overline{A_n}\right) < tc_\beta\right] \\
&\leq \mathbb{P}\left[\max_{1 \leq k \leq d} \frac{1}{n+1}\left(a_k + \sum_{i=0}^{n-1}\left(\nabla_h g\left(X_{i+1}, \theta_i\right)_k\right)^2\right) > \frac{1}{c_\beta^2 t^2}\right].
\end{aligned}
$$

Then, Markov inequality for $p > 2$ and Jensen inequality yields

$$
\begin{aligned}
\mathbb{P}&\left[\max_{1 \leq k \leq d} \sqrt{\frac{1}{n+1}\left(a_k + \sum_{i=0}^{n-1}\left(\nabla_h g\left(X_{i+1}, \theta_i\right)_k\right)^2\right)} > \frac{1}{c_\beta t}\right] \\
&\leq c_\beta^{2p} t^{2p} \mathbb{E}\left[\left(\max_{1 \leq k \leq d} \frac{1}{n+1}\left(a_k + \sum_{i=0}^{n-1}\left(\nabla_h g\left(X_{i+1}, \theta_i\right)_k\right)^2\right)\right)^p\right] \\
&\leq c_\beta^{2p} t^{2p} \mathbb{E}\left[\left(\frac{1}{n+1}\left(\sum_{i=1}^{d} a_k + \sum_{i=0}^{n-1}\left\|\nabla_h g\left(X_{i+1}, \theta_i\right)\right\|^2\right)\right)^p\right] \\
&\leq c_\beta^{2p} t^{2p} \frac{1}{n+1}\left(\left(\sum_{i=1}^{d} a_k\right)^p + \sum_{i=0}^{n-1} \mathbb{E}\left[\left\|\nabla_h g\left(X_{i+1}, \theta_i\right)\right\|^{2p}\right]\right).
\end{aligned}
$$

Then, using Assumption **(A1)** and then **(A2)** we get

$$
\begin{aligned}
\mathbb{P}&\left[\max_{1 \leq k \leq d} \sqrt{\frac{1}{n+1}\left(a_k + \sum_{i=0}^{n-1}\left(\nabla_h g(X_{i+1}, \theta_i)_k\right)^2\right)} > \frac{1}{c_\beta t}\right] \\
&\leq c_\beta^{2p} t^{2p} \frac{1}{n+1}\left(\left(\sum_{i=1}^{d} a_k\right)^p + nC_1'' + C_2'' \sum_{i=0}^{n-1} \mathbb{E}\left[\left\|\theta_i - \theta\right\|^{2p}\right]\right) \\
&\leq c_\beta^{2p} t^{2p} \frac{1}{n+1}\left(\left(\sum_{i=1}^{d} a_k\right)^p + nC_1'' + \frac{2^p C_2''}{\mu^p} \sum_{i=0}^{n-1} \mathbb{E}\left[V_n^p\right]\right).
\end{aligned}
$$

By the bound $\mathbb{E}\left[V_n^p\right] \leq V_p^p$ from Lemma 7.1, we finally get

$$
\mathbb{P}\left[\max_{1 \leq k \leq d} \sqrt{\frac{1}{n+1}\left(a_k + \sum_{i=0}^{n-1}\left(\nabla_h g\left(X_{i+1}, \theta_i\right)_k\right)^2\right)} > \frac{1}{c_\beta t}\right] \leq v_n t^{2p}
$$

with

$$
v_n = c_\beta^{2p}\left(\left(\frac{1}{n} \sum_{i=1}^{d} a_k\right)^p + C_1'' + \frac{2^p C_2'' V_p^p}{\mu^p}\right). \tag{80}
$$

### B.3 Proof of Lemma 7.3

Set $E_k = \mathbb{E}\left[\nabla_h g\left(X, \theta\right)_k^2\right]$ and $\partial_k^2 g(h) = \mathbb{E}\left[\nabla_h g(X, h)_k^2\right]$. Then, by Jensen's inequality for $p' \geq 2$,

$$
\left|\left(\overline{A_n}\right)_{kk}\right|^{-2p'} \leq 2^{p'-1}\left|\frac{1}{n+1} \sum_{i=0}^{n-1} \nabla_h g\left(X_{i+1}, \theta_i\right)_k^2 - \partial_k^2 g\left(\theta_i\right)\right|^{p'} + 2^{p'-1}\left|\frac{a_k}{n+1} + \frac{1}{n+1} \sum_{i=0}^{n-1} \partial_k^2 g(\theta_i)\right|^{p'}.
$$

Hence, for any $x > 0$,

$$\mathbb{P}\left[\left|\left(\overline{A_n}\right)_{kk}\right| < \frac{1}{x}\right] = \mathbb{P}\left[\left|\left(\overline{A_n}\right)_{kk}\right|^{-2p'} > x^{2p'}\right] \leq \mathbb{P}\left[\left|\frac{1}{n+1}\sum_{i=0}^{n-1}\nabla_h g\left(X_{i+1},\theta_i\right)_k^2 - \partial_k^2 g\left(\theta_i\right)\right|^{p'} > \frac{x^{2p'}}{2^{p'}}\right]$$

$$+ \mathbb{P}\left[\left|\frac{a_k}{n+1} + \frac{1}{n+1}\sum_{i=0}^{n-1}\partial_k^2 g(\theta_i)\right|^{p'} > \frac{x^{2p'}}{2^{p'}}\right]. \tag{81}$$

Set $M_0 = 0$ and for $n \geq 1$,

$$M_n = \sum_{i=0}^{n-1}\nabla_h g(X_{i+1},\theta_i)_k^2 - \partial_k^2 g(\theta_i).$$

Then, $(M_n)_{n\geq 0}$ is a martingale, and thus by Burkholder's inequality, see (Hall & Heyde, 2014, Theorem 2.10) there exists an explicit constant $C_{p'}$ such that

$$\mathbb{E}\left[|M_n|^{p'}\right] \leq C_{p'}\mathbb{E}\left[\left|\sum_{i=1}^{n}(M_i - M_{i-1})^2\right|^{p'/2}\right] \leq C_{p'}n^{p'/2-1}\sum_{i=1}^{n}\mathbb{E}\left[|M_i - M_{i-1}|^{p'}\right]$$

$$\leq C_{p'}n^{p'/2-1}\sum_{i=0}^{n-1}\mathbb{E}\left[\left|\nabla_h g\left(X_{i+1},\theta_i\right)_k^2 - \partial_k^2 g\left(\theta_i\right)\right|^{p'}\right],$$

where we used Jensen's inequality on the second inequality. By Assumption **(A1)**, the strong convexity of $G$ and Lemma 7.1,

$$\mathbb{E}\left[\left(\nabla_h g\left(X_{i+1},\theta_i\right)_k^2 - \partial_k^2 g\left(\theta_i\right)\right)^{p'}\right] \leq 2^{p'}\mathbb{E}\left[\left(\nabla_h g\left(X_{i+1},\theta_i\right)_k\right)^{2p}\right] \leq 2^{p'}\mathbb{E}\left[\|\nabla_h g\left(X_{i+1},\theta_i\right)\|^{2p'}\right]$$

$$\leq 2^{p'}C_1^{(p')} + 2^{p'}C_2^{(p')}\mathbb{E}\left[\|\theta_i - \theta\|^{2p'}\right]$$

$$\leq 2^{p'}C_1^{(p')} + 2^{2p'}C_2^{(p')}\frac{V_p^{p'}}{\mu^p}.$$

Hence,

$$\mathbb{E}\left[\left|\frac{1}{n+1}\sum_{i=0}^{n-1}\nabla_h g\left(X_{i+1},\theta_i\right)_k^2 - \partial_k^2 g\left(\theta_i\right)\right|^{p'}\right] = \mathbb{E}\left[\left|\frac{1}{n+1}M_n\right|^{p'}\right] \leq 2^{p'}\frac{C_1^{(p')} + 2^{p'}C_2^{(p')}\frac{V_p^{p'}}{\mu^{p'}}}{(n+1)^{p'/2}}, \tag{82}$$

which yields for $x > 0$

$$\mathbb{P}\left[\left|\frac{1}{n+1}\sum_{i=0}^{n-1}\nabla_h g\left(X_{i+1},\theta_i\right)_k^2 - \partial_k^2 g\left(\theta_i\right)\right|^{p'} > \frac{x^{2p'}}{2^{p'}}\right] \leq \frac{2^{2p'}}{x^{2p'}}\frac{C_1^{(p')} + 2^{p'}C_2^{(p')}\frac{V_p^{p'}}{\mu^{p'}}}{(n+1)^{p'/2}}. \tag{83}$$

Next, by Jensen inequality,

$$\left|\frac{a_k}{n+1} + \frac{1}{n+1}\sum_{i=0}^{n-1}\left(\partial_k^2 g\left(\theta_i\right)\right)\right|^{p'} \leq \frac{1}{n+1}\left(|a_k|^{p'} + \sum_{i=0}^{n-1}\left|\partial_k^2 g(\theta_i)\right|^{p'}\right).$$

Using Assumption **(A1)** and then strong convexity yields

$$\left|\partial_k^2 g\left(\theta_i\right)\right|^{p'} \leq C_1^{(p')} + 2^{p'}C_2^{(p')}\frac{V_p^{p'}}{\mu^{p'}},$$

so that

$$\left| \frac{a_k}{n+1} + \frac{1}{n+1} \sum_{i=0}^{n-1} \left( \partial_k^2 g(\theta_i) \right) \right|^{p'} \leq C_1^{(p')} + \frac{|a_k|^{p'}}{n+1} + \frac{2^{p'} C_2^{(p')}}{\mu^{p'}} \left( \frac{1}{n+1} \sum_{i=0}^{n-1} V_i^{p'} \right).$$

Hence, for $\frac{x^{2p'}}{2p'} > C_1^{(p')}$,

$$\mathbb{P}\left[ \left| \frac{a_k}{n+1} + \frac{1}{n+1} \sum_{i=0}^{n-1} \partial_k^2 g(\theta_i) \right|^{p'} > \frac{x^{2p'}}{2p'} \right] \leq \mathbb{P}\left[ \frac{1}{n+1} \left( |a_k|^{p'} + \frac{2^{p'} C_2^{(p')}}{\mu^{p'}} \sum_{i=0}^{n-1} V_i^{p'} \right) > \frac{x^{2p'}}{2p'} - C_1^{(p')} \right]$$

$$\leq \frac{1}{n+1} \frac{\mathbb{E}\left[ |a_k|^{p'} + \frac{2^{p'} C_2^{(p')}}{\mu^{p'}} \sum_{i=0}^{n-1} V_i^{p'} \right]}{\frac{x^{2p'}}{2p'} - C_1^{(p')}}.$$

By equation 33 and the fact that $\frac{1}{n+1} \sum_{i=0}^{n-1} (i+1)^{-\frac{2(1-\gamma)\gamma(\gamma-2\beta)p}{2-\gamma}} \leq \frac{1}{n+1} + \frac{1}{\left| 1 - \frac{2(1-\gamma)\gamma(\gamma-2\beta)p}{2-\gamma} \mathbf{1}_{\frac{2(1-\gamma)\gamma(\gamma-2\beta)p}{2-\gamma} \neq 1} \right|} \frac{\log(n+1)}{(n+1)^{\frac{2(1-\gamma)\gamma(\gamma-2\beta)p}{2-\gamma} \wedge 1}}$, and denoting $\tilde{1} = 1 + \frac{1}{\left| 1 - \frac{2(1-\gamma)\gamma(\gamma-2\beta)p}{2-\gamma} \mathbf{1}_{\frac{2(1-\gamma)\gamma(\gamma-2\beta)p}{2-\gamma} \neq 1} \right|}$,
it comes

$$\frac{1}{n+1} \mathbb{E}\left[ |a_k|^{p'} + \frac{2^{p'} C_2^{(p')}}{\mu^{p'}} \sum_{i=0}^{n-1} V_i^{p'} \right] = \frac{|a_k|^{p'} + \frac{2^{p'} C_2^{(p')}}{\mu^{p'}} \sum_{i=0}^{n-1} \mathbb{E}\left[ V_i^{p'} \right]}{n+1}$$

$$\leq \frac{2^{p'} C_2^{(p')}}{\mu^{p'}} \tilde{K}_2 \tilde{1} \frac{\log(n+1)}{(n+1)^{\frac{2(1-\gamma)\gamma(\gamma-2\beta)p}{2-\gamma} \wedge 1}} + \frac{2^{p'} C_2^{(p')}}{\mu^{p'}(n+1)} \left[ 1 + |a_k|^{p'} + \tilde{K}_1 \sum_{i=0}^{\infty} \exp\left( -c_\gamma \mu \lambda_0 i^{1-(\lambda+\gamma)} (1 - \varepsilon'(i)) \right) \right]$$

$$\leq M(\beta) \frac{\log(n+1)}{(n+1)^{\frac{2(1-\gamma)\gamma(\gamma-2\beta)p}{2-\gamma} \wedge 1}}$$

with for $n \geq 2$

$$M(\beta) = \frac{2^{p'} C_2^{(p')}}{\mu^{p'}} \left[ \tilde{K}_2 \tilde{1} + 1 + |a_k|^{p'} + \tilde{K}_1 \sum_{n=0}^{+\infty} \exp\left( -c_\gamma \mu \lambda_0 n^{1-(\lambda+\gamma)} (1 - \varepsilon'(n)) \right) \right]$$

Choosing

$$\lambda_0 = \left[ 2^{p'} (C_1^{(p')} + 1) \right]^{-\frac{1}{2p'}} \tag{84}$$

yields then

$$\mathbb{P}\left[ \left| \frac{a_k}{n+1} + \frac{1}{n+1} \sum_{i=0}^{n-1} \partial_k^2 g(\theta_i) \right|^{p'} > \frac{\lambda_0^{-2p'}}{2p'} \right] \leq \frac{M(\beta) \log(n+1)}{(n+1)^{\frac{2(1-\gamma)\gamma(\gamma-2\beta)p}{2-\gamma} \wedge 1}}.$$

Putting the latter inequality with equation 81 and equation 83 gives then

$$\mathbb{P}\left[ \lambda_{\min}\left( \overline{A_n} \right) < \lambda_0 \right] \leq \sum_{k=1}^{d} \mathbb{P}\left[ \left| \left( \overline{A_n} \right)_{kk} \right| < \lambda_0 \right]$$

$$\leq \frac{dM(\beta) \log(n+1)}{(n+1)^{\frac{2(1-\gamma)\gamma(\gamma-2\beta)p}{2-\gamma} \wedge 1}} + \frac{d 2^{p'} \left( C_1^{(p')} + 2^{p'} C_2^{(p')} \frac{V_p^{p'}}{\mu^{p'}} \right)}{(C_1^{(p')} + 1) n^{p'/2}}$$

$$\leq \frac{v_0 \log(n+1)}{(n+1)^{\frac{2(1-\gamma)\gamma(\gamma-2\beta)p}{2-\gamma} \wedge 1}}$$

with

$$v_0 = dM(\beta) + \frac{d2^{p'}\left(C_1^{(p')} + 2^{p'}C_2^{(p')}\frac{V_p^{p'}}{\mu^{p'}}\right)}{C_1^{(p')} + 1}. \tag{85}$$

Since $\mathbb{P}\left[\lambda_{\min}(A_n) < \lambda_0\right] \leq \mathbb{P}\left[\lambda_{\min}(\overline{A_n}) < \lambda_0\right]$, the result is deduced.

### B.4  Proof of Lemma 7.4

Set $E_k = \mathbb{E}\left[\nabla_h(X,\theta)_k^2\right]$ and $\partial_k^2 g(h) = \mathbb{E}\left[\nabla_h(X,h)_k^2\right]$. Then

$$\mathbb{E}\left[\left|(\overline{A_n})_{kk}^{-2} - E_k\right|^{p'}\right] \leq 2^{p'-1}\mathbb{E}\left[\left|\frac{1}{n+1}\sum_{i=0}^{n-1}\nabla_h g(X_{i+1},\theta_i)_k^2 - \partial_k^2 g(\theta_i)\right|^{p'}\right]$$

$$+ 2^{p'-1}\mathbb{E}\left[\left|\frac{a_k - E_k}{n+1} + \frac{1}{n+1}\sum_{i=0}^{n-1}(\partial_k^2 g(\theta_i) - E_k)\right|^{p'}\right].$$

By equation 82,

$$\mathbb{E}\left[\left|\frac{1}{n+1}\sum_{i=0}^{n-1}\nabla_h g(X_{i+1},\theta_i)_k^2 - \partial_k^2 g(\theta_i)\right|^{p'}\right] \leq 2^{p'}\frac{C_1^{(p')} + 2^{p'}C_2^{(p')}\frac{V_p^{p'}}{\mu^{p'}}}{(n+1)^{p'/2}}.$$

Next, by Jensen inequality,

$$\mathbb{E}\left[\left|\frac{a_k - E_k}{n+1} + \frac{1}{n+1}\sum_{i=0}^{n-1}\left(\partial_k^2 g(\theta_i) - E_k\right)\right|^{p'}\right] \leq \frac{1}{n+1}\left((a_k - E_k)^{p'} + \sum_{i=0}^{n-1}\mathbb{E}\left[\left|\partial_k^2 g(\theta_i) - E_k\right|^{p'}\right]\right).$$

Using Cauchy-Schwarz inequality, Assumption **(A1')** and then Assumption **(A1)** yields

$$\mathbb{E}\left[\left|\partial_k^2 g(\theta_i) - E_k\right|^{p'}\right] = \mathbb{E}\left[\left|\mathbb{E}\left[\nabla_h g(\theta_i,X)_k^2 - \nabla_h g(\theta,X)_k^2\mid\theta_i\right]\right|^{p'}\right]$$

$$\leq \mathbb{E}\left[\left|\mathbb{E}\left[(\nabla_h g(\theta_i,X)_k - \nabla_h g(\theta,X)_k)(\nabla_h g(\theta_i,X)_k + \nabla_h g(\theta,X)_k)\mid\theta_i\right]\right|^{p'}\right]$$

$$\leq \mathbb{E}\left[\mathbb{E}\left[(\nabla_h g(\theta_i,X)_k - \nabla_h g(\theta,X)_k)^2\mid\theta_i\right]^{p'/2}\mathbb{E}\left[(\nabla_h g(\theta_i,X)_k + \nabla_h g(\theta,X)_k)^2\mid\theta_i\right]^{p'/2}\right]$$

$$\leq 2^{p'/2-1}L_{\nabla g}^{p'/2}\mathbb{E}\left[\|\theta_i - \theta\|^{p'}(2C_1^{p'/2} + C_2^{p'/2}\|\theta_i - \theta\|^{p'})\right]$$

$$\leq \frac{2^{p'}L_{\nabla g}^{p'/2}C_1^{p'/2}}{\mu^{p'/2}}\mathbb{E}\left[V_i^{p'/2}\right] + \frac{2^{3p'/2-1}C_2^{p'/2}L_{\nabla g}^{p'/2}}{\mu^{p'}}\mathbb{E}\left[V_i^{p'}\right]$$

$$\leq \frac{2^{p'}L_{\nabla g}^{p'/2}C_1^{p'/2}}{\mu^{p'/2}}\sqrt{c_i} + \frac{2^{3p'/2-1}C_2^{p'/2}L_{\nabla g}^{p'/2}}{\mu^{p'}}c_i,$$

where $c_i$ is given in equation 33. Putting all the latter bounds together yields, using that $E_k \leq C_1$,

$$\mathbb{E}\left[\left|\frac{a_k - E_k}{n+1} + \frac{1}{n+1}\sum_{i=0}^{n-1}\left(\partial_k^2 g(\theta_i) - E_k\right)\right|^{p'}\right]$$

$$\leq \frac{1}{n+1}\left[2^{p'-1}(a_k^{p'} + C_1^{p'}) + \sum_{i=0}^{n-1}\left(\frac{2^{p'}L_{\nabla g}^{p'/2}C_1^{p'/2}}{\mu^{p'/2}}\sqrt{c_i} + \frac{2^{3p'/2-1}C_2^{p'/2}L_{\nabla g}^{p'/2}}{\mu^{p'}}c_i\right)\right].$$

Hence, noting that $V_p < \infty$ by Assumption **(A1')** and Lemma 7.1,

$$\mathbb{E}\left[|(\overline{A_n})_{kk}^{-2} - E_k|^{p'}\right]$$

$$\leq 2^{p'-1}\frac{C_1^{(p')} + 2^{p'}C_2^{(p')}\frac{V_p^{p'}}{\mu^{p'}}}{n} + \underbrace{\frac{2^{p'-1}}{n+1}\left[2^{p'-1}(a_k^{p'} + C_1^{p'}) + \sum_{i=0}^{n-1}\left(\frac{2^{p'}L_{\nabla g}^{p'/2}C_1^{p'/2}}{\mu^{p'/2}}\sqrt{c_i} + \frac{2^{3p'/2-1}C_2^{p'/2}L_{\nabla g}^{p'/2}}{\mu^{p'}}c_i\right)\right]}_{:=\bar{c}_n},$$

with, by equation 33, $\bar{c}_n = O\left(\log(n)n^{-\left[\frac{(1-\gamma)\gamma(\gamma-2\beta)p}{2-\gamma}\wedge 1\right]}\right)$. Since by **(A6)** we have $E_k \geq \alpha$, we deduce by Markov's inequality that

$$\mathbb{P}\left[\left(\overline{A_n}\right)_{kk}^{-1} \leq \sqrt{\alpha/2}\right] = \mathbb{P}\left[\left(\overline{A_n}\right)_{kk}^{-2} \leq \alpha/2\right] \leq \frac{2^{p'}}{\alpha^{p'}}\mathbb{E}\left[\left|\left(\overline{A_n}\right)_{kk}^{-2} - E_k\right|^{p'}\right] \leq \frac{2^{p'}\bar{c}_n}{\alpha^{p'}}.$$

Hence, we have

$$\mathbb{E}\left[(A_n)_{kk}^4\right] = \mathbb{E}\left[\mathbf{1}_{\left(\overline{A_n}\right)_{kk}\geq\sqrt{\frac{2}{\alpha}}}(A_n)_{kk}^4\right] + \mathbb{E}\left[\mathbf{1}_{\left(\overline{A_n}\right)_{kk}<\sqrt{\frac{2}{\alpha}}}(A_n)_{kk}^4\right]$$

$$\leq \mathbb{E}\left[\mathbf{1}_{\left(\overline{A_n}\right)_{kk}\geq\sqrt{\frac{2}{\alpha}}}c_\beta^4 n^{4\beta}\right] + \mathbb{E}\left[\mathbf{1}_{\left(\overline{A_n}\right)_{kk}<\sqrt{\frac{2}{\alpha}}}\left(\overline{A_n}\right)_{kk}^4\right]$$

$$\leq c_\beta^4 n^{4\beta}\mathbb{P}\left[\left(\overline{A_n}\right)_{kk}^{-1} \leq \sqrt{\alpha/2}\right] + \frac{4}{\alpha^2} \leq \frac{2^{p'}c_\beta^4 n^{4\beta}\bar{c}_n}{\alpha^{p'}} + \frac{4}{\alpha^2}.$$

Since $\bar{c}_n = O\left(\log(n)n^{-\left[\frac{(1-\gamma)\gamma(\gamma-2\beta)p}{2-\gamma}\wedge 1\right]}\right)$, for $\beta < \frac{(1-\gamma)\gamma(\gamma-2\beta)p}{4(2-\gamma)} \wedge \frac{1}{4}$ we have $\left[\frac{(1-\gamma)\gamma(\gamma-2\beta)p}{2-\gamma} \wedge 1\right] - 4\beta > 0$ and thus

$$w(\beta) = \sup_{n\geq 1}\bar{c}_n n^{4\beta} < +\infty,$$

and finally

$$\mathbb{E}\left[\|A_n\|^4\right] \leq \sum_{k=1}^d \mathbb{E}\left[(A_n)_{kk}^4\right] \leq C_S^4$$

with

$$C_S^4 = d\left[\frac{2^{p'}c_\beta^4 w(\beta)}{\alpha^{p'}} + \frac{4}{\alpha^2}\right]. \tag{86}$$

## B.5   Proof of Lemma 7.5

First, we have by **(A6')**

$$\mathbb{E}\left[\left((\overline{A_n})_{kk}\right)^{-2}\right] = \mathbb{E}\left[\frac{1}{n+1}\sum_{i=0}^{n-1}\nabla_h g\left(X_{i+1}, \theta_i\right)_k^2\right] = \frac{1}{n+1}\sum_{i=0}^{n-1}\mathbb{E}\left[\nabla_h g\left(X_{i+1}, \theta_i\right)_k^2\right] \geq \alpha.$$

Then, as in the proof of Lemma 7.4,

$$\mathbb{E}\left[\left|\frac{1}{n+1}\sum_{i=0}^{n-1}\nabla_h g\left(X_{i+1}, \theta_i\right)_k^2 - \frac{1}{n+1}\sum_{i=0}^{n-1}\mathbb{E}\left[\nabla_h g\left(X_{i+1}, \theta_i\right)_k^2\right]\right|^2\right] \leq \frac{C_1' + C_2'\frac{4V_2^2}{\mu^2}}{n}.$$

Hence, by Markov inequality,

$$\mathbb{P}\left[\left((\overline{A_n})_{kk}\right)^{-2} \leq \alpha/2\right] \leq \mathbb{P}\left[\left|\frac{1}{n+1}\sum_{i=0}^{n-1}\nabla_h g\left(X_{i+1}, \theta_i\right)_k^2 - \frac{1}{n+1}\sum_{i=0}^{n-1}\mathbb{E}\left[\nabla_h g\left(X_{i+1}, \theta_i\right)_k^2\right]\right|^2 > \frac{\alpha^2}{4}\right]$$

$$\leq \frac{4\left(C_1' + C_2'\frac{4V_2^2}{\mu^2}\right)}{n\alpha^2}.$$

We deduce as in Lemma 7.4 that

$$\mathbb{E}\left[(A_n)_{kk}^4\right] \le c_\beta^4 n^{4\beta} \mathbb{P}\left[\left(\overline{A_n}\right)_{kk}^{-1} \le \sqrt{\alpha/2}\right] + \frac{4}{\alpha^2} \le \frac{4\left(C_1' + C_2' \frac{4V_2^2}{\mu^2}\right)}{n^{1-4\beta}\alpha^2} + \frac{4}{\alpha^2}.$$

When $\beta < 1/4$, we finally get

$$\mathbb{E}\left[\|A_n\|^4\right] \le \sum_{k=1}^{d} \mathbb{E}\left[(A_n)_{kk}^4\right] \le C_S^4$$

with

$$C_S^4 = \frac{4d\left(1 + C_1' + C_2' \frac{4V_2^2}{\mu^2}\right)}{\alpha^2}. \tag{87}$$

# C  Proof of technical Lemma and Propositions for linear regression

## C.1  Proof of Lemma 7.6

Remark that

$$\left\|\widetilde{S}_n\right\| \le \frac{\alpha_+}{n+m}\left(m\|S_0\| + \sum_{i=1}^{n}\left\|X_i X_i^T\right\|\right) \le \frac{\alpha_+}{n}\left(m\|S_0\| + \sum_{i=1}^{n}\|X_i\|^2\right).$$

Hence, for $\lambda > 0$,

$$\mathbb{P}\left[\lambda_{\min}\left(\widetilde{S}_n^{-1}\right) < \lambda\right] = \mathbb{P}\left[\|\widetilde{S}_n\| > 1/\lambda\right] \le \mathbb{P}\left[\frac{\alpha_+}{n}\left(m\|S_0\| + \sum_{i=1}^{n}\|X_i\|^2\right) > \lambda^{-1}\right].$$

Taking $\lambda_0 = \left(2\alpha_+ \mathbb{E}\left[\|X\|^2\right]\right)^{-1}$ yields then

$$\mathbb{P}\left[\lambda_{\min}\left(\widetilde{S}_n^{-1}\right) < \lambda_0\right] \le \mathbb{P}\left[\frac{1}{n}\left(m\|S_0\| + \sum_{i=1}^{n}\left(\|X_i\|^2 - \mathbb{E}\left[\|X\|^2\right]\right)\right) > \mathbb{E}\left[\|X\|^2\right]\right].$$

Taking the $p$-power, applying Markov inequality and then Rosenthal inequality yields that

$$\mathbb{P}\left[\frac{1}{n}\left(m\|S_0\| + \sum_{i=1}^{n}\left(\|X_i\|^2 - \mathbb{E}\left[\|X\|^2\right]\right)\right) > \mathbb{E}\left[\|X\|^2\right]\right]$$

$$\le \mathbb{P}\left[\left(\frac{1}{n}\left(m\|S_0\| + \left|\sum_{i=1}^{n}\left(\|X_i\|^2 - \mathbb{E}\left[\|X\|^2\right]\right)\right|\right)\right)^p > \left(\mathbb{E}\left[\|X\|^2\right]\right)^p\right]$$

$$\le \frac{1}{\left(\mathbb{E}\left[\|X\|^2\right]\right)^p}\mathbb{E}\left[\frac{1}{n^p}\left(m\|S_0\| + \left|\sum_{i=1}^{n}\left(\|X_i\|^2 - \mathbb{E}\left[\|X\|^2\right]\right)\right|\right)^p\right]$$

$$\le \frac{2^{p-1}}{\left(\mathbb{E}\left[\|X\|^2\right]\right)^p}\left(C_1(p)n^{1-p}\mathbb{E}\left[|Z|^p\right] + C_2(p)n^{-p/2}\left(\mathbb{E}\left[|Z|^2\right]\right)^{p/2} + m^p\|S_0\|^p n^{-p}\right),$$

with $Z = \|X\|^2 - \mathbb{E}\left[\|X\|^2\right]$.

If $X$ is a subgaussian with subgaussian norm $\|X\|_{\psi_2}$, a similar reasoning yields

$$\mathbb{P}\left[\lambda_{\min}\left(\widetilde{S}_n^{-1}\right) < \lambda_0\right] \le \mathbb{P}\left[\frac{\alpha_+}{n}\left(m\|S_0\| + \sum_{i=1}^{n}\|X_i\|^2\right) > \lambda_0^{-1}\right]$$

$$\le \mathbb{P}\left[\sum_{i=1}^{n}\left(\|X_i\|^2 - \mathbb{E}[\|X_i\|^2]\right) > n(\lambda_0^{-1}/\alpha_+ - \mathbb{E}[\|X\|^2]) - m\|S_0\|\right]$$

$$\le 2\exp\left(-\frac{c\left(n(\lambda_0^{-1}/\alpha_+ - \mathbb{E}[\|X\|^2]) - m\|S_0\|\right)^2}{n\|X\|_{\psi_2}}\right),$$

with $c > 0$ absolute constant, where we used the generalized Hoeffding inequality for sub-Gaussian random variables and the fact that centering alters the sub-Gaussian norm by a universal constant, see (Vershynin, 2018, Theorem 2.6.2) and (Vershynin, 2018, Lemma 2.6.8). Taking $\lambda_0 = \left(2\alpha_+ \mathbb{E}\left[\|X\|^2\right]\right)^{-1}$ and assuming that $m \leq \frac{n\mathbb{E}[\|X\|^2]}{2\|S_0\|}$ yields then

$$\mathbb{P}\left[\lambda_{\min}\left(\widetilde{S}_n^{-1}\right) < \lambda_0\right] \leq 2\exp\left(-\frac{cn(\mathbb{E}[\|X\|^2])}{\|X\|_{\psi_2}}\right)$$

for some numeric $c > 0$.

## C.2  Proof of Lemma 7.7

By definition of $\overline{S}_n$, $\overline{S}_n = \widetilde{S}_n$ on the event $T_n = \{\lambda_{\min}\left(\widetilde{S}_n\right) \geq \frac{1}{c_\beta n^\beta}\}$. Hence, for the same $\lambda_0$ as in Lemma 7.6,

$$\mathbb{P}\left[\lambda_{\min}\left(\bar{S}_n^{-1}\right) < \lambda_0\right] = \mathbb{P}\left[T_n \cap \left\{\lambda_{\min}\left(\widetilde{S}_n^{-1}\right) < \lambda_0\right\}\right] + \mathbb{P}\left[T_n^c\right]$$
$$\leq \mathbb{P}\left[\lambda_{\min}\left(\widetilde{S}_n^{-1}\right) < \lambda_0\right] + \mathbb{P}\left[T_n^c\right]. \tag{88}$$

By Lemma 7.6,

$$\mathbb{P}\left[\lambda_{\min}\left(\widetilde{S}_n^{-1}\right) < \lambda_0\right] \leq \tilde{v}_n, \tag{89}$$

with $\tilde{v}_n$ given in Lemma 7.6. Then, for $n \geq n_0$, where $n_0$ is defined in equation 36, we have $n \geq \left(\frac{1}{c_\beta c_2 \alpha_-}\left(\frac{n+m}{n}\right)\right)^{-1/\beta}$, and thus $\frac{n\alpha_-}{n+m}c_2 \geq \frac{1}{c_\beta n^\beta}$. In particular, on the event $\left\{\lambda_{\min}\left(\frac{1}{n}\sum_{i=1}^n X_i X_i^T\right) \geq c_2\right\}$, we have

$$\lambda_{\min}\left(\widetilde{S}_n\right) = \lambda_{\min}\left(\frac{\alpha_n}{n+m}\left(mS_0 + \sum_{i=1}^n X_i X_i^T\right)\right)$$
$$\geq \frac{n\alpha_-}{n+m}\lambda_{\min}\left(\frac{1}{n}\sum_{i=1}^n X_i X_i^T\right) \geq \frac{n\alpha_-}{n+m}c_2 \geq \frac{1}{c_\beta n^\beta}.$$

Hence, for $n \geq n_0$, $\left\{\lambda_{\min}\left(\frac{1}{n}\sum_{i=1}^n X_i X_i^T\right) \geq c_2\right\} \subset T_n$ and thus by Proposition 7.2 and the fact that $n \geq c_1 d$,

$$\mathbb{P}\left[T_n^c\right] \leq \mathbb{P}\left[\lambda_{\min}\left(\frac{1}{n}\sum_{i=1}^n X_i X_i^T\right) < c_2\right] \leq \exp(-c_3 n). \tag{90}$$

Using equation 89 and equation 90 in equation 88 yields then

$$\mathbb{P}\left[\lambda_{\min}\left(\overline{S}_n^{-1}\right) < \lambda_0\right] \leq \tilde{v}_n + 2\exp(-c_3 n)$$

for $n \geq n_0$. The statement of the lemma is then a rewriting of the latter inequality.

## C.3  Proof of Lemma 7.8

Since we have

$$\|\bar{S}_n^{-1}\| = \min\left\{\|\tilde{S}_n^{-1}\|, \beta_{n+1}\right\} = \min\left\{\frac{1}{\lambda_{\min}\left\{\tilde{S}_n\right\}}, \beta_{n+1}\right\},$$

for $c_1, c_2, c_3$ given in Proposition 7.2, $n \geq c_1 d \vee m$ and $\kappa > 0$,

$$\mathbb{E}\left[\|\bar{S}_n^{-1}\|^\kappa\right] \leq \beta_{n+1}^\kappa \mathbb{P}\left[\lambda_{\min}\left(\frac{1}{n}\sum_{i=1}^n X_i X_i^T\right) \leq c_2\right] + \left(\frac{n}{n+m}\alpha_- c_2\right)^{-\kappa}$$
$$\leq 2\beta_{n+1}^\kappa \exp\left(-c_3 n\right) + \left(\alpha_- c_2/2\right)^{-\kappa}.$$

Since $\tilde{S}_n = \frac{\alpha_n}{n+m}\left(mS_0 + \sum_{i=1}^n X_i X_i^T\right)$ and $\sum_{i=1}^n X_i X_i^T \geq 0$, $\alpha_n \geq \alpha_-$, we have $\tilde{S}_n \geq \frac{m\alpha_-}{n+m}S_0$ and thus $\|\bar{S}_n^{-1}\| \leq \|\tilde{S}_n^{-1}\| \leq \frac{(n+m)}{\alpha_- m}\|S_0^{-1}\|$ for $n \geq 1$. Hence, for $n \leq c_1 d \vee m$, $\|\tilde{S}_n^{-1}\| \leq \frac{(c_1 d+2)}{\alpha_-}\|S_0^{-1}\|$ and we finally get the result.

## C.4 Proof of Proposition 7.3

Recall that $\beta_n = c_\beta n^\beta$. Since, for $\kappa > 0$, the map $g : t \mapsto (c_\beta t^\beta)^\kappa \exp(-c_3 t)$ is bounded from above by $c_\beta^\kappa \left(\frac{\beta\kappa}{ec_3}\right)^{\beta\kappa}$, we get

$$\sup_{n \geq c_1 d} \mathbb{E}\left[\|\bar{S}_n^{-1}\|^\kappa\right] \leq 2c_\beta^\kappa \left(\frac{\beta\kappa}{ec_3}\right)^{\beta\kappa} + (\alpha_- c_2/2)^{-\kappa}.$$

Taking into account the case $n \leq c_1 d \vee m$ yields then

$$\sup_{n \geq 1} \mathbb{E}\left[\|\bar{S}_n^{-1}\|^2\right] \leq \max\left\{2c_\beta^2 \left(\frac{2\beta}{ec_3}\right)^{2\beta} + (\alpha_- c_2/2)^{-2}, \left[\frac{c_1 d + 2}{\alpha_-}\|S_0^{-1}\|\right]^2\right\},$$

and

$$\sup_{n \geq 1} \mathbb{E}\left[\|\bar{S}_n^{-1}\|^4\right] \leq \max\left\{2c_\beta^4 \left(\frac{4\beta}{ec_3}\right)^{4\beta} + (\alpha_- c_2/2)^{-4}, \left[\frac{c_1 d + 2}{\alpha_-}\|S_0^{-1}\|\right]^4\right\}.$$

## C.5 Proof of Lemma 7.9

First notice that

$$\left\|\overline{S}_n^{-1} - H^{-1}\right\| = \left\|\overline{S}_n^{-1}(H - \overline{S}_n)H^{-1}\right\| \leq \left\|\overline{S}_n^{-1}\right\| \|H - \overline{S}_n\| \|H^{-1}\|.$$

Under hypothesis of Proposition 7.2,

$$\mathbb{P}\left[\lambda_{\min}\left(\tilde{S}_n\right) \leq \alpha_- c_2/2\right] \leq 2\exp\left(-c_3 n\right)$$

for $n \geq c_1 d \vee m$. Since $\left\|\bar{S}_n^{-1}\right\| \leq \left\|\tilde{S}_n^{-1}\right\|$, $\lambda_{\min}\left(\bar{S}_n\right) \geq \lambda_{\min}\left(\tilde{S}_n\right)$ and thus we also have

$$\mathbb{P}\left[\lambda_{\min}\left(\bar{S}_n\right) \leq \alpha_- c_2/2\right] \leq 2\exp\left(-c_3 n\right)$$

for $n \geq c_1 d \vee m$. Hence, for $n \geq n_0$,

$$\mathbb{E}\left[\left\|\overline{S}_n^{-1} - H^{-1}\right\|^2\right] = \mathbb{E}\left[\mathbf{1}_{\lambda_{\min}(\overline{S}_n) \leq \alpha_- c_2/2}\left\|\overline{S}_n^{-1} - H^{-1}\right\|^2\right] + \mathbb{E}\left[\mathbf{1}_{\lambda_{\min}(\overline{S}_n) > \alpha_- c_2/2}\left\|\overline{S}_n^{-1} - H^{-1}\right\|^2\right]$$

$$\leq \frac{1}{(\lambda_{\min}\beta_n)^2}\mathbb{E}\left[\mathbf{1}_{\lambda_{\min}(\overline{S}_n) \leq \alpha_- c_2/2}\left\|\overline{S}_n - H\right\|^2\right] + \frac{1}{(\lambda_{\min}\alpha_- c_2/2)^2}\mathbb{E}\left[\left\|\tilde{S}_n - H\right\|^2\right],$$

where we used on the last equality that for $n \geq n_0$, $\bar{S}_n = \tilde{S}_n$ on the event $\{\lambda_{\min}(\bar{S}_n > \alpha_- c_2/2)\}$, as in the proof of Lemma 7.7. The first summand can be bounded using Hölder inequality with $\frac{1}{q} + \frac{1}{q'} = 1$ and $q' = p/2$ as

$$\mathbb{E}\left[\mathbf{1}_{\lambda_{\min}(\overline{S}_n) \leq \alpha_- c_2/2}\left\|\overline{S}_n - H\right\|^2\right] \leq \mathbb{P}\left[\lambda_{\min}\left(\overline{S}_n\right) \leq \alpha_- c_2/2\right]^{1/q}\mathbb{E}\left[\left\|\overline{S}_n - H\right\|^{2q'}\right]^{1/q'}$$

$$\leq 2\exp(-c_3(p-2)n/p)\mathbb{E}\left[\left\|\overline{S}_n - H\right\|^p\right]^{2/p}.$$

Using the upper bound on $H$ and the convexity inequality $(a + b)^p \leq 2^{p-1}(a^p + b^p)$ yields the rough bound

$$\mathbb{E}\left[\left\|\overline{S}_n - H\right\|^p\right]^{2/p} \leq \mathbb{E}\left[\left(\|\overline{S}_n\| + \|H\|\right)^p\right]^{2/p} \leq 2^{2-2/p}\left(\mathbb{E}\left[\|\overline{S}_n\|^p\right] + \lambda_{\max}^p\right)^{2/p}$$

$$\leq 4\max\left\{\lambda_{\max}^2, \mathbb{E}\left[\|\overline{S}_n\|^p\right]^{\frac{2}{p}}\right\}$$

Since $X$ admits moments of order $2p$, we get

$$\mathbb{E}\left[\|\overline{S}_n\|^p\right]^{1/p} \leq \frac{\alpha_+ m}{n+m}\|S_0\| + \frac{\alpha_+ n}{n+m}\mathbb{E}\left[\left(\frac{1}{n}\sum_{i=1}^n \|X_i\|^2\right)^p\right]^{1/p} \leq \alpha_+\|S_0\| + \alpha_+\left(\mathbb{E}\left[\|X\|^{2p}\right]\right)^{1/p}.$$

We hence get

$$\mathbb{E}\left[\mathbf{1}_{\lambda_{\min}(\overline{S}_n)\leq\alpha-c_2/2}\left\|\overline{S}_n-H\right\|^2\right] \leq 8\exp\left(-c_3(p-2)n/p\right)\max\left\{\lambda_{\max}^2, 2\alpha_+\left(\|S_0\|^2+\left(\mathbb{E}\left[\|X\|^{2p}\right]\right)^{2/p}\right)\right\}$$
$$= 16\alpha_+\exp\left(-c_3(p-2)n/p\right)\left(\|S_0\|^2+\left(\mathbb{E}\left[\|X\|^{2p}\right]\right)^{2/p}\right).$$

For the second summand, using the relation between Frobenius norm and operator norm yields

$$\mathbb{E}\left[\left\|\tilde{S}_n-H\right\|^2\right] \leq \frac{2|\alpha_n-1|^2}{(n+m)^2}\mathbb{E}\left[\|mS_0+\sum_{k=1}^{n}X_kX_k^T\|^2\right] + 2\mathbb{E}\left[\left\|\frac{1}{n+m}\left(mS_0+\sum_{k=1}^{n}X_kX_k^T\right)-H\right\|_F^2\right]$$
$$:= R_1 + R_2.$$

By hypothesis, $|\alpha_n-1|\leq\frac{C_\alpha}{n}$ so that by Jensen inequality the first term is bounded by

$$R_1 \leq \frac{2C_\alpha}{n^2}\left(\|S_0\|^2+\mathbb{E}[\|X\|^2]\right),$$

and the second term is bounded by

$$R_2 \leq \frac{2}{(n+m)^2}\|\alpha_n mS_0-H\|_F^2 + \frac{2}{(n+m)^2}\mathbb{E}\left[\left\|\sum_{k=1}^{n}\left(X_kX_k^T-\mathbb{E}\left[XX^T\right]\right)\right\|_F^2\right]$$
$$\leq \frac{2}{(n+m)^2}\|mS_0-H\|_F^2 + \frac{2}{n+m}\mathbb{E}\left[\|XX^T-\mathbb{E}\left[XX^T\right]\|_F^2\right]$$
$$\leq \frac{2}{(n+m)^2}\|mS_0-H\|_F^2 + \frac{2}{n+m}\mathbb{E}\left[\|X\|^4\right].$$

Putting all the above bounds together yields the bound of the statement.

### C.6 Proof of Lemma 7.10

Remark first that as in the proof of Lemma 7.1, one has

$$V_{n+1} \leq V_n - \gamma_{n+1}\left(g'_{n+1}\right)^T A_n \int_0^1 \nabla G\left(\theta_n+t\left(\theta_{n+1}-\theta_n\right)\right)dt.$$

Then,

$$\left\|\int_0^1 \nabla G\left(\theta_n+t\left(\theta_{n+1}-\theta_n\right)\right)dt\right\| \leq L_{\nabla G}\left(\|\theta_n-\theta\|+\gamma_{n+1}\|A_n\|\|g'_{n+1}\|\right),$$

which implies that

$$V_{n+1} \leq V_n + \gamma_{n+1}\|A_n\|L_{\nabla G}\left(\|g'_{n+1}\|\cdot\|\theta_n-\theta\|+\gamma_{n+1}\|A_n\|\cdot\|g'_{n+1}\|^2\right).$$

Using the fact that $g'_{n+1}=(\epsilon+X_{n+1}^T(\theta-\theta_n))X_{n+1}$ yields then

$$V_{n+1} \leq V_n + \gamma_{n+1}\|A_n\|L_{\nabla G}\left(\|X_{n+1}\|^2\|\theta_n-\theta\|^2+|\epsilon|\|\theta_n-\theta\|+2\gamma_{n+1}\|A_n\|\cdot\left(\|X_{n+1}\|^4\|\theta_n-\theta\|^2+\epsilon^2\right)\right)$$
$$\leq V_n\left(1+\frac{2(n+1)\gamma_{n+1}L_{\nabla G}}{\mu}\|S_0^{-1}\|\left(2+\|X_{n+1}\|^2\right)+\frac{4(n+1)^2\gamma_{n+1}^2L_{\nabla G}}{\mu}\|S_0^{-2}\|\|X_{n+1}\|^4\right)$$
$$+ 2|\epsilon|^2L_{\nabla G}\left(\gamma_{n+1}(n+1)\|S_0^{-1}\|+\gamma_{n+1}^2(n+1)^2\|S_0^{-2}\|\right),$$

where we used the strong convexity of $G$ and bounded $A_n$ by $(n+1)\|S_0^{-1}\|$. Hence, taking the square in the above inequality and taking the expectation conditioned on $\mathcal{F}_n$ gives, for $p \geq 1$,

$$
\begin{aligned}
\mathbb{E}[V_{n+1}^p] \leq & 2^{p-1}\mathbb{E}[V_n^p]\bigg(1 + \frac{2(n+1)\gamma_{n+1}L_{\nabla G}}{\mu}\|S_0^{-1}\|\left(2 + \mathbb{E}\left[\|X_{n+1}\|^8\right]^{1/4}\right) \\
& \qquad\qquad + \frac{4(n+1)^2\gamma_{n+1}^2 L_{\nabla G}}{\mu}\|S_0^{-2}\|\mathbb{E}\left[\|X_{n+1}\|^8\right]^{1/2}\bigg)^p \\
& + 2^{2p-1}\mathbb{E}\left[|\epsilon|^{2p}\right]L_{\nabla G}^p\left(\gamma_{n+1}(n+1)\|S_0^{-1}\| + \gamma_{n+1}^2(n+1)^2\|S_0^{-2}\|\right)^p \\
\leq & \mathbb{E}[V_n^p]\exp\left((p-1)\log(2) + C_{lin,1}(n+1)^{2-2\gamma}\right) + C_{lin,2}(n+1)^{p(2-2\gamma)},
\end{aligned}
$$

with

$$
\begin{aligned}
C_{lin,1} =& \frac{2pc_\gamma L_{\nabla G}}{\mu}\|S_0^{-1}\|\left(2 + \mathbb{E}\left[\|X_{n+1}\|^8\right]^{1/4}\right) + \frac{2pc_\gamma^2 L_{\nabla G}}{\mu}\|S_0^{-2}\|\mathbb{E}\left[\|X_{n+1}\|^8\right]^{1/2}, \qquad (91)\\
C_{lin,2} =& 2^{2p-1}\mathbb{E}\left[|\epsilon|^{2p}\right]L_{\nabla G}^p\left(c_\gamma\|S_0^{-1}\| + c_\gamma^2\|S_0^{-2}\|\right)^p.
\end{aligned}
$$

We deduce that

$$
\mathbb{E}\left[V_n^p\right] \leq \exp\left((p-1)\log(2)n + C_{lin,1}\frac{n^{3-2\gamma}}{3-2\gamma}\right)\left(1 + C_{lin,2}\frac{n^{(p+1)-p\gamma}}{(p+1)-p\gamma}\right)\mathbb{E}\left[V_0^p\right].
$$

## D  Proof of technical Lemma and Propositions for generalized linear model

### D.1  Proof of Lemma 7.11

With the help of inequality equation 7, it comes

$$
\|\bar{S}_n\| \leq \frac{1}{n+1}\|S_0\| + \frac{L_{\nabla l}}{n+1}\sum_{i=1}^n\|X_i\|^2 + \frac{\sigma d}{n+1}\sum_{i=1}^n\|Z_i\|^2.
$$

with $Z_i = e_{i[d]+1}$. Then, a similar proof as the one of Lemma 7.7 yields that for $\lambda_0 = \left(2L_{\nabla l}\mathbb{E}\left[\|X\|^2\right] + 2\sigma\right)^{-1}$,

$$
\begin{aligned}
& \mathbb{P}\left[\lambda_{\min}\left(\overline{S}_n^{-1}\right) < \lambda_0\right] \\
& \leq \mathbb{P}\left[\frac{\|S_0\|}{n} + \frac{L_{\nabla l}}{n}\sum_{i=1}^n\left(\|X_i\|^2 - \mathbb{E}\left[\|X\|^2\right]\right) + \frac{\sigma}{n}\sum_{i=1}^n\left(\|Z_i\|^2 - 1\right) > L_{\nabla l}\mathbb{E}\left[\|X\|^2\right] + \sigma\right].
\end{aligned}
$$

Then, by Markov inequality for $p \geq 1$, we then get

$$
\begin{aligned}
\mathbb{P}\left[\lambda_{\min}\left(\overline{S}_n^{-1}\right) < \lambda_0\right] \leq & \frac{\mathbb{E}\left[\left(\frac{1}{n}\|S_0\| + \frac{1}{n}\sum_{i=1}^n L_{\nabla l}\left(\|X_i\|^2 - \mathbb{E}\left[\|X\|^2\right]\right) + \sigma\left(\|Z_i\|^2 - 1\right)\right)^p\right]}{\left(L_{\nabla l}\mathbb{E}[\|X\|_2^2] + \sigma\right)^p} \\
\leq & \frac{2^{p-1}}{\left(L_{\nabla l}\mathbb{E}\left[\|X\|^2\right] + \sigma\right)^p}\left(n^{-p}\|S_0\|^p + C_1(p)n^{1-p}\mathbb{E}\left[|T|^p\right] + C_2(p)n^{-p/2}\left(\mathbb{E}\left[\|T\|^2\right]\right)^{p/2}\right),
\end{aligned}
$$

with $T = L_{\nabla l}\left(\|X\|^2 - \mathbb{E}\left[\|X\|^2\right]\right) + \sigma\left(\|Z\|^2 - 1\right)$.

### D.2  Proof of Proposition 7.5

One directly has for all $n \geq 2d$

$$
\lambda_{\min}\left(\overline{S}_n\right) \geq \frac{\lfloor n/d\rfloor\sigma}{(n+1)} \geq \frac{n+1-d}{d(n+1)}\sigma \geq \frac{1}{2d}\sigma,
$$

and $\overline{S}_n \geq \frac{1}{2d} S_0$ for $n \leq 2d - 1$, so that

$$\sup_{n \geq 1} \left\| \bar{S}_n^{-1} \right\| \leq 2d \max \left\{ \frac{1}{\sigma}, \left\| S_0^{-1} \right\| \right\}.$$

### D.3 Proof of Proposition 7.7

Let us denote

$$H(\theta_\sigma) = \mathbb{E} \left[ \nabla_h^2 \ell \left( Y, \theta_\sigma^T X \right) X X^T \right] \qquad \text{and} \qquad \overline{H}_n = \frac{1}{n+1} \left( S_0 + \sum_{i=0}^{n-1} \nabla_h^2 \ell \left( Y_{i+1}, \langle \theta_i, X_{i+1} \rangle \right) X_{i+1} X_{i+1}^T \right).$$

One can decompose $\overline{H}_n - H(\theta_\sigma)$ as

$$\begin{aligned} \overline{H}_n - H(\theta_\sigma) =& \frac{1}{n+1} \sum_{i=0}^{n-1} \nabla_h^2 \ell \left( Y_{i+1}, \langle \theta_i, X_{i+1} \rangle \right) X_{i+1} X_{i+1}^T + \frac{1}{n+1} S_0 - H(\theta_\sigma) \\ =& \frac{1}{n+1} \sum_{i=0}^{n-1} \nabla_h^2 \ell \left( Y_{i+1}, \langle \theta_i, X_{i+1} \rangle \right) X_{i+1} X_{i+1}^T - H(\theta_i) \\ &+ \frac{1}{n+1} \sum_{i=0}^{n-1} \left( H(\theta_i) - H(\theta_\sigma) \right) + \frac{1}{n+1} \left( S_0 - H(\theta_\sigma) \right). \end{aligned}$$

Let us now give a rate of convergence of each term on the right-hand side of previous equality. Set $M_n := \sum_{i=0}^{n-1} \left( \nabla_h^2 \ell \left( Y_{i+1}, \theta_i^T X_{i+1} \right) X_{i+1} X_{i+1}^T - H(\theta_i) \right)$. Since $\mathbb{E} \left[ \nabla_h^2 \ell \left( Y_{i+1}, \theta_i^T X_{i+1} \right) X_{i+1} X_{i+1}^T | \mathcal{F}_i \right] = H(\theta_i)$, where $(\mathcal{F}_i)$ is the $\sigma$-algebra generated by the sample, i.e $\mathcal{F}_i := \sigma \left( (X_1, Y_1), \dots, (X_i, Y_i) \right)$. Then, $(M_n)_{n \geq 1}$ is a martingale and thus

$$\frac{1}{(n+1)^2} \mathbb{E} \left[ \|M_n\|^2 \right] \leq \frac{1}{(n+1)^2} \sum_{i=0}^{n-1} \mathbb{E} \left[ \left\| \left( \nabla_h^2 \ell \left( Y_{i+1}, \theta_i^T X_{i+1} \right) X_{i+1} X_{i+1}^T - H(\theta_i) \right) \right\|^2 \right] \leq \frac{L_{\nabla l}^2 \mathbb{E} \left[ \|X\|^4 \right]}{n}$$

It then remains to handle $\frac{1}{n+1} \sum_{i=0}^{n-1} \left( H(\theta_i) - H(\theta_\sigma) \right)$. With the help of Assumption **(GLM1)**, one has

$$\begin{aligned} \mathbb{E} \left[ \left\| \frac{1}{n+1} \sum_{i=0}^{n-1} \left( H(\theta_i) - H(\theta_\sigma) \right) \right\|^2 \right] \leq& \frac{1}{n} \sum_{i=0}^{n-1} \mathbb{E} \left[ \| H(\theta_i) - H(\theta_\sigma) \|^2 \right] \\ \leq& \frac{L_{\nabla^2 L}^2}{n} \sum_{i=0}^{n-1} \mathbb{E} \left[ \|\theta_i - \theta_\sigma\|^2 \right] \leq \frac{L_{\nabla^2 L}^2}{\sigma n} \sum_{i=0}^{n-1} v_{i,\text{GLM}}, \end{aligned}$$

with $v_{i,\text{GLM}}$ defined in Proposition 7.6. Then, since

$$\left\| \frac{d\sigma}{n} \sum_{i=1}^{n} e_{i[d]+1} e_{i[d]+1}^T - \sigma I_d \right\|^2 = \left\| \frac{d\sigma}{n} \sum_{i=d\lfloor \frac{n}{d} \rfloor}^{n} e_{i[d]+1} e_{i[d]+1}^T + \left( \frac{d\sigma}{n} \left\lfloor \frac{n}{d} \right\rfloor - \sigma \right) I_p \right\|^2$$

and

$$\frac{d^2 \sigma^2}{n^2} \left\| \sum_{k=d\lfloor \frac{n}{d} \rfloor}^{n} e_{i[d]+1} e_{i[d]+1}^T \right\|^2 \leq \frac{d^2 \sigma^2}{n^2} \left( n - d \left\lfloor \frac{n}{d} \right\rfloor \right) \sum_{k=d\lfloor \frac{n}{d} \rfloor}^{n} \left\| e_{i[d]+1} e_{i[d]+1}^T \right\|^2 \leq \frac{d^4 \sigma^2}{n^2},$$

it comes

$$\left\| \overline{S}_n - H_\sigma \right\|^2 \leq \frac{4}{n} \left( L_{\nabla l}^2 \mathbb{E} \left[ \|X\|^4 \right] + \frac{L_{\nabla^2 L}^2}{\sigma} \sum_{i=0}^{n-1} v_{i,\text{GLM}} + \frac{1}{n} \| S_0 - H(\theta_\sigma) \|^2 \right) + \frac{16 d^4 \sigma^2}{n^2}$$

Now, notice as in Lemma 7.9 that

$$\left\|\overline{S}_n^{-1} - H_\sigma^{-1}\right\| = \left\|\overline{S}_n^{-1}(H_\sigma - \overline{S}_n)H_\sigma^{-1}\right\| \leq \left\|\overline{S}_n^{-1}\right\| \|H_\sigma - \overline{S}_n\| \left\|H_\sigma^{-1}\right\|,$$

which yields, thanks to Proposition 7.5

$$\mathbb{E}\left[\left\|\overline{S}_n^{-1} - H_\sigma^{-1}\right\|^2\right] \leq \frac{C_{S,\sigma}^2}{\sigma^2}\mathbb{E}\left[\left\|\overline{S}_n - H_\sigma\right\|^2\right],$$

i.e one has

$$\mathbb{E}\left[\left\|\overline{S}_n^{-1} - H_\sigma^{-1}\right\|^2\right] \leq \frac{4C_{S,\sigma}^2}{\sigma^2 n}\left(L_{\nabla l}^2\mathbb{E}\left[\|X\|^4\right] + \frac{L_{\nabla^2 L}^2}{\sigma}\sum_{i=0}^{n-1} v_{i,\mathrm{GLM}} + \frac{1}{n}\|S_0 - H(\theta_\sigma)\|^2\right) + \frac{16d^4 C_{S,\sigma}^2}{n^2}. \quad (92)$$

## E  How to verify (GLM3) for the logistic regression

Remark that $\theta_\sigma$ is the unique solution to $\mathbb{E}\left[\nabla_h \ell\left(Y, X^T\theta_\sigma\right)X + \sigma\theta_\sigma\right] = 0$, so that

$$\mathbb{E}\left[\left|\nabla_h l\left(Y, X^T\theta_\sigma\right)X_k + \sigma(\theta_\sigma)_k\right|^2\right] = Var\left[\nabla_h l\left(Y, X^T\theta_\sigma\right)X_k\right].$$

For the logistic regression, we have $Y \in \{-1, 1\}$ and $\nabla_h l\left(Y, X^T\theta_\sigma\right) = \frac{-Y\exp(-Y\theta_\sigma^T X)}{1+\exp(-Y\theta_\sigma^T X)}$, and thus we need to get a lower bound on the variance of $\frac{-YX_k\exp(-Y\theta_\sigma^T X)}{1+\exp(-Y\theta_\sigma^T X)}$ for all $1 \leq k \leq d$. To guarantee Assumptions **(GLM3)**, we impose a minimal randomness on $(X, Y)$ given by the existence of $a < b$, $\eta, \epsilon > 0$ $M > 1$ and for all $1 \leq k \leq d$ an event $A_k \in \sigma(Y, X_{k'})$ with $\mathbb{P}\left[A_k \cap \{M^{-1} \leq |X_k| \leq M\}\right] > \eta$ such that on $A_k$ we have

$$\mathbb{P}\left[\sum_{i\neq k} X_i(\theta_\sigma)_i < a | Y, X_k\right] > \epsilon \quad \text{and} \quad \mathbb{P}\left[\sum_{i\neq k} X_i(\theta_\sigma)_i > b | Y, X_k\right] > \epsilon.$$

In particular, since $u \mapsto \frac{\alpha\exp(-Yu)}{1+\beta\exp(-yu)}$ is $\mathcal{C}^1$ and monotonic for all $\alpha, \beta > 0$ and $y \in \{-1, 1\}$, for $M$ small enough, there exist constants $c^-, c^+$ explicitly depending on $M, (\theta_\sigma)_k, a, b$ such that on $B_k := A_k \cap \{M^{-1} \leq |X_k| \leq M\}$,

$$\mathbb{P}\left[\frac{-YX_k\exp(-Y\theta_\sigma^T X)}{1+\exp(-Y\theta_\sigma^T X)} > c^+ | Y, X_k\right] > \epsilon,$$

and

$$\mathbb{P}\left[\frac{-YX_k\exp(-Y\theta_\sigma^T X)}{1+\exp(-Y\theta_\sigma^T X)} < c^- | Y, X_k\right] > \epsilon.$$

We deduce that on the event $B_k$ we have

$$Var\left[\frac{-YX_k\exp(-Y\theta_\sigma^T X)}{1+\exp(-Y\theta_\sigma^T X)}\bigg| Y, X_k\right] \geq \epsilon\left(\frac{c^+ - c^-}{2}\right)^2.$$

Hence,

$$Var\left[\frac{-YX_k\exp(-Y\theta_\sigma^T X)}{1+\exp(-Y\theta_\sigma^T X)}\right] \geq \mathbb{E}\left[\mathbf{1}_B Var\left[\frac{-YX_k}{1+\exp(-Y\theta_\sigma^T X)}\bigg| Y, X_k\right]\right] \geq \eta\epsilon\left(\frac{c^+ - c^-}{2}\right)^2,$$

and we can choose

$$\alpha_\sigma = \eta\epsilon\left(\frac{c^+ - c^-}{2}\right)^2.$$

# F  Counter-example for the quadratic convergence of the stochastic Newton algorithm without regularization

We show here that even in the simplest case $d = 1$, stochastic Newton algorithm may not converge in quadratic mean. Suppose that we define here the naive Newton adaptive matrix $A_n$

$$A_n = \left[ \frac{1}{n+1} \left( Id + \sum_{i=0}^{n-1} \nabla_h^2 g(X_{i+1}, \theta_i) \right) \right]^{-1}.$$

Recall that is known (Boyer & Godichon-Baggioni, 2020) that $\theta_n$ converges almost-surely to the minimizer $\theta_0$ at speed $n^{-\gamma}$ for $\gamma \in (1/2, 1)$.

**Counter-example with $\nabla g$ almost everywhere defined**

Set $g((x, y), \theta) = (x\theta)^2 + y\lfloor\theta\rfloor\theta$ and let $(X, Y)$ be a random vector with independent coordinates such that $X \simeq Ber(1/2)$ and $\mathbb{P}[Y = 1] = \mathbb{P}[Y = -1] = 1/2$. Then, $G(\theta) = \mathbb{E}\left[X^2\right]\theta^2 + \mathbb{E}[Y]\lfloor\theta\rfloor\theta = \theta^2/2$ and we have Lebesgue almost surely $\nabla_h g((x, y), h) = 2x^2 h + y\lfloor h\rfloor$ and $\nabla_h^2 g((x, y), h) = 2x^2$.

Let $n \geq 1$. Then, $\mathbb{P}[X_1 = 0, \ldots, X_n = 0, Y_1 = -1, \ldots, Y_n = -1] = 2^{-2n}$ and on the event $\{X_1 = 0, \ldots, X_n = 0, Y_1 = -1, \ldots, Y_n = -1\}$, as long as $\theta_k \notin \mathbb{N}$ for all $k \geq 0$ (which will be temporarily assumed),

$$A_k^{-1} = \frac{1}{k} \left( 1 + \sum_{i=0}^{k-1} 2X_{i+1}^2 \right) = \frac{1}{k}.$$

Hence, $A_k = k$ and $(\theta_k)_{1 \leq k \leq n}$ is defined recursively by

$$\theta_k = \theta_{k-1} - \gamma_k A_k \lfloor\theta_{k-1}\rfloor Y_k = \theta_{k-1} + k\gamma_k \lfloor\theta_{k-1}\rfloor.$$

If $\gamma_k = k^{-\alpha}$ for some $\alpha < 1$, we then have $k\gamma_k = k^{1-\alpha}$, and thus for $\theta_0 > 1$

$$\theta_k \geq (1 + k^{1-\alpha}/2)\theta_{k-1}.$$

We deduce that $\theta_n \geq \prod_{k=1}^n (1 + k^{1-\alpha}/2) \geq (n!)^{1-\alpha} 2^{-n}$. In particular,

$$\mathbb{E}\left[\|\theta_n - \theta_0\|^2\right] \geq 2^{-3n}(n!)^{1-\alpha} \xrightarrow{n \to \infty} \infty$$

when $\theta_k \notin \mathbb{N}$ for all $k \geq 0$. Since for each $k \geq 1$, $\theta_k \notin \mathbb{N}$ for almost every $\theta_0 \in (1, 2]$, the latter hypothesis holds for Lebesgue almost every choice of $\theta_0 \in ]1, 2]$.

**Counter-example with $\nabla g$ continuous**

Let $f$ be such that $f''(\theta) = \mathbf{1}_{\mathbb{Z}+]-1/3,1/3[}$, and set $g((x, y), \theta) = (x\theta)^2 + yf(\theta)$. Let $(X, Y)$ be a random vector with independent coordinates satisfying $X \simeq Ber(1/2)$ and $Y \sim \mathcal{U}([-2, 2])$. Then, $G(\theta) = \mathbb{E}\left[X^2\right]\theta^2 + \mathbb{E}[Y]f(\theta) = \theta^2/2$ and $\nabla_h g((x, y), \theta) = 2x^2\theta + yf'(\theta)$. Then, $A_0 = 1$,

$$A_k^{-1} = \frac{k}{k+1}(A_{k-1}^{-1} + 2X_k^2 + f''(\theta_{k-1})Y_k)$$

for $k \geq 1$ and

$$\theta_n = \theta_{n-1} - A_{n-1}\gamma_n \nabla_h g((X_n, Y_n), \theta_{n-1}) = \theta_{n-1} - A_{n-1}\gamma_n(2X_n^2\theta_{n-1} + Y_n f'(\theta_{n-1})).$$

Set $\theta_0 = 3/2$ and $\gamma_k = k^{-\gamma}$ for $k \geq 1$, and consider $(X_i, Y_i)_{0 \leq i \leq n}$ satisfying the following conditions:

- $X_i = 0$ for all $1 \leq i \leq n$, which yields $\mathbb{P}[X_1 = 0, \ldots, X_n = 0] = 2^{-n}$ and for all $k \geq 1$,

$$\theta_k = \theta_{k-1} - A_{k-1}\gamma_k Y_k f'(\theta_{k-1}).$$

- $\theta_{k-1}$ being known, $Y_k \in \frac{1}{\gamma_k A_{k-1} f'(\theta_{k-1})} \left((\mathbb{Z}+]1/3, 2/3[) - \theta_{k-1}\right) \cap [-2, -1] := T_k$ (remark that $T_k$ will be shown to be non-empty).

**Lemma F.1.** *The following facts hold for $k \geq 1$.*

1. *$\theta_k \geq k + 1$,*

2. *$A_k = k + 1$,*

3. *with $\ell$ denoting the Lebesgue measure,*

$$\ell \left( \frac{1}{\gamma_k A_{k-1} f'(\theta_{k-1})} \left((\mathbb{Z}+]1/3, 2/3[) - \theta_{k-1}\right) \cap [-2, -1]\right) \geq 1/6.$$

*Proof.* We will prove those three facts by induction on $k \geq 1$. For $k = 1$, we have $A_0 = \gamma_1 = 1$ and $f'(3/2) = 1$ so that $\theta_1 = 3/2 - Y_1$. Since

$$T_1 = \frac{1}{A_0 \gamma_1 f'(\theta_0)} \left((\mathbb{Z}+]1/3, 2/3[) - \theta_{k-1}\right) \cap [-2, -1] = (-3/2 + \mathbb{Z}+]1/3, 2/3[) \cap [-2, -1]$$
$$= ] - 7/6, -1] \cup [-2, -11/6[,$$

$\ell(T_1) \geq 1/3$. On the other hand, for $Y_1 \in T_1$, $\theta_1 \geq 3/2 + 1 \geq 2$.

Let us show the induction. Set $k \geq 2$ and suppose the result is true for $l \leq k - 1$. Then $\theta_l \in \mathbb{Z} + [1/3, 2/3]$ for all $l \leq k$, which implies that $A_{k-1} = k - 1$. Hence,

$$\theta_k = \theta_{k-1} + k^{1-\gamma} Y_k f'(\theta_{k-1}).$$

By induction, $\theta_{k-1} \geq k$, and since $f'(\theta) \geq \theta/2$ for $\theta \geq 0$,

$$T_k = ((b + a\mathbb{Z} + a]1/3, 2/3[) \cap [-2, -1]$$

with $a = 1/(A_{k-1} \gamma_k f'(\theta_{k-1})) \leq \frac{2}{k^{2-\gamma}} \leq 1$ and $b = -\theta_{k-1}/a$. We deduce by pigeonhole principle that $\ell(T_k) \geq 1/3 - \frac{1}{2} \cdot \frac{a}{3} \geq 1/6$. Finally, for $Y_k \in A_k$ we have $Y_k \leq -1$ so that

$$\theta_k \geq k + \frac{1}{2} k^{2-\gamma} \geq k + 1.$$

$\square$

By the previous result,

$$\mathbb{P}\left[X_1 = \cdots = X_n = 0, Y_1 \in T_1, \ldots, Y_n \in T_n\right] \geq 2^{-n} \cdot 6^{-n} = 12^{-n}.$$

Moreover, from what we showed previously, on this event we have for $1 \leq k \leq n$

$$\theta_k = \theta_{k-1} - Y_k A_{k-1} \gamma_k f'(\theta_{k-1}) \geq \theta_{k-1} + k^{2-\gamma}/2 \geq \theta_{k-1} k^{2-\gamma}/2.$$

We deduce that $\theta_n \geq \theta_{k-1}(k-1)^{1-\gamma}/3 \geq (n!)^{2-\gamma}/2^n$. In particular,

$$\mathbb{E}\left[\|\theta_n - \theta_0\|^2\right] \geq (n!)^{2-\gamma}/24^n \xrightarrow{n \to \infty} \infty.$$

Remark that the latter result can be easily adapted to get a counter-example with $g$ as smooth as desired.

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
