# OpenReview forum: "Non asymptotic analysis of Adaptive stochastic gradient algorithms and applications"
_TMLR — Accepted by TMLR_

### Review · Reviewer_rYhH · 2025-02-25

**Summary Of Contributions:**

This paper provides a non-asymptotic analysis of adaptive stochastic gradient algorithms for strongly convex objective functions. The main contributions are:

1. Establishing convergence rates for adaptive methods when the adaptation matrix may diverge (with controlled divergence bounds)
2. Deriving standard convergence rates under the additional assumption that the adaptation matrix has uniformly bounded fourth-order moments
3. Providing a general framework for analyzing both stochastic Newton and Adagrad algorithms
4. Applying the theoretical results to practical examples including linear regression and ridge-regularized generalized linear models
5. Handling the technically challenging case of unbounded gradients, which many previous analyses avoid

The authors evaluate their theoretical framework on two important application domains: linear models and generalized linear models (including logistic regression), demonstrating both the versatility and applicability of their approach.

**Audience:**

Yes

**Broader Impact Concerns:**

This paper is primarily theoretical in nature, with limited direct societal implications. The improved understanding of adaptive optimization methods could lead to more efficient and effective machine learning algorithms, indirectly benefiting applications across various domains.

**Claims And Evidence:**

Yes

**Requested Changes:**

### Recommendations for Enhancement
1. **Consider including numerical experiments**: The theoretical contribution would be strengthened by empirical validation demonstrating the tightness of the derived bounds. Comparisons between standard implementations and the proposed modifications on benchmark datasets would illustrate the practical implications of the theoretical results.
2. **Enhancing with intuitive interpretations**: The paper would benefit from additional intuitive explanations of key technical results and assumptions, particularly highlighting their practical implications for algorithm design and implementation. This would make the sophisticated mathematical content more accessible to practitioners.
3. **Further comparison with existing literature**: The valuable theoretical contribution could be highlighted more effectively through a comprehensive comparison with existing bounds in the literature. Clarifying how these new results improve upon previous work (particularly Défossez et al. (2020) and  Gower et al., (2019)) would contextualize the advances made.
4. **Guidance on assumption verification**: Practitioners would benefit from additional guidance on verifying the technical assumptions in practice, especially (H1a) and (H1b). A discussion of scenarios where these assumptions might be violated would provide valuable practical insight.
5. **Exploration of relaxed assumptions**: Considering possibilities for relaxing the strong convexity assumption, at least for special cases, would broaden the impact of this work. A discussion of how the results might extend to weakly convex or non-convex settings would be valuable for many modern machine learning applications.
6. **Discussion of computational considerations**: Adding insights on the computational overhead introduced by the proposed modifications would provide helpful guidance for practitioners implementing these methods. This practical perspective would complement the strong theoretical foundation already established.
7. **Standardize capitalization in references**: The bibliography would benefit from consistent capitalization, particularly for mathematical notation and acronyms. For instance, the asymptotic notation "o (1/n)" should be capitalized to "O (1/n)" in the Bach and Moulines reference, and "Sgd" should be standardized to "SGD" in the Gower et al. reference.

**Strengths And Weaknesses:**

### Strengths:

1. **Technical novelty**: The paper addresses a significant gap in the existing literature by providing non-asymptotic convergence rates for adaptive stochastic gradient methods without requiring uniformly bounded gradients, a condition often violated in practice.
2. **Comprehensive theoretical framework**: The analysis covers multiple adaptive algorithms (Adagrad, stochastic Newton) within a unified framework, enabling direct comparison of their convergence properties.
3. **Practical relevance**: The theoretical results are directly applied to common machine learning models including linear regression and generalized linear models, demonstrating real-world applicability.
4. **Mathematical rigor**: The paper includes detailed proofs and careful handling of technical conditions, providing a solid theoretical foundation for the results.
5. **Treatment of edge cases**: The authors explicitly address potential divergence scenarios and provide practical modifications to ensure convergence.
6. **Clear and well-structured writing**: The paper presents complex mathematical concepts and proofs in a clear, organized manner, making the technical content accessible despite its sophistication.
### Weaknesses:
1. **Absence of experimental validation**:
   The paper lacks numerical experiments to validate the tightness of the theoretical bounds or illustrate their practical implications. While the focus is theoretical, empirical validation would strengthen the paper considerably by showing whether the derived rates match practical performance.
2. **Restrictive assumptions**:
   Some of the technical assumptions, particularly (H1b) regarding the spectral norm of the adaptation matrix, are acknowledged by the authors as potentially restrictive. The provided workarounds (modifications to the algorithms) might alter their practical performance in ways not thoroughly analyzed.
3. **Limited scope to strongly convex objectives**:
   The analysis is restricted to strongly convex objective functions, which limits applicability to many modern machine learning problems involving non-convex optimization.
4. **Insufficient comparison with existing bounds**:
   While the paper mentions some previous work, it would benefit from a more thorough comparison of the derived bounds with existing results in the literature, highlighting specific improvements.
5. **Insufficient intuition for technical results**:
   The mathematical development, while rigorous, could be better complemented with intuitive explanations of key results and their practical significance, especially for practitioners.

---

> ### Author Response · Authors · 2025-04-04
> **Response**
>
> We warmly thank you for your constructive feedbacks. We will now address your remarks one by one:
>
> Weakness:
>
> 1. We have now added a simulation section in which we examine two examples: linear regression and ridge logistic regression. The simulation results appear to support our theoretical findings.
>
>
> 2. Upon conducting a more thorough review of the literature, we realized that the assumptions we aim to relax are, in fact, less restrictive than those commonly used in the field.
>
>     For instance, in the case of Newton-type algorithms, it is often assumed that the Hessian estimator (and its inverse) is uniformly bounded above and below by constants, whereas in our framework, we only assume a lower bound given by a decreasing sequence and an upper bound given by an increasing sequence.
>
>     Similarly, in Adagrad, it is often assumed that gradients are bounded, which ensures that the conditioning matrix is uniformly lower-bounded by a constant. Additionally, a bias term is typically introduced to ensure the positivity of the diagonal elements of the variance estimator of the gradient, which in turn guarantees that the conditioning matrix remains bounded above by a constant.
>
> 3.  We fully agree that extending our results to the non-convex case is particularly crucial, especially for deep learning applications. Our goal with this work was to lay the groundwork for the theoretical study of adaptive algorithms in a setting that is typically considered simple, namely the strongly convex case. Through our proofs, we realized that even in this supposedly simple framework, there were many challenges to overcome. This, in turn, has paved the way for obtaining results in the non-convex setting as a continuation of this work (see Surendran et al. (2024) for instance).
>
> 4. Thank you very much. We are now comparing our results with existing bounds. We observe that our results are quite analogous to these bounds and that, for instance, the Newton algorithm eliminates the dependence on the smallest eigenvalue in the exponential decay while also modifying the variance.
>
> 5. You are perfectly true, we have now added a sketch of the proof.
>
>
> Recommendations for enhancement:
>
> First of all, thank you very much for these insightful recommendations. We will now address them point by point.
>
> 1. We have now added a simulation section in which we examine two examples: linear regression and ridge logistic regression. The simulation results appear to support our theoretical findings.
>
>
> 2. We have now added a sketch of the proof at the beginning of Section 7. More specifically, we explain that the core of our results relies on a Taylor expansion of the objective function, which we make explicit. We then clarify that the remaining work consists in controlling the eigenvalues of the conditioning matrix.
>
>
>  3. Thank you very much for these references. We now provide a  discussion of the assumptions, particularly in comparison with the existing literature. For example, in Defossez et al., the authors obtain (weaker, but understandably so) results for non-convex functions, but under the assumption of bounded gradients. Their setting is therefore quite different from ours. Moreover, they do not address stochastic Newton methods.
>
> 4. Regarding assumption  (H1b), as mentioned, there are some standard transformations of the conditioning matrices that allow it to hold. We also emphasize that one should refer to the proofs of Theorems 4.1, 4.2, 5.1, and 5.2 to understand how to verify these assumptions in practice. Furthermore, we explicitly mention that a counterexample showing that $\theta_{n}$ can diverge if (H1b) does not hold is provided in Appendix F.
>
> 5. We have now added a conclusion in which we highlight that our work constitutes a first step in weakening assumptions on conditioning matrices. A natural avenue for future work would be to relax assumptions on the function to be minimized, extending the results to non-convex settings.
>
> 6. In practice, when applying the truncation
>
> $$
> \tilde{A}\_{n} = \frac{  || A\_{n}  ||\_{op}  , \beta_{n+1}  }{|| A\_{n} ||\_{op}  } A\_{n}
> $$
>
> it is preferable to use the Frobenius norm if the conditioning matrix is not diagonal (as in the case of stochastic Newton algorithms). Indeed, computing it only requires $O(d^{2})$ operations, which is of the same order as updating $\theta_{n+1}$ (if the conditioning matrix is not diagonal). More generally, all modifications introduced incur, at most, the same computational cost as updating $\theta_{n}$, i.e., $O(d^{2})$ for stochastic Newton algorithms and $O(d)$ for Adagrad.
>
> 7. Thank you for pointing this out. We have now corrected these typos.
>
>
> References:
>
> Surendran, S., Fermanian, A., Godichon-Baggioni, A., and Le Corff, S. (2024). Non-asymptotic analysis of biased adaptive stochastic approximation. \textit{Advances in Neural Information Processing Systems}, 37, 12897-12943.

---

### Review · Reviewer_xi3x · 2025-03-20

**Summary Of Contributions:**

This paper offers a comprehensive non-asymptotic analysis of adaptive stochastic gradient algorithms, with a particular focus on their convergence properties in strongly convex optimization problems.

**Audience:**

Yes

**Claims And Evidence:**

Yes

**Requested Changes:**

See the Weakness part.

**Strengths And Weaknesses:**

Overall, I think this paper makes good contributions.

While the theoretical insights are valuable, there are a few areas that could be improved:

1. **Empirical Validation**: The paper emphasizes theoretical results, which would be strengthened by empirical validation. The derived convergence rates are persuasive in theory, but including experiments would enhance the paper’s impact and provide practical insights.

2. **Clarification of Assumptions**: The paper discusses a variety of assumptions, which could benefit from clearer organization. When talking about related work, a table summarizing the assumptions used in each method, along with the corresponding convergence results, would provide a more straightforward comparison and enhance the clarity of the discussion.

3. **Extension to Non-Convex Problems**: The authors could explore the potential for extending the non-asymptotic analysis to a broader class of functions, beyond strongly convex ones. Such an extension would be particularly valuable for practitioners dealing with non-convex problems, which are prevalent in fields like deep learning and machine learning applications.

---

> ### Author Response · Authors · 2025-04-04
> **Response**
>
> We warmly thank you for your constructive feedbacks. We will now address your remarks one by one:
>
>
> 1. You are perfectly right, we added a section with simulation in the stochastic Newton and Adagrad case (Section 6 of the new manuscript).
>
> 2. Regarding the assumptions, we have now added a discussion comparing them with what is commonly done in the literature.
>         As for the idea of including tables summarizing the assumptions, we are not entirely convinced that it would be beneficial. The reason is that we are discussing different methods (Adagrad, Newton, and SGD), each with its own specificities and potentially distinct research communities, leading to a wide variety of slightly different assumptions.
>        For instance, our assumptions (A2) and (A4) are quite similar to the expected smoothness assumption. However, if you believe that including one or more tables is absolutely necessary, we will, of course, incorporate them accordingly.
>
> 3. We fully agree that extending our results to the non-convex case is particularly crucial, especially for deep learning applications. Our goal with this work was to lay the groundwork for the theoretical study of adaptive algorithms in a setting that is typically considered simple, namely the strongly convex case. Through our proofs, we realized that even in this supposedly simple framework, there were many challenges to overcome. This, in turn, has paved the way for obtaining results in the non-convex setting as a continuation of this work (see Surendran et al. (2024) for instance).
>
> References:
>
> Surendran, S., Fermanian, A., Godichon-Baggioni, A., and Le Corff, S. (2024). Non-asymptotic analysis of biased adaptive stochastic approximation. \textit{Advances in Neural Information Processing Systems}, 37, 12897-12943.

---

### Review · Reviewer_atrp · 2025-03-26

**Summary Of Contributions:**

This submission studies theoretical non-asymptotic convergence about adaptive optimization methods including AdaGrad and stochastic Newton algorithms. Particularly, if focus on strongly convex problems without bounded gradient assumptions. The analysis depend on assumptions of bounded higher-order moments of gradients (A1), twice continuously differentiable (A2), bounded Hessian (A3), and strong convexity (A4). Analysis has been applied to practical examples, including linear regression and regularized generalized linear models.

**Audience:**

Yes

**Broader Impact Concerns:**

NA.

**Claims And Evidence:**

Yes

**Requested Changes:**

See above.

**Strengths And Weaknesses:**

Strengths:
Provide analysis of AdaGrad and Newton algorithms without bounded gradient assumptions. Applied the results to linear regression and regularized generalized linear models.

Weakness:
1. The analysis only applies for strongly-convex cases. It would be beneficial to generalize the analysis to convex and non-convex regime, or discuss the challenges to do so.

2. What is the theoretically optimal choice of $\gamma$. Although both $\gamma\in(0, 1/2]$ and $\gamma\in[1/2, 1)$ have been analyzed, it seems unclear about what should be the choice of $\gamma$ to derive the best optimal convergence results. For example, according to Proposition 3.2, is it correct to claim that a larger $\gamma$, as long as $\gamma < 1$, is always better?

3. The following paper covered the Adabound algorithm without bounded gradient assumption, discussed in the Remark under Assumption 2. Please check:
guo et. al, On stochastic moving-average estimators for non-convex optimization (https://arxiv.org/pdf/2104.14840).

4. The obtained convergence rates $O(n^{-\gamma})$ seem to be slower than known results under bounded gradient assumptions since $\gamma < 1$.

5. There is a lack of experiments to demonstrate the analysis. For example, it would be helpful to include experiments where bounded gradient assumption does not hold and then analyze whether the behavior of the optimization process is consistent with the theoretical analysis. Also, it would helpful to include some ablation studies to investigate the role of $\gamma$.

Minor:
Line 6 under Introduction "nam a few" --> "name a few"

---

> ### Author Response · Authors · 2025-04-04
> **Response**
>
> We warmly thank you for your constructive feedbacks. We will now address your remarks one by one:
>
>
> 1. We fully agree that extending our results to the non-convex case is particularly crucial, especially for deep learning applications. Our goal with this work was to lay the groundwork for the theoretical study of adaptive algorithms in a setting that is typically considered simple, namely the strongly convex case. Through our proofs, we realized that even in this supposedly simple framework, there were many challenges to overcome. This, in turn, has paved the way for obtaining results in the non-convex setting as a continuation of this work (see Surendran et al. (2024) for instance).
>
> 2. With an infinite sample size, it is clear that the optimal choice for $ \gamma$ is to take it as close to 1 as possible (or even equal to1  in the case of Newton-type algorithms). However, our bounds show that, as with standard gradient algorithms, there are two phases of convergence:
>    - A first phase that depends on the initialization error and decreases at an exponential rate of approximately $e^{-n^{1-\gamma}}$. However, choosing $ \gamma $ close to 1 would hinder the convergence of this term.
>     - A second phase, once convergence has been reached, where the dominant term is of order $n^{-\gamma}$, and where it would be beneficial to take $ \gamma $ close to 1.
>     Thus, finding an optimal trade-off is challenging. However, as with stochastic gradient methods, an averaging step can accelerate convergence and achieve a rate of$ 1/n$. A remark has now been added in this regard.
>
> 3. Thank you very much for this reference, which we now cite. In fact, Theorems 3.1 and 3.2, as well as Propositions 3.1 and 3.2, remain valid if the strong convexity assumption is replaced by the condition that the function G satisfies the Polyak–Łojasiewicz inequality. A remark has now been added in this regard.
>
> 4. To the best of our knowledge, it is not possible to achieve a better rate with this type of step size in the online setting. However, once again, a widely used technique to accelerate convergence is to introduce an averaging step. A remark has now been added in this regard.
>
> 5. We have now added a simulation section in which we examine two examples: linear regression and ridge logistic regression. The simulation results appear to support our theoretical findings.
>
>
> References:
>
> Surendran, S., Fermanian, A., Godichon-Baggioni, A., and Le Corff, S. (2024). Non-asymptotic analysis of biased adaptive stochastic approximation. \textit{Advances in Neural Information Processing Systems}, 37, 12897-12943.

---

> ### Author Response · Authors · 2025-05-20
>
> We would like to once again thank the action editor and the reviewers for their constructive feedback. We have conducted one final thorough review of the proofs and have removed all red text. In addition, we now provide the codes.

---

### Comment · Action_Editor_CHjY · 2025-02-13
**Initial comments, gathering reviewers**

Dear Authors,

I'm now in the process of finding reviewers, and will let you know when reviewing begins.

As just an initial comment, I think it would be helpful to the reader if your convergence results for adaptive SGD were contrasted to the convergence results of plain-vanilla SGD under the same exact assumptions.

For instance, excluding your additional (H1) on the adaptive preconditioner, your Theorem 3.3 uses the exact same assumptions as Theorem 7 in the following paper:

Provable convergence guarantees for black-box variational inference, Justin Domke, Robert Gower, Guillaume Garrigos, Neurips 23
https://proceedings.neurips.cc/paper_files/paper/2023/file/d0bcff6425bbf850ec87d5327a965db9-Paper-Conference.pdf

If you know of an earlier reference that includes this, please use that instead.

I expect that your result will have worst constants, but that is ok. It is just to understand what you have lost/gained in comparison to plain SGD.

---

> ### Author Response · Authors · 2025-02-14
>
> Dear Action Editor,
>
> We are not certain that the referenced paper addresses exactly the same problem. Indeed, the proposed method in that work appears to operate in an offline setting, whereas our algorithms are designed for an online setting.
>
> Nevertheless, under analogous assumptions, for plain-vanilla SGD, there exists a positive constant \( C_0 \) such that (see, for instance, \texttt{https://hal.science/hal-00608041/document}):
>
> $$
> \mathbb{E}\left[ \left\| \theta_n - \theta \right\|^2 \right] \leq C_0 \exp \left( - \frac{\mu}{4} c_{\gamma} n^{1-\gamma} \right) + \frac{4 c_{\gamma} \sigma^2 }{\mu n^{\gamma}}
> $$
>
> and
>
> $$
> \mathbb{E} \left[ G(\theta_n) - G(\theta) \right] \leq \frac{L_{\nabla G} C_0}{2} \exp \left( - \frac{\mu}{4} c_{\gamma} n^{1-\gamma} \right) + \frac{2 L_{\nabla G} c_{\gamma} \sigma^2 }{\mu n^{\gamma}}.
> $$
>
> Best regards,

---

### Decision · Action_Editor_CHjY · 2025-05-08

**Recommendation:** Accept with minor revision

**Comment:**

This paper focuses on the non-asymptotic analysis of adaptive gradient algorithm, under the assumption that the loss is strongly convex. They also assume the full batch loss is smooth, and the stochastic gradients satisfy a rather general bound on the expected norm of gradient bound. The theory is very general, and applies to a large class of stochastic preconditioned methods so long as the stochastic proeconditioner satisfies some high probability bounds (And deterministic spectral bound). The several of generality is impressive, but as a consequence of being generalm the results are also loose as compared to SGD with no preconditioner.

To show case their general theory, the develop tighter instantiations of their theory for stochastic Newton and Adagrad. In particular I'm not aware of the work that analysis stochastic Newton in this generality, so it could be of interest to the stochastic second order community.

Because most reviewers were not able to verify the proofs, I ask that the authors take one more pass now, making the proofs as legible as possible, and removing the commented red text. I will then take one final pass over the paper, before having it published in TMLR.

**Audience:**

This paper will interested a subset of the TMLR community that are interested in analyzing non-asymptotic convergence rates of stochastic algorithms.

**Claims And Evidence:**

The paper is theoretical, with all the assumptions clearly spelt out, and all proof provided. Only the clarity of the proofs could be improved, since all the reviewers struggled to go through the proofs (most not managing).